# Recovering Latent Causal Factor for Generalization to Distributional Shifts

**Xinwei Sun**[*1], **Botong Wu**[†2], **Xiangyu Zheng**[†2], **Chang Liu**[1], **Wei Chen**[1], **Tao Qin**[1], **Tie-Yan Liu**[1]

[1] Microsoft Research Asia, Beijing, 100080
[2] Peking University, Beijing, 100871
{xinsun,changliu,wche,taoqin,tyliu}@microsoft.com, botongwu@pku.edu.cn

## Abstract

Distributional shifts between training and target domains may degrade the prediction accuracy of learned models, mainly because these models often learn features that possess only correlation rather than causal relation with the output. Such a correlation, which is known as "spurious correlation" statistically, is domain-dependent hence may fail to generalize to unseen domains. To avoid such a spurious correlation, we propose **La**tent **C**ausal **I**nvariance **M**odels (LaCIM) that specifies the underlying causal structure of the data and the source of distributional shifts, guiding us to pursue only causal factor for prediction. Specifically, the LaCIM introduces a pair of correlated latent factors: (a) causal factor and (b) others, while the extent of this correlation is governed by a domain variable that characterizes the distributional shifts. On the basis of this, we prove that the distribution of observed variables conditioning on latent variables is shift-invariant. Equipped with such an invariance, we prove that the causal factor can be recovered without mixing information from others, which induces the ground-truth predicting mechanism. We propose a Variational-Bayesian-based method to learn this invariance for prediction. The utility of our approach is verified by improved generalization to distributional shifts on various real-world data. Our code is freely available at https://github.com/wubotong/LaCIM.

## 1 Introduction

Current data-driven deep learning models, revolutionary in various tasks though, often exploit all types of correlations to fit data well. Among such correlations, there can be spurious ones corresponding to biases (*e.g.*, confounding bias due to the presence of a third unseen factor) inherited from the data provided. Such data-dependent spurious correlations can erode the prediction power on unseen domains with distributional shifts, which can cause serious consequences especially in safety-critical tasks such as healthcare.

Recently, there is a Renaissance of causality in machine learning, expected to pursue causal relationships [59] to achieve stable generalization across domains. The so-called area of "causality" is pioneered by Structural Causal Models [51], as a mathematical formulation of this metaphysical concept grasped in the human mind. The incorporation of these human priors about cause and effect endows the model with the ability to identify the causal structure [51] which entails not only the data but also the underlying process of how they are generated. To achieve causal modeling, the old-school methods [52, 10] directly causally related the output label $Y$ to a subset of *covariates X*, which is however *not* conceptually reasonable in applications with sensory-level data (*e.g. model pixels as causal factors of the output does not make sense in image classification [11]*).

---

[*]Corresponding author
[†]Work done during an internship at Microsoft Research Asia.

35th Conference on Neural Information Processing Systems (NeurIPS 2021).

For such applications, we rather adopt the manner of human visual perception [8, 9, 80] to causally relate the label $Y$ to unobserved abstractions denoted by $S$, *i.e.*, $Y \leftarrow S$. We further assume the existence of another non-causal latent factor (of $Y$) denoted as $Z$, that together with $S$ generate the input $X$: $X \leftarrow (S, Z)$. Such an assumption is similarly adopted in the literature [25, 27, 35, 75, 71]. To model shifts across domains, we allow $Z$ to be spuriously correlated with $S$ (hence also the output), as marked by the bidirected arrow in Fig. 1 (a). Taking image classification as an example, the $S$ and $Z$ respectively refer to object-related abstractions (*e.g.*, contour, texture) and contextual information (*e.g.*, background, view). Due to this correlation, the model can learn contextual information into prediction, which may fail to generalize to the domain such that this correlation is broken.

We encapsulate above assumptions into the skeleton illustrated in Fig. 1 (a), in which the spurious correlation between $S$ and $Z$ varies across domains, as marked by the red bi-directed arrow in Fig. 1 (b). Taking a closer inspection, such a domain-dependent spurious correlation is governed by an auxiliary domain variable $D$ in Fig. 1 (c), which causes the domain shifts. We call the set of causal models augmented with $D$ as **La**tent **C**ausal **I**nvariance **M**odels (LaCIM). Here, the "Causal Invariance" refers to $P(Y|S)$, which together with $P(X|S,Z)$, can be proved to be stable to the shifts across domains, under the assumptions embedded in the causal structure of LaCIM. Equipped with such an invariance, we prove that the $S$ and the ground-truth predictor: $P(Y|s^\star)$ for $x$ generated from $(s^\star, z^\star)$, are identifiable up to transformations that do not mix the non-causal information.

Under such an identifiability guarantee, we propose to learn the $P(Y|S)$ and $P(X|S,Z)$ by reformulating the Variational Auto-encoder (VAE) [37] to fit the joint distribution of the input and output variables from training domains. During the test stage, we first infer the value of $S$ by optimizing the estimated $P(X|S,Z)$ over latent space, followed by the learned $P(Y|S)$ for prediction. We first use simulated data to verify the correctness of the identifiability claim. Then, to demonstrate the utility, we test our approach on real-world data, consistently achieving better generalization to the new distribution; besides, we find that our inferred causal factor can be concentrated in highly explainable semantic regions for the task of image classification.

We summarize our contribution as follows: **Methodologically** (in sec. 4.1), we propose LaCIM in which the causal assumptions of two latent factors and the distributional shifts are incorporated; **Theoretically** (in theorem 4.4), we prove the identifiability of the causal factor and the ground-truth predicting mechanism; **Algorithmically** (in sec. 4.3), guided by the identifiability, we reformulate Variational Bayesian method to learn $P(X|S,Z), P(Y|S)$ for prediction; **Experimentally** (in sec. 5.2), our approach generalizes better to distributional shifts, compared with others.

## 2    Related Work

**Causality for Domain Generalization.** Due to its stable transferability, the concept of causality has been introduced in many recent works for domain generalization [39, 59, 52, 10, 40, 21, 68]. Most of these works learned the assumed (causal) invariance for generalizing to unseen domains. However, they suffer from either **i)** lacking explicit causal modeling; or **ii)** inappropriate causal relations made for the output. Specifically, for **i)**, the [39, 59] are still data-driven methods to learn stable correlation (*i.e.*, invariance) without incorporating causal assumptions [51] beyond data, which may impede its generalization to a broader set of domains; for **ii)**, the [52, 10, 40, 21, 68] causally relate the output with covariates, which is inappropriate for sensory-level data.

**Our Specification**. We explicitly incorporate the causal assumptions. Specifically, we introduce **i)** latent factors and separate them into the causal and the non-causal factor; **ii)** the domain variable $D$, as a selecting mechanism to generate the varied $S$-$Z$ correlation across domains. Such a causal modeling makes it possible to recover the causal factor $S$ for generalization. In independent and concurrent works, [75] and [28] also explore latent variables in causal relation. As comparisons, [75] did not differentiate $S$ from $Z$. The spurious correlation in [28] is limited in the correlation between domains and the output; while it is allowed in our setting to exist in a single domain, which is more aligned with real scenarios, *e.g.*, the dog is more associated with grass than snow in a domain when most samples are collected in sunny morning.

**Other Conceptually Related Works: i)** transfer learning that leverages invariance in the context of domain adaptation [60, 81, 17] or domain generalization [43, 63]; **(ii)** causal inference [51, 53] which builds structural causal models and define intervention (*a.k.a*, "do-calculus") for cause-effect reasoning and counterfactual learning; and **(iii)** latent generative model that assumes generation from latent space to observed data [37, 71] *but* aims at learning generator in the unsupervised scenario.

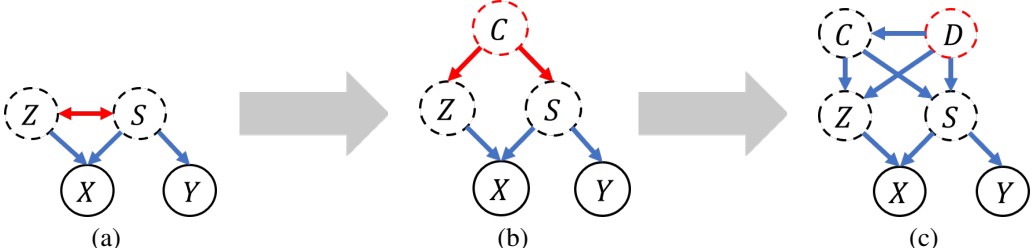

Figure 1: The introduction of *Latent Causal Invariant Models*, a set of SCMs augmented with a domain variable $D$ in (c), starting from the skeleton in (a). Specifically, from (a) to (b): the spurious correlation between $S$ and $Z$ is due to an unobserved confounder $C$, with distributional shifts referring to change of $P(C)$ and $P(S, Z|C)$ across domains; from (b) to (c): we ascribe such shifts to the domain variable $D$, as the source of distributional shifts. Here, the observed and unobserved variables are respectively marked by solid and dot circles; the directed arrow ("→" or "←") represents direct causal relation; while the bidirected arrows ("↔") represent the spurious correlation that can vary across domains; the red and blue respectively means the invariant and varied values/distributions.

## 3 Preliminaries

**Problem Setting.** Let $X, Y$ respectively denote the input and output variables. The training data $\{\mathcal{D}^e\}_{e \in \mathcal{E}_{\text{train}}}$ are collected from multiple environments $e \in \mathcal{E}_{\text{train}}$, where each $e$ is associated with a distribution $\mathrm{P}^e(X, Y)$ over $\mathcal{X} \times \mathcal{Y}$ and $\mathcal{D}^e := \{x_i^e, y_i^e\}_{i \in [n_e]} \overset{i.i.d}{\sim} \mathrm{P}^e$ with $[k] := \{1, ..., k\}$ for any $k \in \mathbb{Z}^+$. Our goal is to learn a robust predictor $f : \mathcal{X} \to \mathcal{Y}$ that only exploit the causal factor for prediction and generalize well to all domains $\mathcal{E} \supset \mathcal{E}_{\text{train}}$. We use respectively upper, lower case letter and Cursive letter to denote the random variable, the instance and the space, *e.g.*, $a$ is an instance in the space $\mathcal{A}$ of random variable $A$. For $\mathcal{A} := f(\mathcal{X}) \cap \mathcal{B}$ with $\mathcal{B} := \mathbb{R}^p[i_1] \times \mathbb{R}^p[i_2] \times ... \times \mathbb{R}^p[i_k]$, the $[f(x)]_{\mathcal{A}}$ denotes the $f(x)$ restricted on dimensions of $\mathcal{A}$, *i.e.*, $[f(x)]_{\mathcal{A}} := [f_{i_1}(x), ..., f_{i_k}(x)]$. The Sobolev space $W^{k,p}(\mathcal{A})$ contains all $f$ such that $\int_{\mathcal{A}} \left| \partial_A f^{\alpha} |_{A=a} \right|^p d\mu(a) < \infty, \forall \alpha \leq k$.

**Structural Causal Model.** The structural causal model (SCM) is defined as a triplet $M := \langle G, \mathcal{F}, P(\varepsilon) \rangle$, in which **i)** the causal structure $G := (V, E)$ ($V, E$ respectively denote the node and edge set) is described by a directed acyclic graph (DAG); **ii)** the structural equations $\mathcal{F} := \{f_k\}_{V_k \in V}$ are autonomous, *i.e.*, intervening on $V_k$ does not affect others, based on which we can define the *do-*operator and calculate the causal effect; **iii)** the $P(\varepsilon)$ are probability measure for exogenous variables $\{\varepsilon_k\}_k$. By assuming independence among $\{\varepsilon_k\}_k$, we obtain according to *Causal Markov Condition* that each $P$ that is compatible with $G$ has $\mathrm{P}(\{V_k = v_k\}_{V_k \in V}) = \Pi_k \mathrm{P}(V_k = v_k | Pa(k) = pa(k))$. An acyclic directed mixed graph (ADMG) can further allow the existence of bidirectional arrows $\leftrightarrow$, meaning the spurious correlation between two variables connected.

## 4 Methodology

We first incorporate the causal assumptions into LaCIM in sec. 4.1. Under such assumptions, we identify the invariant distributions $P(X|S, Z)$ and $P(Y|S)$, which are repectively dubbed as *generative invariance* and *causal invariance* that are robust to domain shifts. Equipped with these invariances, we in sec. 4.2 show that the causal factor can be identified without mixing information from non-causal one during prediction. Finally, we introduce our learning method in sec. 4.3 to estimate the $P(X|S, Z)$ and $P(Y|S)$, which are respectively resorted in the inference and prediction that constitute a robust predictor during test stage.

### 4.1 Latent Causal Invariance Models

In this section, we introduce a set of structural causal models dubbed as **La**tent **C**ausal **I**nvariance **M**odel (LaCIM), which incorporates the causal assumptions mentioned above and also the source of distributional shifts. The corresponding causal structure of LaCIM is illustrated in Fig. 1 (c), which we will introduce step-by-step from the skeleton in Fig. 1 (a).

**Fig. 1 (a).** Specifically, the ADMG in Fig. 1 (a) introduces latent factors $V := \{S, Z\}$ to model the abstractions/concepts that generate the observed variables $(X, Y)$, as similarly assumed in

unsupervised latent generative models [37] for image generation. Further, we explicitly separate the $V$ into $S$ and $Z$, with only $S$ causally related to the label $Y$. In image classification, such a causal factor refers to the (shape,contour) of the object need to be classified; while the image $X$ is additionally affected by contextual factor such as *light, view*.

**Fig. 1 (a) → Fig. 1 (b).** In addition, we assume that $S$ is spuriously correlated with $Z$, as marked by the red "↔" in Fig. 1 (a). Such a spurious correlation corresponds to the bias inherited from data, *e.g.* the contextual information in image classification. Therefore, the magnitude of this correlation is distribution-dependent and thus can vary across domains. Statistically, the "spurious correlation" implicates the presence of a third unobserved (we use dot circle to represent unobserved variables) confounder, which is denoted as $C$ in Fig. 1 (b). The unblocked path from $Z$ to $Y$ induced by $C$ can lead to learning the non-causal factor during data-fitting, which can degrade the performance on unseen domains if the correlation between this non-causal factor and the output is broken.

**Fig. 1 (b) → Fig. 1 (c).** Taking a further inspection in Fig. 1 (b), the varying degree of correlation can be either due to the distributional shift of $S, Z|C$ or of the $C$ itself across domains (we use red color to mean varied distributions). As both shifts are domain-dependent, we in Fig. 1 (c) ascribe them to a domain variable $D$, which causes the mutation of its children nodes' distribution, *i.e.*, $S, Z$ and $C$. Such a domain variable has been similarly introduced in [69, 68] to generate mutable variables. In our scenario, we do not require $D$ to be observed; rather, we only need the domain index $\tilde{d}^e$ (one-hot encoded vector with length $m := |\mathcal{E}_{\text{train}}|$). The set of SCMs augmented with $D$, with the SCM Markovian compatible to the DAG of $C, S, Z, X, Y$ in Fig. 1 (c), is dubbed as **La**tent **C**ausal **I**nvariance **M**odels (LaCIM) that is formally defined as follows:

**Definition 4.1** (LaCIM). *The LaCIM denotes a set of SCMs augmented with the domain variable $D$, i.e., $\{\langle M^e, d^e \rangle\}_{e \in \mathcal{E}}$, in which $d^e$ denotes the value of $D$ and $M^e := \langle G, \mathcal{F}^e, P(\varepsilon) \rangle$ for $e$. The $G$ denotes the DAG restricted on $C, S, Z, X, Y$. For each environment/domain $e$, the $\mathcal{F}^e := \{f_x, f_y, f_s^e, f_z^e, f_c^e\}$ correspond to generating mechanism of $X, Y, S, Z, C$, with $f_c^e(\varepsilon_c) := g_c(\varepsilon_c, d^e)$, $f_s^e(c, \varepsilon_s) := g_s(c, \varepsilon_s, d^e)$ and $f_z^e(c, \varepsilon_z) := g_z(c, \varepsilon_z, d^e)$ from some $g_c, g_s, g_z$.*

**Remark 1.** *Different from scenarios in which $X$ generates [28] nor generated from $Y$ [1], we consider the scenario when the $X$ and $Y$ are generated concurrently, which can widely exist but ignored in the literature. For example, the clinicians are recording the disease status while implementing the ultrasound test at the same time, during medical diagnosis.*

As an illustration, we consider the following example, in which the distributional shifts caused by domain variable $D$ can refer to sampling bias in data.

**Example 4.1** (Sampling Bias). *Consider the cat/dog classification, in which the animal in each image is either associated with the snow or grass. The $D$ refers to the sampler, which generates the $C$ that denotes time and weather to collect each sample. The $S, Z$ respectively refer to the features of animals and context. Since each sampler may have a fixed sampling pattern (e.g. gets used to going out in the sunny morning (or in the snowy evening)), the data one collects may have sampling bias: dogs (cats) more associated with grass (snow) in the sunny morning (or snowy evening).*

The Def. 4.1 specifies the generating mechanisms across environments and how they differ. Equipped with such a specification, we can identify the invariant mechanisms that are stable to domain shifts:

**Proposition 4.2** (Causal Invariance & Generative Invariance). *For LaCIM in Def. 4.1, the $P(Y|S)$ and $P(X|S, Z)$ are invariant to shifts across $\mathcal{E}$, and are respectively denoted as **C**ausal **I**nvariance (CI) and **G**enerative **I**nvariance (GI).*

**Remark 2.** *The generating process from latent variables to observed variables follows from physical law, e.g., the shape, contour, color, view, light should satisfy physical constraints to generate a reasonable image. Therefore, it is naturally hold that such generating processes are invariant.*

The $P(X|S, Z)$ and $P(Y|S)$ can induce an invariant predicting mechanism. Specifically, for a new sample $x \leftarrow f_x(s^\star, z^\star, \varepsilon_x)$, $y \leftarrow f_y(s^\star, \varepsilon_y)$, we can first infer the causal factor $s^\star$ from $p_{f_x}(x|s, z)$ by maximizing log-likelihood of $p_{f_x}(x|s, z)$ over $\mathcal{S} \times \mathcal{Z}$ and then feed the estimated $s$ into $p_{f_y}(y|s^\star)$ for prediction. To ensure the robustness of such a two-step invariant prediction, we need to answer two following *identifiability* questions:

1. *Can the inferred causal factor $S$ not mix the information of (disentangled from) others?*
2. *Can such an invariant predictor recover the ground-truth predictor $P(Y|s^\star)$?*

We will answer these questions in the subsequent section, followed by our learning methods to identify the causal factor and the causal/generative invariance for prediction.

## 4.2 Identifiability Analysis

We present the identifiability results regarding **(i)** the disentanglement of inferred causal factor $S$ from non-causal $Z$, and **(ii)** the induced true predicting mechanism $P(Y|s^\star)$ for $x \leftarrow f_x(s^\star, z^\star, \varepsilon_x)$, which respectively echo the two questions imposed in the last section.

Our main results are presented in theorem 4.4. To distinguish the causal factor $S$ from others, our results require that the degree of diversity of $S$-$Z$ correlation across environments is large enough, which has been similarly assumed in the literature of identifiability [52, 1]. Such a *diversity* condition implies the dramatical change of correlation between $Z$ and $Y$, thus providing a clue to disentangle the $S$. Such a disentanglement analysis, is crucial to causal prediction but is ignored in existing literature about identifiability, such as those identifying the discrete latent confounders [32, 62], or those relying on *Additive Noise Model* (ANM) assumption [31], or linear Independent Component Analysis (ICA) [14, 35, 36, 75] (Please refer to supplement D.1 for more exhaustive reviews). More importantly, we will later in theorem 4.5 show the extension of above analysis from exponential family of $P(S, Z|C)$ to Sobelev space; and from ANM for $Y$ to categorical distribution for $Y$.

We assume the ANM for $f_x(s, z, \varepsilon_x) = \hat{f}_x(s, z) + \varepsilon_x$ (we replace $\hat{f}_x$ with $f_x$ for simplicity), which has been widely adopted to identify the causal factor [30, 54, 35]. We assume the $f_x$ to be bijective and invertible (we will discuss it later). We first narrow our interest to a subset of LaCIM denoted as $\mathcal{P}_{\exp}$ in which any model in $\mathcal{P}_{\exp}$ satisfies that **(i)** the $S, Z$ belong to the exponential family; and **(ii)** the $Y$ is generated from the ANM:

$$\mathcal{P}_{\exp} = \Big\{ \text{LaCIM with any } m > 0 \big| \, y = f_y(s) + \varepsilon_y, p^e(s, z|c) := \Pi_{t=s,z} p_{\mathbf{T}^t, \mathbf{\Gamma}^t_{c,d^e}}(t|c), \forall e \Big\}, \text{ with}$$

$$p_{\mathbf{T}^t, \mathbf{\Gamma}^t_{c,d^e}}(t) = \prod_{i=1}^{q_t} \exp \Big( \sum_{j=1}^{k_t} T^t_{i,j}(t_i) \Gamma^t_{c,d^e,i,j} + B_i(t_i) - A^t_{c,d^e,i} \Big), \forall k_t, q_t \quad (1)$$

for $t = s, z$ and $e \in \mathcal{E}$, with $q_t, k_t$ respectively denoting the dimension of $t = s, z$ and the number of natural parameters in each dimension. The $\{T^t_{i,j}(t_i)\}$, $\{\Gamma^t_{c,d^e,i,j}\}$ denote the sufficient statistics and natural parameters, $\{B_i\}$ and $\{A^t_{c,d^e,i}\}$ denote the base measures and normalizing constants to ensure the integral of distribution equals to 1. Let $\mathbf{T}^t(t) := [\mathbf{T}^t_1(t_1), ..., \mathbf{T}^t_{q_t}(t_{q_t})] \in \mathbb{R}^{k_t \times q_t}$ $(\mathbf{T}^t_i(t_i) := [T^t_{i,1}(t_i), ..., T^t_{i,k_t}(t_i)], \forall i \in [q_t])$, $\mathbf{\Gamma}^t_{c,d^e} := [\mathbf{\Gamma}^t_{c,d^e,1}, ..., \mathbf{\Gamma}^t_{c,d^e,q_t}] \in \mathbb{R}^{k_t \times q_t}$ $(\mathbf{\Gamma}^t_{c,d^e,i} := [\Gamma^t_{c,d^e,i,1}, ..., \Gamma^t_{c,d^e,i,k_t}], \forall i \in [q_t])$. We further assume that the $P^e(C)$ serves to discrete distributions on the set $\{c_1, ..., c_R\}$, with which the $p^e(s, z) := \int p(s|c)p(z|c)dP^e(c) = \sum_r p^e(s, z|c_r)p^e(c_r)$ can be regarded as the mixture of exponential family distributions. Rather than uniquely inference, we target on disentangling the $S$ from $Z$ and also recovering the ground-truth predictor, which is formally defined as $\sim_{\exp}$-identifiability as follows:

**Definition 4.3** ($\sim_{\exp}$-identifiability)**.** *Suppose the $\mathcal{X} \supseteq f_x(\mathcal{S} \times \mathcal{Z})$. We define a binary relation $\theta \sim_{\exp} \tilde{\theta}$ on the parameter space of $\mathcal{X} \times \mathcal{Y}$: there exist two sets of permutation matrices and vectors, $(M_s, a_s)$ and $(M_z, a_z)$ for s and z respectively, such that for any $(x, y) \in \mathcal{X} \times \mathcal{Y}$, the following hold:*

$$\tilde{\mathbf{T}}^s([\tilde{f}_x^{-1}]_\mathcal{S}(x)) = M_s \mathbf{T}^s([f_x^{-1}]_\mathcal{S}(x)) + a_s, \, \tilde{\mathbf{T}}^z([\tilde{f}_x^{-1}]_\mathcal{Z}(x)) = M_z \mathbf{T}^z([f_x^{-1}]_\mathcal{Z}(x)) + a_z; \quad (2)$$

$$p_{\tilde{f}_y}(y|[\tilde{f}_x^{-1}]_\mathcal{S}(x)) = p_{f_y}(y|[f_x^{-1}]_\mathcal{S}(x)). \quad (3)$$

*We then say that $\theta$ is $\sim_{\exp}$-identifiable, if for any $\tilde{\theta}$, $p_\theta^e(x, y) = p_{\tilde{\theta}}^e(x, y) \, \forall e \in \mathcal{E}_{\text{train}}$, implies $\theta \sim_{\exp} \tilde{\theta}$.*

This definition is inspired by but beyond the scope of unsupervised scenario considered in nonlinear ICA [27, 35] in that, the former further disentangle $S$ from $Z$ (in Eq. (2)) and identify the true predicting mechanism (in Eq. (3)). To see disentanglement, note that for any clean (noise-free) sample $x \leftarrow f_x(s^\star, z^\star)$, the Eq. (2) ensures that the inferred causal factor $\tilde{\mathbf{T}}^s([\tilde{f}_x^{-1}]_\mathcal{S}(x))$ does *not* mix the information of others, unless the extreme case that there is a deterministic function between $S$ and $Z$, in which it is impossible for $S$ to be identified. With such an identification of $s$, the Eq. (3) further guarantees that the learned $p_{\tilde{f}_y}(y|[\tilde{f}^{-1}]_\mathcal{S}(x))$ can recover the ground-truth prediction probability density, i.e., $p_{f_y}(y|[f_x^{-1}]_\mathcal{S}(x)) = p_{f_y}(y|s^\star)$. With noise, the $s^\star$ can be inferred with some indeterminacy. The formal result is presented in theorem 4.4.

**Theorem 4.4** ($\sim_{\exp}$-identifiability). *For $\theta$ of $\mathcal{P}_{\exp}$ in Def. 4.1 with $m := |\mathcal{E}_{\text{train}}|$, we have that the $\theta$ is $\sim_{\exp}$ identifiable under following assumptions:*

1. *The characteristic functions of $\varepsilon_x, \varepsilon_y$ are almost everywhere nonzero.*
2. *$f_x, f'_x, f''_x$ are continuous and $f_x, f_y$ are bijective;*
3. *The $\{T^t_{i,j}\}_{1 \leq j \leq k_t}$ are linearly independent in $\mathcal{S}$ or $\mathcal{Z}$ for each $i \in [q_t]$ for any $t = s, z$; and $T^t_{i,j}$ are twice differentiable for any $t = s, z, i \in [q_t], j \in [k_t]$;*
4. *The $\left\{ \left( \mathbf{T}^s([f^{-1}]_{\mathcal{S}}(x)), \mathbf{T}^z([f^{-1}]_{\mathcal{Z}}(x)) \right); \mathcal{B}(x) > 0 \right\}$ contains a non-empty open set in $\mathbb{R}^{q_s \times k_s + q_z \times k_z}$, with $\mathcal{B}(x) := \prod_{i_s \in [q_s]} B_{i_s}([f^{-1}]_{i_s}(x)) \prod_{i_z \in [q_z]} B_{i_z}([f^{-1}]_{i_z}(x))$.*
5. *The $L := [P^{e_1}(C)^{\mathsf{T}}, ..., P^{e_m}(C)^{\mathsf{T}}]^{\mathsf{T}} \in \mathbb{R}^{m \times R}$ and $\left[ [\mathbf{\Gamma}^{t=s,z}_{c_2,d^{e_1}} - \mathbf{\Gamma}^{t=s,z}_{c_1,d^{e_1}}]^{\mathsf{T}}, ..., [\mathbf{\Gamma}^{t=s,z}_{c_R,d^{e_m}} - \mathbf{\Gamma}^{t=s,z}_{c_1,d^{e_1}}]^{\mathsf{T}} \right]^{\mathsf{T}} \in \mathbb{R}^{(R \times m) \times (q_t \times k_t)}$ have full column rank.*

The assumptions 1-3 are trivial and easy to satisfy. The characteristics functions of $\varepsilon_x, \varepsilon_y$ can be almost everywhere non-zero for most continuous variables, such as Gaussian, exponential, beta, gamma distribution. This assumption can ensure the identifiability of $p(f^{-1}(x)$, as will be shown in the appendix. The bijectivity of $f_x$ and $f_y$ have been widely assumed in [30, 54, 53, 35, 75] as a basic condition for identifiability. It naturally holds for $f_x$ to be bijective since it has been empirically proven in auto-encoder [38] that the low-dimension embeddings (*i.e.*, $q_s + q_z < q_x$) can recover the original input well and also that the variational auto-encoder can extract meaningful representations from $x$. For the $\theta$ with categorical $Y$ such that $p(y = k|s) = [f_y]_k(s)/(\sum_k [f_y]_k(s))$, the $f_y$ may not satisfy the bijectivity condition. We will shown identifiability for such a categorical case later in theorem 4.5. The assumption 3 can be uniformly satisfied for all distributions in the strongly exponential family. The containment of an open set in assumption (4) for $\left\{ \left( \mathbf{T}^s([f^{-1}]_{\mathcal{S}}(x)), \mathbf{T}^z([f^{-1}]_{\mathcal{Z}}(x)) \right); \mathcal{B}(x) > 0 \right\}$ implies that space expanded by sufficient statistics are dense in some open set, as a sufficient condition for the mixture distribution $P^e(C)$ and also $P^e(X, Y|c)$ to be identified. The diversity assumption (5) implies that **i)** $m \geq R$ and $m * R \geq \max(k_z * q_z, k_s * q_s) + 1$; and that **ii)** different environments are diverse enough in terms of $S$-$Z$ correlation, as an almost a necessary for the invariant one to be identified (a different version is assumed in [1]). In supplement B.2, we will show that the **ii)** can hold unless the space of $\mathbf{\Gamma}$ belong to a zero-(Lebesgue) measure set. As indicated by the formulation, a larger $m$ would be easier to satisfy the condition, which agrees with the intuition that more environments can provide more complementary information. Besides, our result can be extended to non-independent case among $\{s_1, ..., s_{q_s}\}$ (or $\{z_1, ..., z_{q_z}\}$), *i.e.*, $p_{\mathbf{T}^t, \mathbf{\Gamma}^t_{c,d^e}}(t) = \exp(\langle \mathbf{T}^t(t), \mathbf{\Gamma}^t_{c,d^e} \rangle + B(t) - A^t_{c,d^e})$ $(t = s, z)$, which will shown in supplement B.2.

**Extension to the general forms of LaCIM.** We extend to general forms of LaCIM in theorem 4.5 as long as its $\mathrm{P}(S, Z|C = c) \in W^{r,2}(\mathcal{S} \times \mathcal{Z})$ (for some $r \geq 2$) and categorical $Y$, in the following theorem. This is accomplished by proving that any model in LaCIM can be approximated by a sequence of distributions with parameterization in $\mathcal{P}_{\exp}$, motivated by [3] that the exponential family is dense in the set of distributions with bounded support, and in [44] that the continuous variable with multinomial logit model can be approximated by a series of distributions with *i.i.d* Gumbel noise as the temperature converges to infinity. The proof is left in the supplement.

**Theorem 4.5** (Asymptotic $\sim_{\exp}$-identifiability). *Suppose the LaCIM satisfy that $p(x|s, z)$ and $p(y|s)$ are smooth w.r.t $s$, $z$ and $s$ respectively. For each $e$ and $c \in \mathcal{C}$, suppose $P^e(S, Z|c) \in W^{r,2}(\mathcal{S} \times \mathcal{Z})$ for some $r \geq 2$, we have that the LaCIM is asymptotically $\sim_{\exp}$-identifiable: $\forall \epsilon > 0, \exists \sim_{\exp}$-identifiable $\tilde{P}_\theta \in \mathcal{P}_{\exp}$, s.t. $d_{\mathrm{Pok}}(p^e(X, Y), \tilde{p}^e_\theta(X, Y)) < \epsilon, \forall e \in \mathcal{E}_{\text{train}}$ [3].*

Our proof is built on [3] that any probability in Sobolev space can be approximated by a sequence of distribution with the number of natural paramters going to infinity, *i.e.*, $k_t \to \infty$.

### 4.3 Learning and Inference

Guided by the identifiability result, we propose to learn $P(X|S, Z)$ and $P(Y|S)$ via generative modeling following from Fig. 1 (c). Then to predict the label for a new sample $x$ generated from $(s^\star, z^\star)$, we first leverage the learned $p(x|s, z)$ to infer $s^\star$ that is ensured to be able to not mix the non-causal information, followed by learned $P(y|\tilde{s}^\star)$ for prediction.

---

[3]The $d_{\mathrm{Pok}}(\mu_1, \mu_2)$ denotes the Pokorov distance between $\mu_1$ and $\mu_2$, with $\lim_{n \to \infty} d_{\mathrm{Pok}}(\mu_n, \mu) \to 0 \iff \mu_n \xrightarrow{d} \mu$.

### 4.3.1 Learning Method

To learn the $P(X|S,Z), P(Y|S)$ for invariant prediction, we reformulate the objective function of Variational Auto-Encoder (VAE) in the supervised scenario, in order to fit $\{p^e(x,y)\}_{e \in \mathcal{E}_{\text{train}}}$. As a latent generative model, the VAE was originally proposed for unsupervised generation from latent variables $V$ to high-dimensional input variable $X$. To make such a generation tractable, the VAE introduced a variational distribution $q_\psi$ parameterized by $\psi$ to approximate the intractable posterior by maximizing the following **E**vidence **L**ower **Bo**und (ELBO): $-\mathcal{L}_{\theta,\psi} = \mathbb{E}_{p(x)}\left[\mathbb{E}_{q_\psi(v|x)} \log \frac{p_\theta(x,v)}{q_\psi(v|x)}\right] \leq \mathbb{E}_{p(x)}[\log p_\theta(x)]$, where the equality is achieved when $q_\psi(v|x) = p_\theta(v|x)$. Therefore, maximizing the ELBO over $p_\theta$ and $q_\psi$ will drive **(i)** $q_\psi(v|x)$ to approximate $p_\theta(v|x)$; **(ii)** $p_\theta$ to estimate the ground-truth model $p$.

To adapt the above surrogate loss to our DAG in Fig. 1 (c), we introduce the variational distribution $q_\psi^e(s,z|x,y)$ for each environment $e$. The corresponding ELBO for $e$ is

$$-\mathcal{L}_{\theta,\psi}^e \triangleq \mathbb{E}_{p^e(x,y)}\left[\mathbb{E}_{q_\psi^e(s,z|x,y)} \log \frac{p_\theta^e(x,y,s,z)}{q_\psi^e(s,z|x,y)}\right], \text{ where } p_\theta^e(x,y,s,z) = p_\theta(x|s,z)p_\theta(y|s)p^e(s,z).$$

Similarly, minimizing $\mathcal{L}_{\theta,\psi}^e$ can drive $p_\theta(x|s,z), p_\theta(y|s)$ to approximate the $p(x|s,z), p(y|s)$, and also $q_\psi^e(s,z|x,y)$ to estimate $p_\theta^e(s,z|x,y)$. Therefore, the $q_\psi$ can inherit the properties of $p_\theta$. As $p_\theta^e(s,z|x,y) = \frac{p_\theta^e(s,z|x)p_\theta(y|s)}{p_\theta^e(y|x)}$ for our DAG in Fig. 1 (c), we can similarly reparameterize $q_\psi^e(s,z|x,y)$ as $\frac{q_\psi^e(s,z|x)p_\theta(y|s)}{q_\psi^e(y|x)}$ with $q_\psi(y|s)$ replaced by $p_\theta(y|s)$ (since the goal of $q_\psi$ is to mimic the behavior of $p_\theta$). Then, the $\mathcal{L}_{\theta,\psi}^e$ can be rewritten as:

$$\mathcal{L}_{\theta,\psi}^e = \mathbb{E}_{p^e(x,y)}\left[-\log q_\psi^e(y|x) - \mathbb{E}_{q_\psi^e(s,z|x)}\frac{p_\theta(y|s)}{q_\psi^e(y|x)} \log \frac{p_\theta(x|s,z)p_\theta^e(s,z)}{q_\psi^e(s,z|x)}\right], \qquad (4)$$

where $q_\psi^e(y|x) = \int_{\mathcal{S}} q_\psi^e(s|x)p_\theta(y|s)ds$. We correspondingly parameterize the prior model $p_\theta^e(s,z)$ and inference model $q_\psi^e(s,z|x)$ as $p_\theta(s,z|\tilde{d}^e)$ and $q_\psi(s,z|x,\tilde{d}^e)$, in which $\tilde{d}^e$ (of environment $e$) denotes the domain *index* that can be represented by the one-hot encoded vector with length $m := |\mathcal{E}_{\text{train}}|$. The overall loss function is:

$$\mathcal{L}_{\theta,\psi} \triangleq \sum_{e \in \mathcal{E}_{\text{train}}} \mathcal{L}_{\theta,\psi}^e. \qquad (5)$$

The training datasets $\{\mathcal{D}^e\}_{e \in \mathcal{E}_{\text{train}}}$ are applied to optimize the prior models $\{p(s,z|\tilde{d}^e)\}_e$, inference models $\{q_\psi(s,z|x,\tilde{d}^e)\}_e$, generative model $p_\theta(x|s,z)$ and predictive model $p_\theta(y|s)$. Particularly, the parameters of $p_\theta(x|s,z)$ and $p_\theta(y|s)$ are shared among all environments, motivated by the the invariance property of $P(X|S,Z)$ and $P(Y|S)$ across all domains.

### 4.3.2 Inference & Prediction.

We leverage the learned $P(X|S,Z), P(Y|S)$ for prediction. According to Prop. 4.2 and Eq. (3) in theorem 4.4, the induced predictor via $P(X|S,Z), P(Y|S)$ can recover the true predicting mechanism for any distributional shifts from $\mathcal{E}$. Specifically, for any $x$ generated by $(s^\star, z^\star)$, we first optimize the following log-likelihood of $p_\theta(x|s,z)$ over $\mathcal{S} \times \mathcal{Z}$ to infer $s^\star, z^\star$,

$$\max_{s,z} \log p_\theta(x|s,z) + \lambda_s\|s\|_2^2 + \lambda_z\|z\|_2^2, \qquad (6)$$

with hyperparameters $\lambda_s > 0$ and $\lambda_z > 0$ in order to control the learned $s, z$ in a reasonable scale. Note that Eq. Eq. (6) is different from the maximum a posterior estimation since the posterior $q_\psi^e(s,z|x)$ is parameterized differently for different $e$ while the $p_\theta(x|s,z)$ is invariantly parameterized for $\mathcal{E}$ (this is because $p(x|s,z)$ is invariant). For optimization, we adopt the strategy in [61] that first sample some candidate points from $\mathcal{N}(0,I)$ and select the optimal one in terms of Eq. (6) as initial point; then use Adam to optimize for another $T$ iterations. The implementation details and optimization effect are shown in supplement E.2. Finally, with estimated $\tilde{s}^\star, \tilde{z}^\star$, we implement the learned $p_\theta(y|\tilde{s}^\star)$ for prediction: $\tilde{y} := \arg\max_y p_\theta(y|\tilde{s}^\star)$.

# 5 Experiments

We first verify the identifiability claims of theorem 4.4 in sec. 5.1. Then we evaluate LaCIM on real-world data in sec. 5.2: Non-I.I.D. Image dataset with Contexts (NICO); Colored MNIST (CMNIST); Alzheimer's Disease Neuroimaging Initiative (ADNI www.loni.ucla.edu/ADNI for early prediction of Alzheimer's Disease), to verify the generalization ability of our method on the target domain with distributional shifts.

## 5.1 Simulation

To verify the identifiability claims, we implement LaCIM on synthetic data. We generate $C, S, Z, X, Y$ following Fig. 1 (with details left in supplementary). We choose $m = 3, 5$ with the same total number of samples. To verify the advantage of learning on multiple diverse domains ($m > 1$), we compare with pool-LaCIM: minimizing the loss Eq. (4) on the pooled data from all $m$ domains. We compute the mean correlation coefficient (MCC) adopted in [35], which measures the goodness of identifiability under permutation by introducing cost optimization to assign each learned component to the source component. We run all methods for 100 times, with the average recorded in Fig. 2a. The superiority of LaCIM over pool-LaCIM, together with the fact that LaICM with $m = 5$ performs better than $m = 3$, verify the benefit of more domains to satisfy the diversity condition. To illustrate the learning effect, we visualize the learned $Z$ (with $S$ left in supplement E.1) in Fig. 2b.

| Method | MCC for $Z$ | MCC for $S$ |
|---|---|---|
| pool-LaCIM | 0.41 | 0.71 |
| LaCIM (**Ours**, $m = 3$) | 0.63 | **0.84** |
| LaCIM (**Ours**, $m = 5$) | **0.71** | **0.84** |

(a) Mean Correlation Coefficient for $S$ and $Z$.

(i) pool-LaCIM     (ii) LaCIM     (iii) $p_{\theta^\star}(z|D)$

(b) Estimated posteriors ((i),(ii)) and the ground-truth ((iii)).

Figure 2: Identification of Latent Variables. As shown qualitatively (b) and quantitatively (a), the LaCIM can identify the latent variables (up to permutation and point-wise transformation).

## 5.2 Real-world Data

We verify the generalization ability of LaCIM on three data: NICO, CMNIST and ADNI.

**Dataset.** We describe the datasets as follows ($X, Y$ denotes the input and output; $D$ is unobserved):

- *NICO*. We consider the cat/dog classification in "Animal" dataset in NICO, a benchmark for non-i.i.d problem in [20]. Each animal is associated with "grass","snow" contexts. The $D$ denotes the attributes of the sampler. The $C$ denotes the time and weather of sampling, which generates the $S, Z$ that respectively denote the semantic and contextual features. We split the dataset into $m$ training domains and the test domain, in which each domain has different proportions of contexts associated with each animal, *i.e.*, (%cat in grass, %cat in snow, %dog in grass, %dog in snow), due to different sampling strategies determined by $D$. The proportion vectors of all domains are introduced in Tab. 3. The distributional shift refers to the spurious correlation between the context and the label.

- *CMNIST*: We relabel the digits 0-4 and 5-9 as $y = 0$ and $y = 1$, based on MNIST. Then we color $p^e$ ($1 - p^e$) of images with $y = 0$ ($y = 1$) as green and others as red. We set $m = 2$ with $p^{e_1} = 0.95, p^{e_2} = 0.99$; while the $p^{e_{\text{test}}}$ for the test domain is set to 0.1. The $D$ denotes the attributes of the painter. The $Z, S$ respectively represent the features related to the color and the digit. Their confounder $C$ denotes the time and weather for which the painter $D$ draws the number and color, *e.g.*, the painter tends to draw red 0 more often than green 1 in the sunny morning. In this regard, the distributional shift refers to the spurious correlation between the color and the label.

- *ADNI*. The $\mathcal{Y} := \{0, 1, 2\}$, with 0,1,2 respectively denoting Normal Control, Mild Cognitive Impairment and AD. The $X$ is structural Magnetic resonance imaging. We split the data into $m = 2$ training domains and the test domain, with different values of $D$ that denotes Age, TAU (a biomarker [24]). The $C, S$ ($Z$) respectively denote the hormone level that

affects the brain structure development and the disease-related (-unrelated) brain regions. The distributional shifts among all domains are due to different values of $D$.

**Compared Baselines & Implementation Details.** We compare with (i) Empirical Risk Mnimization from $X \to Y$ (ERM), (ii) domain-adversarial neural network (DANN) [15], (iii) Maximum Mean Discrepancy with Adversarial Auto-Encoder (MMD-AAE) [43], (iv) Domain Invariant Variational Autoencoders (DIVA) [29], (v) Invariant Risk Mnimization (IRM) [1], (vi) Supervised VAE (sVAE): our LaCIM implemented by VAE without disentangling $S, Z$. For all methods, the network structures of $q_\psi^e(s, z|x)$, $p_\theta(x|s, z)$ and $p_\theta(y|s)$ for CMNIST, NICO and ADNI are shared (details introduced in supplement E.4, E.5, E.6, Tab. 7, 8). We implement SGD as optimizer, with learning rate (lr) 0.5 and weight decay (wd) $1e$-5 for CMNIST; lr 0.01 with decaying $0.2\times$ every 60 epochs, wd $5e$-5 for NICO and ADNI (wd is $2e$-4). The batch-size are set to 256, 30 and 4 for CMNIST, NICO, ADNI.

Table 1: Accuracy (%) on test domain. Average over 10 runs.

| Dataset | NICO | | | | CMNIST | | ADNI ($m = 2$) | | |
|---|---|---|---|---|---|---|---|---|---|
| | $m = 8$ | | $m = 14$ | | $m = 2$ | | $D$: Age | $D$: TAU | |
| Method | ACC | # Params | ACC | # Params | ACC | # Params | ACC | ACC | # Params |
| ERM | $60.3 \pm 2.8$ | 18.08M | $59.3 \pm 2.1$ | 18.08M | $91.9 \pm 0.9$ | 1.12M | $62.1 \pm 3.2$ | $64.3 \pm 1.0$ | 28.27M |
| DANN | $58.9 \pm 1.7$ | 19.13M | $60.1 \pm 2.6$ | 26.49M | $84.8 \pm 0.7$ | 1.1M | $61.0 \pm 1.5$ | $65.2 \pm 1.1$ | 30.21M |
| MMD-AAE | $60.8 \pm 3.4$ | 19.70M | $64.8 \pm 7.7$ | 19.70M | $92.5 \pm 0.8$ | 1.23M | $60.3 \pm 2.2$ | $65.2 \pm 1.5$ | 36.68M |
| DIVA | $58.8 \pm 3.4$ | 14.86M | $58.1 \pm 1.4$ | 14.87M | $86.1 \pm 1.0$ | 1.69M | $61.8 \pm 1.8$ | $64.8 \pm 0.8$ | 33.22M |
| IRM | $61.4 \pm 3.8$ | 18.08M | $62.8 \pm 4.6$ | 18.08M | $92.9 \pm 1.2$ | 1.12M | $62.2 \pm 2.6$ | $65.2 \pm 1.1$ | 28.27M |
| sVAE | $60.4 \pm 2.1$ | 18.25M | $64.3 \pm 1.2$ | 19.70M | $93.6 \pm 0.9$ | 0.92M | $62.7 \pm 2.5$ | $66.6 \pm 0.8$ | 37.78M |
| LaCIM (**Ours**) | $\mathbf{63.2 \pm 1.7}$ | 18.25M | $\mathbf{66.4 \pm 2.2}$ | 19.70M | $\mathbf{96.6 \pm 0.3}$ | 0.92M | $\mathbf{63.8 \pm 1.1}$ | $\mathbf{67.3 \pm 0.9}$ | 37.78M |

**Main Results & Discussions.** We report accuracy over 10 runs for each method. As shown in Tab. 1, our LaCIM consistently outperforms others on all data. Specifically, the advantage over IRM and ERM may due to the incorporation of causal assumptions embedded in Fig. 1 (c). Further, the improvement over sVAE is benefited from the separation of $S$ from others to avoid spurious correlation. Besides, a larger $m$ (with the total sample size fixed) can bring further benefit on NICO, which may due to the easier satisfaction of the diversity condition in theorem 4.4.

**Interpretability.** We visualize the learned $S$ and $Z$ on CMNIST and NICO. Specifically, for CMNIST, we visualize the generated image (with only digit "0" among all classes that belong to $Y = 0$ and digit "7" among all classes that belong to $Y = 1$) by interpolating $S$ (and $Z$) with fixed $Z$ (and $S$); for NICO, we adopt the gradient method [67], which visualizes the derivatives of the $S^\star$ (*i.e.*, dimension of $S$ that has the highest correlation with $Y$) with respect to each image.

As shown in Fig. 3a, the generated sequential images in the 1st and 2nd row look more like "7" from "0" as $s$ increases; while the sequential images in the 2nd-row change from red to green as $z$ increases. Besides, different dimensions of $S$ can learn different differentiating semantic information. For example, the first dimension can learn to add the dash in the hand-writing "7"; while the second dimension can learn to remove the left part of "0" to "7" as interpolated. For the dimension of $Z$, it learned other non-differentiating factors such as width, color. This result reflects that the learned $S$ and $Z$ correspond to the digit (causal factor of $Y$) and color-related features. For NICO, Fig. 3b shows the ability of identifying more explainable semantic features of LaCIM than ERM in which the learned features can mix the background information. Supplement E.5 provides more results.

## 6 Conclusions & Discussions

We propose recovering *latent causal factor* that is robust to distributional shifts caused by a domain variable. We introduce the causal and non-causal latent factors that are spuriously correlated with each other, and generate the input and the output via invariant mechanisms. Under this invariance, the causal factor is guaranteed to be disentangled from the non-causal one, which induces the ground-truth predictor that holds on all domains. A reformulated generative model is proposed for inferring the causal factor and prediction. A possible drawback of our model lies in our requirement of the number of environments for identifiability, the relaxation of which is left in future work.

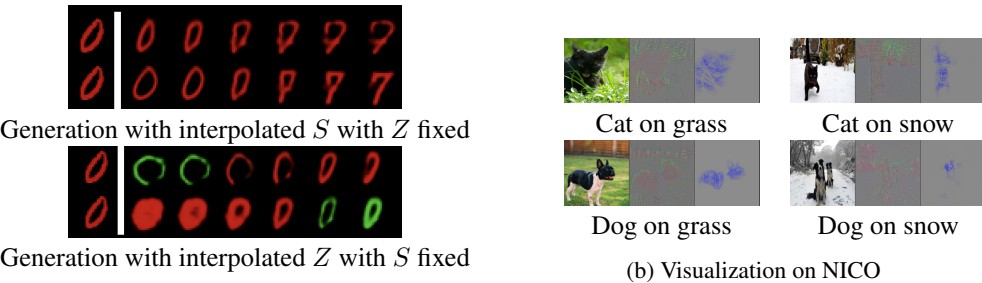

Generation with interpolated $S$ with $Z$ fixed

Generation with interpolated $Z$ with $S$ fixed

(a) Visualization on CMNIST

Cat on grass      Cat on snow

Dog on grass      Dog on snow

(b) Visualization on NICO

Figure 3: (a) Visualization on CMNIST via interpolation. (b) Visualization on NICO via gradient [67]. From the left to right: original image, ERM and LaCIM.

## Broader Impact

We claim that this work does not present any foreseeable negative social impact.

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
