## Supplementary Materials

## A  O.O.D Generalization error Bound

Denote $\mathbb{E}_p[y|x] := \int_{\mathcal{Y}} yp(y|x)dy$ for any $x, y \in \mathcal{X} \times \mathcal{Y}$. We have $\mathbb{E}_{p^e}[y|s] = \int_{\mathcal{Y}} yp(y|s)dy$ according to that $p(y|s)$ is invariant across $\mathcal{E}$, we can omit $p^e$ in $\mathbb{E}_{p^e}[y|s]$ and denote $g(S) := \mathbb{E}[Y|S]$. Then, the OOD bound $\left|\mathbb{E}_{p^{e_1}}(y|x) - \mathbb{E}_{p^{e_2}}(y|x)\right|$, $\forall(x, y)$ is bounded as follows:

**Theorem A.1** (OOD genearlization error). *Consider two causal models in LaCIM $P^{e_1}$ and $P^{e_2}$, suppose that their densities, i.e., $p^{e_1}(s|x)$ and $p^{e_2}(s|x)$ are absolutely continuous having support $(-\infty, \infty)$. For any $(x, y) \in \mathcal{X} \times \mathcal{Y}$, assume that*

- *$g(S)$ is a Lipschitz-continuous function;*
- *$\pi_x(s) := \frac{p^{e_2}(s|x)}{p^{e_1}(s|x)}$ is differentiable and $\mathbb{E}_{p^{e_1}}\left[\pi_x(S)\big|g(S) - \mu_1\big|\right] < \infty$ with $\mu_1 := \mathbb{E}_{p^{e_1}}[g(S)|X = x] = \int_{\mathcal{S}} g(s)p^{e_1}(s|x)ds;$*

*then we have $\left|\mathbb{E}_{p^{e_1}}(y|x) - \mathbb{E}_{p^{e_2}}(y|x)\right| \leq \|g'\|_\infty\|\pi_x'\|_\infty \mathrm{Var}_{p^{e_1}}(S|X = x)$.*

When $e_1 \in \mathcal{E}_{\text{train}}$ and $e_2 \in \mathcal{E}_{\text{test}}$, the theorem A.1 describes the error during generalization on $e_2$ for the strategy that trained on $e_1$. The bound is mainly affected by: (i) the Lipschitz constant of $g$, i.e., $\|g\|_\infty$; (ii) $\|\pi_x'\|_\infty$ which measures the difference between $p^{e_1}(s, z)$ and $p^{e_2}(s, z)$; and (iii) the $\mathrm{Var}_{p^{e_1}}(S|x)$ that measures the intensity of $x \to (s, z)$. These terms can be roughly categorized into two classes: (i),(iii) which are related to the invariance property of $P(X|S, Z)$ and $P(Y|S)$ and gave few space for improvement; and the (ii) that describes the distributional change between two environments. Specifically for the first class, the (i) measures the smoothness of $\mathbb{E}(y|s)$ with respect to $s$. The smaller value of $\|g'\|_\infty$ implies that the flatter regions give rise to the same prediction result, hence easier transfer from $e_1$ to $e_2$ and vice versa. For the term (iii), consider the deterministic setting that $\varepsilon_x = 0$ (leads to $\mathrm{Var}_{p^{e_1}}(S|x) = 0$), then $s$ can be determined from $x$ for generalization if the $f$ is bijective function.

The term (ii) measures the distributional change between posterior distributions $p^{e_1}(s|x)$ and $p^{e_2}(s|x)$, which contributes to the difference during prediction: $\left|\mathbb{E}_{p^{e_1}}(y|x) - \mathbb{E}_{p^{e_2}}(y|x)\right| = \int_{\mathcal{S}} (p^{e_1}(s|x) - p^{e_1}(s|x))p_{f_y}(y|s)ds$. Such a change is due to the inconsistency between priors $p^{e_1}(s, z)$ and $p^{e_2}(s, z)$, which is caused by different value of the confounder $d$.

*Proof.* In the following, we will derive the upper bound

$$\left|\mathbb{E}_{p^{e_1}}[Y|X=x] - \mathbb{E}_{p^{e_2}}[Y|X=x]\right| \leq \|g'\|_\infty\|\pi_x'\|_\infty \mathrm{Var}_{p^{e_1}}(S|X = x),$$

where $\pi_x(s) =: \frac{p^{e_2}(s|x)}{p^{e_1}(s|x)}$ and $g(s)$ is assumed to be Lipschitz-continuous.

To begin with, note that

$$\mathbb{E}[Y|X] = \mathbb{E}[\mathbb{E}(Y|X, S)|X] = \mathbb{E}[g(S)|X] = \int g(s)p(s|x)ds.$$

Let $p_1(s|x) = p^{e_1}(s|x)$, $p_2(s|x) = p^{e_2}(s|x)$. For ease of notations, we use $P_1$ and $P_2$ denote the distributions with densities $p_1(s|x)$ and $p_2(s|x)$ and suppose $S_1 \sim P_1$ and $S_2 \sim P_2$, where $x$ is omitted as the following analysis is conditional on a fixed $X = x$.

Then we may rewrite the difference of conditional expectations as

$$\mathbb{E}_{p^{e_2}}[Y|X = x] - \mathbb{E}_{p^{e_1}}[Y|X = x] = \mathbb{E}(g(S_2)) - \mathbb{E}(g(S_1)),$$

where $\mathbb{E}[g(S_j))] = \int g(s)p_j(s|x)ds$ denotes the expectation over $P_j$.

Let $\mu_1 := \mathbb{E}_{p^{e_1}}[g(S)|X = x] = \mathbb{E}[g(S_1)] = \int g(s)p_1(s|x)ds$. Then

$$\mathbb{E}_{p^{e_2}}[Y|X = x] - \mathbb{E}_{p^{e_1}}[Y|X = x] = \mathbb{E}(g(S_2)) - \mathbb{E}(g(S_1)) = \mathbb{E}[g(S_2) - \mu_1].$$

Further, we have the following transformation

$$\mathbb{E}[g(S_2) - \mu_1] = \int (g(s) - \mu_1)\pi_x(s)p_1(s|x)ds = \mathbb{E}[(g(S_1) - \mu_1)\pi_x(S_1)]. \qquad (7)$$

In the following, we will use the results of the Stein kernel function. Please refer to Definition A.2 for a general definition. Particularly, for the distribution $P_1 \sim p_1(s|x)$, the Stein kernel $\tau_1(s)$ is

$$\tau_1(s) = \frac{1}{p_1(s|x)} \int_{-\infty}^{s} (\mathbb{E}(S_1) - t)p_1(t|x)dt, \tag{8}$$

where $\mathbb{E}(S_1) = \int s \cdot p_1(s|x)ds$. Further, we define $(\tau_1 \circ g)(s)$ as

$$(\tau_1 \circ g)(s) = \frac{1}{p_1(s|x)} \int_{-\infty}^{s} (\mathbb{E}(g(S_1)) - g(t))p_1(t|x)dt = \frac{1}{p_1(s|x)} \int_{-\infty}^{s} (\mu_1 - g(t))p_1(t|x)dt. \tag{9}$$

Under the second condition listed in Theorem A.1, we may apply the result of Lemma A.3. Specifically, by the equation (12), we have

$$\mathbb{E}\left[(g(S_1) - \mu_1)\pi_x(S_1)\right] = \mathbb{E}\left[(\tau_1 \circ g)(S_1)\pi_x'(S_1)\right].$$

Then under the first condition in Theorem A.1, we can obtain the following inequality by Lemma A.4,

$$\mathbb{E}\left[(\tau_1 \circ g)(S_1)\pi_x'(S_1)\right] = \mathbb{E}\left[\left(\frac{(\tau_1 \circ g)}{\tau_1}\pi_x'\tau_1\right)(S_1)\right] \leq \mathbb{E}\left[\left|\frac{(\tau_1 \circ g)}{\tau_1}(S_1)\right| \cdot \left|\pi_x'\tau_1(S_1)\right|\right]$$
$$\leq \|g'\|_\infty \mathbb{E}\left[|(\pi_x'\tau_1)(S_1)|\right] \leq \|g'\|_\infty \|\pi_x'\|_\infty \mathbb{E}\left[|\tau_1(S_1)|\right]. \tag{10}$$

In the following, we show that the Stein kernel is non-negative, which enables $\mathbb{E}\left[|\tau_1(S_1)|\right] = \mathbb{E}\left[\tau_1(S_1)\right]$. According to the definition, $\tau_1(s) = \frac{1}{p_1(s|x)} \int_{-\infty}^{s} (\mathbb{E}(S_1) - t)p_1(t|x)dt$, where $\mathbb{E}(S_1) = \int_{-\infty}^{\infty} t \cdot p_1(t|x)dt$. Let $F_1(s) = \int_{-\infty}^{s} p_1(t|x)dt$ be the distribution function for $P_1$. Note that

$$\int_{-\infty}^{s} \mathbb{E}(S_1)p_1(t|x)dt = F_1(s)\mathbb{E}(S_1) = F_1(s)\mathbb{E}(S_1),$$

$$\int_{-\infty}^{s} tp_1(t|x)dt = F_1(s)\int_{-\infty}^{s} t\frac{p_1(t|x)}{F_1(s)}dt = F_1(s)\mathbb{E}(S_1|S_1 \leq s) \leq F_1(s)\mathbb{E}(S_1),$$

The last inequality is based on $\mathbb{E}(S_1|S_1 \leq s) - \mathbb{E}(S_1) \leq 0$ that can be proved as the following

$$\int_{-\infty}^{s} t\frac{p_1(t|x)}{F_1(s)}dt - \int_{-\infty}^{\infty} tp_1(t|x)dt = \int_{-\infty}^{s} t\left(\frac{1}{F_1(s)} - 1\right)p_1(t|x)dt - \int_{s}^{\infty} tp_1(t|x)dt$$
$$\leq s\int_{-\infty}^{s}\left(\frac{1}{F_1(s)} - 1\right)p_1(t|x)dt - s\int_{s}^{\infty} p_1(t|x) = 0.$$

Therefore, $\tau_1(s) \geq 0$ and hence $\mathbb{E}\left[|\tau_1(S_1)|\right] = \mathbb{E}\left[\tau_1(S_1)\right]$ in (10).

Besides, by equation (13), the special case of Lemma A.3, we have

$$\mathbb{E}\left[\tau_1(S_1)\right] = \text{Var}(S_1) = \text{Var}_{p^{e_1}}(S|X = x).$$

To sum up,

$$\mathbb{E}\left[(\tau_1 \circ g)(S_1)\pi_x'(S_1)\right] \leq \|g'\|_\infty \|\pi_x\|_\infty \mathbb{E}\left[\tau_1(S_1)\right] = \|g'\|_\infty \|\pi_x'\|_\infty \text{Var}_{p^{e_1}}(S|X = x).$$

$\qquad\qquad\qquad\qquad\qquad\qquad\qquad\qquad\qquad\qquad\qquad\qquad\qquad\qquad\qquad\qquad\qquad\qquad\square$

**Definition A.2 (the Stein Kernel $\tau_P$ of distribution $P$).** Suppose $X \sim P$ with density $p$. The Stein kernel of $P$ is the function $x \mapsto \tau_P(x)$ defined by

$$\tau_P(x) = \frac{1}{p(x)} \int_{-\infty}^{x} (\mathbb{E}(X) - y)p(y)dy, \tag{11}$$

where Id is the identity function for $\text{Id}(x) = x$. More generally, for a function $h$ satisfying $\mathbb{E}[|h(X)|] < \infty$, define $(\tau_P \circ h)(x)$ as

$$(\tau_P \circ h)(x) = \frac{1}{p(x)} \int_{-\infty}^{x} (\mathbb{E}(h(X)) - h(y))p(y)dy.$$

**Lemma A.3.** *For a differentiable function $\varphi$ such that $\mathbb{E}[|(\tau_P \circ h)(x)\varphi'(X)|] < \infty$, we have*

$$\mathbb{E}\left[(\tau_P \circ h)(x)\varphi'(X)\right] = \mathbb{E}[(h(X) - \mathbb{E}(h(X))\varphi(X)]. \tag{12}$$

*Proof.* Let $\mu_h =: \mathbb{E}(h(X))$. As $\mathbb{E}(h(X) - \mu_h) = 0$,

$$(\tau_P \circ h)(x) = \frac{1}{p(x)} \int_{-\infty}^{x} (\mu_h - h(y))p(y)dy = \frac{-1}{p(x)} \int_{x}^{\infty} (\mu_h - h(y))p(y)dy.$$

Then

$$\mathbb{E}\left[(\tau_P \circ h)(x)\varphi'(X)\right] = \int_{-\infty}^{0} (\tau_P \circ h)(x)\varphi'(x)p(x)dx + \int_{0}^{\infty} (\tau_P \circ h)(x)\varphi'(x)p(x)dx$$

$$= \int_{-\infty}^{0} \int_{-\infty}^{x} (\mu_h - h(y))p(y)\varphi'(x)dydx - \int_{0}^{\infty} \int_{x}^{\infty} (\mu_h - h(y))p(y)\varphi'(x)dydx$$

$$= \int_{-\infty}^{0} \int_{y}^{0} (\mu_h - h(y))p(y)\varphi'(x)dxdy - \int_{0}^{\infty} \int_{0}^{y} (\mu_h - h(y))p(y)\varphi'(x)dxdy$$

$$= \int_{-\infty}^{0} \int_{0}^{y} (h(y) - \mu_h)p(y)\varphi'(x)dxdy + \int_{0}^{\infty} \int_{0}^{y} (h(y) - \mu_h)p(y)\varphi'(x)dxdy$$

$$= \int_{-\infty}^{\infty} (h(y) - \mu_h)p(y) \left( \int_{0}^{y} \varphi'(x)dx \right) dy = \int_{-\infty}^{\infty} (h(y) - \mu_h)p(y)(\varphi(y) - \varphi(0))dy$$

$$= \int_{-\infty}^{\infty} (h(y) - \mu_h)p(y)(\varphi(y))dy = \mathbb{E}[(h(X) - \mathbb{E}(h(X))\varphi(X)]$$

Particularly, taking $h(X) = X$ and $\varphi(X) = X - \mathbb{E}(X)$, we immediately have

$$\mathbb{E}(\tau_P(X)) = \text{Var}(X) \tag{13}$$

$\square$

**Lemma A.4.** *Assume that $\mathbb{E}(|X|) < \infty$ and the density $p$ is locally absolutely continuous on $(-\infty, \infty)$ and $h$ is a Lipschitz continuous function. Then we have $|f_h| \leq \|h'\|_\infty$ for*

$$f_h(x) = \frac{(\tau_P \circ h)(x)}{\tau_P(x)} = \frac{\int_{-\infty}^{x}(\mathbb{E}(h(X)) - h(y))p(y)dy}{\int_{-\infty}^{x}(\mathbb{E}(X) - y)p(y)dy}.$$

*Proof.* This is a special case of Corollary 3.15 in [13], taking the constant $c = 1$. $\square$

## B   Proof of Identifiability

### B.1   Proof of the Equivalence of Definition 4.3

**Proposition B.1.** *The binary relation $\sim_{\exp}$ defined in Def. 4.3 is an equivalence relation.*

*Proof.* The equivalence relation should satisfy three properties as follows:

- *Reflexive* property: The $\theta \sim_{\exp} \theta$ with $M_z$, $M_s$ being identity matrix and $a_s$, $a_z$ being 0.

- *Symmtric* property: If $\theta \sim_{\exp} \tilde{\theta}$, then there exists block permutation matrices $M_z$ and $M_s$ such that

  $$\mathbf{T}^s([f_x]_{\mathcal{S}}^{-1}(x)) = M_s\tilde{\mathbf{T}}^s([\tilde{f}_x]_{\mathcal{S}}^{-1}(x)) + a_s, \ \mathbf{T}^z([f_x]_{\mathcal{Z}}^{-1}(x)) = M_z\tilde{\mathbf{T}}^z([\tilde{f}_x]_{\mathcal{Z}}^{-1}(x)) + a_z,$$
  $$p_{f_y}(y|[f_x]_{\mathcal{S}}^{-1}(x)) = p_{\tilde{f}_y}(y|[\tilde{f}_x]_{\mathcal{S}}^{-1}(x)).$$

The we have $M_s^{-1}$ and $M_z^{-1}$ are also block permutation matrices and such that:

$$\tilde{\mathbf{T}}^s([\tilde{f}_x]_{\mathcal{S}}^{-1}(x)) = M_s^{-1}\mathbf{T}^s([f_x]_{\mathcal{S}}^{-1}(x)) + (-a_s), \ \tilde{\mathbf{T}}^s([\tilde{f}_x]_{\mathcal{Z}}^{-1}(x)) = M_z^{-1}\mathbf{T}^s([f_x]_{\mathcal{Z}}^{-1}(x)) + (-a_z),$$
$$p_{\tilde{f}_y}(y|[\tilde{f}_x]_{\mathcal{S}}^{-1}(x)) = p_{f_y}(y|[f_x]_{\mathcal{S}}^{-1}(x)).$$

Therefore, we have $\tilde{\theta} \sim_{\exp} \theta$.

- *Transitive* property: if $\theta_1 \sim_{\exp} \theta_2$ and $\theta_2 \sim_{\exp} \theta_3$ with $\theta_i :=$ $\{f_x^i, f_y^i, \mathbf{T}^{s,1}, \mathbf{T}^{z,1}, \mathbf{\Gamma}^{s,i}, \mathbf{\Gamma}^{z,i}\}$, then we have

$$\mathbf{T}^{s,1}((f_{x,s}^1)^{-1}(x)) = M_s^1\mathbf{T}^{s,2}((f_{x,s}^2)^{-1}(x)) + a_s^1,$$
$$\mathbf{T}^{z,1}((f_{x,z}^1)^{-1}(x)) = M_z^1\mathbf{T}^{z,2}((f_{x,z}^2)^{-1}(x)) + a_z^2,$$
$$\mathbf{T}^{s,2}((f_{x,s}^2)^{-1}(x)) = M_s^2\mathbf{T}^{s,3}((f_{x,s}^3)^{-1}(x)) + a_s^2,$$
$$\mathbf{T}^{z,2}((f_{x,z}^2)^{-1}(x)) = M_z^2\mathbf{T}^{z,3}((f_z^3)^{-1}(x)) + a_{x,z}^3$$

for block permutation matrices $M_s^1, M_z^1, M_s^2, M_z^2$ and vectors $a_s^1, a_s^2, a_z^1, a_z^2$. Then we have $\mathrm{T}^{s,1}((f_{x,s}^1)^{-1}(x)) = M_s^2 M_s^1 \mathbf{T}^{s,3}((f_{x,s}^3)^{-1}(x)) + (M_s^2 a_s^1) + a_s^2,$ $\mathbf{T}^{z,1}((f_{x,z}^1)^{-1}(x)) = M_z^2 M_z^1 \mathbf{T}^{z,3}((f_{x,z}^3)^{-1}(x)) + (M_z^2 a_z^1) + a_z^2.$

Besides, it is apparent that $p_{f_y^1}(y|(f_x^1)_s^{-1}(x)) = p_{f_y^2}(y|(f_x^2)_s^{-1}(x)) = p_{f_y^3}(y|(f_x^3)_s^{-1}(x)).$

Therefore, we have $\theta_1 \sim_{\exp} \theta_3$ since $M_s^2 M_s^1$ and $M_z^2 M_z^1$ are also permutation matrices.

With above three properties satisfied, we have that $\sim_{\exp}$ is a equivalence relation. $\qquad\square$

## B.2   Proof of Theorem 4.4

In the following, we write $p^e(x, y)$ as $p(x, y|d^e)$ and also $\Gamma_{c,d^e}^{t=s,z} := \Gamma^{t=s,z}(c, d^e), A_{c,d^e,i}^t = A_i^t(c, d^e)$ for $t = s, z$. To prove the theorem 4.4, we first prove the theorem B.6 for the simplest case when $c|d^e := d^e$, then we generalize to the case when $\mathcal{C} := \cup_r\{c_r\}$. The overall roadmap is as follows: we first prove the $\sim_A$-identifiability in theorem B.5, and the combination of which with lemma B.9, B.8 give theorem B.6 in the simplest case when $c|d^e = d^e$. Then we generalize the case considered in theorem B.6 to the more general case when $\mathcal{C} := \cup_r\{c_r\}$.

**Theorem B.2** ($\sim_{\exp}$-identifiability). *For $\theta$ in the LaCIM $p_\theta^e(x, y) \in \mathcal{P}_{\exp}$ for any $e \in \mathcal{E}_{\text{train}}$, we assume that (1) the $f_x, f_x'$ and $f_x''$ are continuous and that $f_x, f_y$ are bijective; (2) that the $\{T_{i,j}^t\}_{j\in[k_t]}$ are linearly indepndent and $T_{i,j}^t$ are twice differentiable for any $t = s, z, i \in [q_t], j \in [k_t]$; (3) the exogenous variables satisfy that the characteristic functions of $\varepsilon_x, \varepsilon_y$ are almost everywhere nonzero; (4) the number of environments, i.e., $m \geq \max(q_s * k_s, q_z * k_z) + 1$ and $\left[\mathbf{\Gamma}_{d^{e_2}}^{t=s,z} - \mathbf{\Gamma}_{d^{e_1}}^{t=s,z}, ..., \mathbf{\Gamma}_{d^{e_m}}^{t=s,z} - \mathbf{\Gamma}_{d^{e_1}}^{t=s,z}\right]$ have full column rank for both $t = s$ and $t = z$, we have that the parameters $\theta := \{f_x, f_y, \mathbf{T}^s, \mathbf{T}^z\}$ are $\sim_{\exp}$ identifiable.*

To prove theorem B.6, We first prove the $\sim_A$-identifiability that is defined as follows:

**Definition B.3** ($\sim_A$-identifiability). *The definition is the same with the one defined in 4.3, with $M_s, M_z$ being invertible matrices which are not necessarily to be the permutation matrices in Def. 4.3.*

**Proposition B.4.** *The binary relation $\sim_A$ defined in Def. B.3 is an equivalence relation.*

*Proof.* The proof is similar to that of proposition B.1. $\qquad\square$

The following theorem states that any LaCIM that belongs to $\mathcal{P}_{\exp}$ is $\sim_A$-identifiable.

**Theorem B.5** ($\sim_A$-identifiability). *For $\theta$ in the LaCIM $p_\theta^e(x,y) \in \mathcal{P}_{\exp}$ for any $e \in \mathcal{E}_{\text{train}}$, we assume* **(1)** *the $f_x, f_y$ are bijective;* **(2)** *that the $\{T_{i,j}^t\}_{j \in [k_t]}$ are linearly indepndent and $T_{i,j}^t$ are differentiable for any $t = s, z, i \in [q_t], j \in [k_t]$;* **(3)** *the exogenous variables satisfy that the characteristic functions of $\varepsilon_x, \varepsilon_y$ are almost everywhere nonzero;* **(4)** *the number of environments, i.e., $m \geq \max(q_s * k_s, q_z * k_z) + 1$ and $\left[[\mathbf{\Gamma}_{d^{e_2}}^t - \mathbf{\Gamma}_{d^{e_1}}^t]^{\mathsf{T}}, ..., [\mathbf{\Gamma}_{d^{e_m}}^t - \mathbf{\Gamma}_{d^{e_1}}^t]^{\mathsf{T}}\right]^{\mathsf{T}}$ have full column rank for $t = s, z$, we have that the parameters $\{f_x, f_y, \mathbf{T}^s, \mathbf{T}^z\}$ are $\sim_{\exp}$ identifiable.*

*Proof.* Suppose that $\theta = \{f_x, f_y, \mathbf{T}^s, \mathbf{T}^z\}$ and $\tilde{\theta} = \{\tilde{f}_x, \tilde{g}_y, \tilde{\mathbf{T}}^s, \tilde{\mathbf{T}}^z\}$ share the same observational distribution for each environment $e \in \mathcal{E}_{\text{train}}$, *i.e.*, $p_{f_x, f_y, \mathbf{T}^s, \mathbf{\Gamma}^s, \mathbf{T}^z, \mathbf{\Gamma}^z}(x, y|d^e) = p_{\tilde{f}_x, \tilde{f}_y, \tilde{\mathbf{T}}^s, \tilde{\mathbf{\Gamma}}^s, \tilde{\mathbf{T}}^z, \tilde{\mathbf{\Gamma}}^z}(x, y|d^e)$.

Then we have $p_{f_x, f_y, \mathbf{T}^s, \mathbf{\Gamma}^s, \mathbf{T}^z, \mathbf{\Gamma}^z}(x|d^e) = p_{\tilde{f}_x, \tilde{f}_y, \tilde{\mathbf{T}}^s, \tilde{\mathbf{\Gamma}}^s, \tilde{\mathbf{T}}^z, \tilde{\mathbf{\Gamma}}^z}(x|d^e)$

$\implies \int_{\mathcal{S} \times \mathcal{Z}} p_{f_x}(x|s,z) p_{\mathbf{T}^s, \mathbf{\Gamma}^s, \mathbf{T}^z, \mathbf{\Gamma}^z}(s, z|d^e) ds dz = \int_{\mathcal{S} \times \mathcal{Z}} p_{\tilde{f}_x}(x|s,z) p_{\tilde{\mathbf{T}}^s, \tilde{\mathbf{\Gamma}}^s, \tilde{\mathbf{T}}^z, \tilde{\mathbf{\Gamma}}^z}(s, z|d^e) ds dz$

$\implies \int_{\mathcal{X}} p_{\varepsilon_x}(x - \bar{x}) p_{\mathbf{T}^s, \mathbf{\Gamma}^s, \mathbf{T}^z, \mathbf{\Gamma}^z}(f_x^{-1}(\bar{x})|d^e) \mathrm{vol} J_{f_x^{-1}}(\bar{x}) d\bar{x}$

$= \int_{\mathcal{X}} p_{\varepsilon_x}(x - \bar{x}) p_{\tilde{\mathbf{T}}^s, \tilde{\mathbf{\Gamma}}^s, \tilde{\mathbf{T}}^z, \tilde{\mathbf{\Gamma}}^z}(\tilde{f}_x^{-1}(\bar{x})|d^e) \mathrm{vol} J_{\tilde{f}_x^{-1}}(\bar{x}) d\bar{x}$

$\implies \int_{\mathcal{X}} \tilde{p}_{\mathbf{T}^s, \mathbf{\Gamma}^s, \mathbf{T}^z, \mathbf{\Gamma}^z, f_x}(\bar{x}|d^e) p_{\varepsilon_x}(x - \bar{x}) d\bar{x} = \int_{\mathcal{X}} \tilde{p}_{\tilde{\mathbf{T}}^s, \tilde{\mathbf{\Gamma}}^s, \tilde{\mathbf{T}}^z, \tilde{\mathbf{\Gamma}}^z, \tilde{f}_x}(\bar{x}|d^e) p_{\varepsilon_x}(x - \bar{x}) d\bar{x}$

$\implies (\tilde{p}_{\mathbf{T}^s, \mathbf{\Gamma}^s, \mathbf{T}^z, \mathbf{\Gamma}^z, f_x} * p_{\varepsilon_x})(x|d^e) = (\tilde{p}_{\tilde{\mathbf{T}}^s, \tilde{\mathbf{\Gamma}}^s, \tilde{\mathbf{T}}^z, \tilde{\mathbf{\Gamma}}^z, \tilde{f}_x}) * p_{\varepsilon_x}(x|d^e)$

$\implies F[\tilde{p}_{\mathbf{T}^s, \mathbf{\Gamma}^s, \mathbf{T}^z, \mathbf{\Gamma}^z, f_x}](\omega) \varphi_{\varepsilon_x}(\omega) = F[\tilde{p}_{\tilde{\mathbf{T}}^s, \tilde{\mathbf{\Gamma}}^s, \tilde{\mathbf{T}}^z, \tilde{\mathbf{\Gamma}}^z, \tilde{f}_x}](\omega) \varphi_{\varepsilon_x}(\omega)$

$\implies F[\tilde{p}_{\mathbf{T}^s, \mathbf{\Gamma}^s, \mathbf{T}^z, \mathbf{\Gamma}^z, f_x}](\omega) = F[\tilde{p}_{\tilde{\mathbf{T}}^s, \tilde{\mathbf{\Gamma}}^s, \tilde{\mathbf{T}}^z, \tilde{\mathbf{\Gamma}}^z, \tilde{f}_x}](\omega)$

$\implies \tilde{p}_{\mathbf{T}^s, \mathbf{\Gamma}^s, \mathbf{T}^z, \mathbf{\Gamma}^z, f_x}(x|d^e) = \tilde{p}_{\tilde{\mathbf{T}}^s, \tilde{\mathbf{\Gamma}}^s, \tilde{\mathbf{T}}^z, \tilde{\mathbf{\Gamma}}^z, \tilde{f}_x}(x|d^e)$

where $\mathrm{vol} J_f(X) := \det(J_f(X))$ for any square matrix $X$ and function $f$ with "$J$" standing for the Jacobian. The $\tilde{p}_{\mathbf{T}^s, \mathbf{\Gamma}^s, \mathbf{T}^z, \mathbf{\Gamma}^z, f_x}(x)$ in Eq. (B.2) is denoted as $p_{\mathbf{T}^s, \mathbf{\Gamma}^s, \mathbf{T}^z, \mathbf{\Gamma}^z}(f_x^{-1}(x|d^e) \mathrm{vol} J_{f^{-1}}(x)$. The '*' in Eq. (B.2) denotes the convolution operator. The $F[\cdot]$ in Eq. (B.2) denotes the Fourier transform, where $\phi_{\varepsilon_x}(\omega) = F[p_{\varepsilon_x}](\omega)$. Since we assume that the $\varphi_{\varepsilon_x}(\omega)$ is non-zero almost everywhere, we can drop it to get Eq. (B.2). Similarly, we have that:

$$p_{f_y, \mathbf{T}^s, \mathbf{\Gamma}^s}(y|d^e) = p_{\tilde{f}_y, \tilde{\mathbf{T}}^s, \tilde{\mathbf{\Gamma}}^s}(y|d^e) \tag{14}$$

$$\implies \int_{\mathcal{S}} p_{f_y}(y|s) p_{\mathbf{T}^s, \mathbf{\Gamma}^s}(s|d^e) ds = \int_{\mathcal{S}} p_{\tilde{f}_y}(y|s) p_{\tilde{\mathbf{T}}^s, \tilde{\mathbf{\Gamma}}^s}(s|d^e) ds \tag{15}$$

$$\implies \int_{\mathcal{Y}} p_{\varepsilon_y}(y - \bar{y}) p_{\mathbf{T}^s, \mathbf{\Gamma}^s}(f_y^{-1}(\bar{y})|d^e) \mathrm{vol} J_{f_y^{-1}}(\bar{y}) d\bar{y} \tag{16}$$

$$= \int_{\mathcal{Y}} p_{\varepsilon_y}(y - \bar{y}) p_{\tilde{\mathbf{T}}^s, \tilde{\mathbf{\Gamma}}^s}(\tilde{f}_y^{-1}(\bar{y})|d^e) \mathrm{vol} J_{\tilde{g}^{-1}}(\bar{y}) d\bar{y} \tag{17}$$

$$\implies \int_{\mathcal{S}} \tilde{p}_{\mathbf{T}^s, \mathbf{\Gamma}^s, f_y}(\bar{y}|d^e) p_{\varepsilon_y}(y - \bar{y}) d\bar{y} = \int_{\mathcal{S}} \tilde{p}_{\tilde{\mathbf{T}}^s, \tilde{\mathbf{\Gamma}}^s, \tilde{f}_y}(\bar{y}|d^e) p_{\varepsilon_y}(y - \bar{y}) d\bar{y} \tag{18}$$

$$\implies (\tilde{p}_{\mathbf{T}^s, \mathbf{\Gamma}^s, f_y} * p_{\varepsilon_y})(y|d^e) = (\tilde{p}_{\tilde{\mathbf{T}}^s, \tilde{\mathbf{\Gamma}}^s, \tilde{f}_y} * p_{\varepsilon_y})(y|d^e) \tag{19}$$

$$\implies F[\tilde{p}_{\mathbf{T}^s, \mathbf{\Gamma}^s, f_y}](\omega) \varphi_{\varepsilon_y}(\omega) = F[\tilde{p}_{\tilde{\mathbf{T}}^s, \tilde{\mathbf{\Gamma}}^s, \tilde{f}_y}](\omega) \varphi_{\varepsilon_y}(\omega) \tag{20}$$

$$\implies F[\tilde{p}_{\mathbf{T}^s, \mathbf{\Gamma}^s, f_y}](\omega) = F[\tilde{p}_{\tilde{\mathbf{T}}^s, \tilde{\mathbf{\Gamma}}^s, \tilde{f}_y}](\omega) \tag{21}$$

$$\implies \tilde{p}_{\mathbf{T}^s, \mathbf{\Gamma}^s, f_y}(y) = \tilde{p}_{\tilde{\mathbf{T}}^s, \tilde{\mathbf{\Gamma}}^s, \tilde{f}_y}(y), \tag{22}$$

and that

$$p_{f_x, f_y \mathbf{T}^s, \mathbf{\Gamma}^s, \mathbf{T}^z, \mathbf{\Gamma}^z}(x, y|d^e) = p_{\tilde{f}_x, \tilde{f}_y, \tilde{\mathbf{T}}^s, \tilde{\mathbf{\Gamma}}^s, \tilde{\mathbf{T}}^z, \tilde{\mathbf{\Gamma}}^z}(x, y|d^e) \tag{23}$$

$$\implies \int_{\mathcal{S} \times \mathcal{Z}} p_{f_x}(x|s,z) p_{f_y}(y|s) p_{\mathbf{T}^s, \mathbf{\Gamma}^s, \mathbf{T}^z, \mathbf{\Gamma}^z}(s, z|d^e) ds dz$$

$$= \int_{\mathcal{S} \times \mathcal{Z}} p_{\tilde{f}}(x|s,z) p_{\tilde{f}_y}(y|s) p_{\tilde{\mathbf{T}}^s, \tilde{\mathbf{\Gamma}}^s, \tilde{\mathbf{T}}^z, \tilde{\mathbf{\Gamma}}^z}(s, z|d^e) ds dz \tag{24}$$

$$\implies \int_{\mathcal{V}} p_\varepsilon(v - \bar{v}) p_{\mathbf{T}^s, \mathbf{\Gamma}^s, \mathbf{T}^z, \mathbf{\Gamma}^z}(h^{-1}(\bar{v})|d^e) \mathrm{vol} J_{h^{-1}}(\bar{v}) d\bar{v} \tag{25}$$

$$= \int_{\mathcal{V}} p_\varepsilon(v - \bar{v}) p_{\tilde{\mathbf{T}}^s, \tilde{\mathbf{\Gamma}}^s, \tilde{\mathbf{T}}^z, \tilde{\mathbf{\Gamma}}^z}(\tilde{h}^{-1}(\bar{v})|d^e) \mathrm{vol} J_{\tilde{h}^{-1}}(\bar{v}) d\bar{v} \tag{26}$$

$$\implies \int_{\mathcal{S} \times \mathcal{Z}} \tilde{p}_{\mathbf{T}^s, \mathbf{\Gamma}^s, \mathbf{T}^z, \mathbf{\Gamma}^z, h, c}(\bar{v}|d) p_\varepsilon(v - \bar{v}) d\bar{v} = \int_{\mathcal{S} \times \mathcal{Z}} \tilde{p}_{\tilde{\mathbf{T}}^s, \tilde{\mathbf{\Gamma}}^s, \tilde{\mathbf{T}}^z, \tilde{\mathbf{\Gamma}}^z, \tilde{h}, d^e}(\bar{v}|d^e) p_\varepsilon(v - \bar{v}) d\bar{v} \tag{27}$$

$$\implies (\tilde{p}_{\mathbf{T}^s, \mathbf{\Gamma}^s, \mathbf{T}^z, \mathbf{\Gamma}^z, h} * p_\varepsilon)(v) = (\tilde{p}_{\tilde{\mathbf{T}}^s, \tilde{\mathbf{\Gamma}}^s, \tilde{\mathbf{T}}^z, \tilde{\mathbf{\Gamma}}^z, \tilde{h}} * p_\varepsilon)(v) \tag{28}$$

$$\implies F[\tilde{p}_{\mathbf{T}^s, \mathbf{\Gamma}^s, \mathbf{T}^z, \mathbf{\Gamma}^z, h}](\omega) \varphi_\varepsilon(\omega) = F[\tilde{p}_{\tilde{\mathbf{T}}^s, \tilde{\mathbf{\Gamma}}^s, \tilde{\mathbf{T}}^z, \tilde{\mathbf{\Gamma}}^z, \tilde{h}}](\omega) \varphi_\varepsilon(\omega) \tag{29}$$

$$\implies F[\tilde{p}_{\mathbf{T}^s, \mathbf{\Gamma}^s, \mathbf{T}^z, \mathbf{\Gamma}^z, h}](\omega) = F[\tilde{p}_{\tilde{\mathbf{T}}^s, \tilde{\mathbf{\Gamma}}^s, \tilde{\mathbf{T}}^z, \tilde{\mathbf{\Gamma}}^z, \tilde{h}}](\omega) \tag{30}$$

$$\implies \tilde{p}_{\mathbf{T}^s, \mathbf{\Gamma}^s, \mathbf{T}^z, \mathbf{\Gamma}^z, h}(v) = \tilde{p}_{\tilde{\mathbf{T}}^s, \tilde{\mathbf{\Gamma}}^s, \tilde{\mathbf{T}}^z, \tilde{\mathbf{\Gamma}}^z, h}(v), \tag{31}$$

where $v := [x^\top, y^\top]^\top$, $\varepsilon := [\varepsilon_x^\top, \varepsilon_y^\top]^\top$, $h(v) = [[f_x]_{\mathcal{Z}}^{-1}(x)^\top, f_y^{-1}(y)^\top]^\top$. According to Eq. (22), we have

$$\log \mathrm{vol} J_{f_y}(y) + \sum_{i=1}^{q_s} \left( \log B_i(f_{y,i}^{-1}(y)) - \log A_i(d^e) + \sum_{j=1}^{k_s} T_{i,j}^s(f_{y,i}^{-1}(y)) \Gamma_{i,j}^s(d^e) \right)$$

$$= \log \mathrm{vol} J_{\tilde{f}_y}(y) + \sum_{i=1}^{q_s} \left( \log \tilde{B}_i(\tilde{f}_{y,i}^{-1}(y)) - \log \tilde{A}_i(d^e) + \sum_{j=1}^{k_s} \tilde{T}_{i,j}^s(\tilde{f}_{y,i}^{-1}(y)) \tilde{\Gamma}_{i,j}^s(d^e) \right) \tag{32}$$

Suppose that the assumption (4) holds, then we have

$$\langle \mathbf{T}^s(f_y^{-1}(y)), \overline{\mathbf{\Gamma}}^s(d^{e_k}) \rangle + \sum_i \log \frac{A_i(d^{e_1})}{A_i(d^{e_k})} = \langle \tilde{\mathbf{T}}^s(\tilde{f}_y^{-1}(y)), \overline{\tilde{\mathbf{\Gamma}}}^s(d^{e_k}) \rangle + \sum_i \log \frac{\tilde{A}_i(d^{e_1})}{\tilde{A}_i(d^{e_k})} \tag{33}$$

for all $k \in [m]$, where $\overline{\mathbf{\Gamma}}(d) = \mathbf{\Gamma}(d) - \mathbf{\Gamma}(d^{e_1})$. Denote $\tilde{b}_s(k) = \sum_i \frac{\tilde{A}_i^s(d^{e_1}) A_i^s(d^{e_k})}{\tilde{A}_i^s(d^{e_k}) A_i^s(d^{e_1})}$ for $k \in [m]$, then we have

$$\overline{\mathbf{\Gamma}}^{s,\top} \mathbf{T}^s(f_y^{-1}(y)) = \overline{\tilde{\mathbf{\Gamma}}}^{s,\top} \tilde{\mathbf{T}}^s(\tilde{f}_y^{-1}(y)) + \tilde{b}_s, \tag{34}$$

Similarly, from Eq. (B.2) and Eq. (31), there exists $\tilde{b}_z, \tilde{b}_s$ such that

$$\overline{\mathbf{\Gamma}}^{s,\top} \mathbf{T}^s([f_x]_{\mathcal{S}}^{-1}(x)) + \overline{\mathbf{\Gamma}}^{z,\top} \mathbf{T}^z([f_x]_{\mathcal{Z}}^{-1}(x)) = \overline{\tilde{\mathbf{\Gamma}}}^{s,\top} \tilde{\mathbf{T}}^s([\tilde{f}_x]_{\mathcal{S}}^{-1}(x)) + \overline{\tilde{\mathbf{\Gamma}}}^{z,\top} \tilde{\mathbf{T}}^z([\tilde{f}_x]_{\mathcal{Z}}^{-1}(x)) + \tilde{b}_z + \tilde{b}_s, \tag{35}$$

where $\tilde{b}_z(k) = \sum_i \frac{\tilde{A}_i^z(d^{e_1}) A_i^z(d^{e_k})}{\tilde{A}_i^z(d^{e_k}) A_i^z(d^{e_1})}$ for $k \in [m]$; and that,

$$\overline{\mathbf{\Gamma}}^{s,\top} \mathbf{T}^s(f_y^{-1}(y)) + \overline{\mathbf{\Gamma}}^{z,\top} \mathbf{T}^z([f_x^{-1}]_{\mathcal{Z}}(x)) = \overline{\tilde{\mathbf{\Gamma}}}^{s,\top} \tilde{\mathbf{T}}^s(\tilde{f}_y^{-1}(y)) + \overline{\tilde{\mathbf{\Gamma}}}^{z,\top} \tilde{\mathbf{T}}^z([\tilde{f}_x^{-1}]_{\mathcal{Z}}(x)) + \tilde{b}_z + \tilde{b}_s. \tag{36}$$

Substituting Eq. (34) to Eq. (35) and Eq. (36), we have that

$$\overline{\mathbf{\Gamma}}^{z,\top} \mathbf{T}^z([f_x^{-1}]_{\mathcal{Z}}(y)) = \overline{\tilde{\mathbf{\Gamma}}}^{z,\top} \tilde{\mathbf{T}}^z([\tilde{f}_x^{-1}]_{\mathcal{Z}}(y)) + \tilde{b}_z, \ \overline{\mathbf{\Gamma}}^{s,\top} \mathbf{T}^s([f_x^{-1}]_{\mathcal{S}}(y)) = \overline{\tilde{\mathbf{\Gamma}}}^{s,\top} \tilde{\mathbf{T}}^s([\tilde{f}_x^{-1}]_{\mathcal{S}}(y)) + \tilde{b}_s. \tag{37}$$

According to assumption (4), the $\overline{\mathbf{\Gamma}}^{s,\top}$ and $\overline{\mathbf{\Gamma}}^{z,\top}$ have full column rank. Therefore, we have that

$$\mathbf{T}^z([f_x^{-1}]_{\mathcal{Z}}(x)) = \left( \overline{\mathbf{\Gamma}}^z \overline{\mathbf{\Gamma}}^{z,\top} \right)^{-1} \overline{\tilde{\mathbf{\Gamma}}}^{z,\top} \tilde{\mathbf{T}}^z([\tilde{f}_x^{-1}]_{\mathcal{Z}}(x)) + \left( \overline{\mathbf{\Gamma}}^z \overline{\mathbf{\Gamma}}^{z,\top} \right)^{-1} \tilde{b}_z \tag{38}$$

$$\mathbf{T}^s([f_x^{-1}]_{\mathcal{S}}(x)) = \left( \overline{\mathbf{\Gamma}}^s \overline{\mathbf{\Gamma}}^{s,\top} \right)^{-1} \overline{\tilde{\mathbf{\Gamma}}}^{s,\top} \tilde{\mathbf{T}}^s([\tilde{f}_x^{-1}]_{\mathcal{S}}(x)) + \left( \overline{\mathbf{\Gamma}}^s \overline{\mathbf{\Gamma}}^{s,\top} \right)^{-1} \tilde{b}_s. \tag{39}$$

$$\mathbf{T}^s(f_y^{-1}(y)) = \left( \overline{\mathbf{\Gamma}}^s \overline{\mathbf{\Gamma}}^{s,\top} \right)^{-1} \overline{\tilde{\mathbf{\Gamma}}}^{s,\top} \tilde{\mathbf{T}}^s(\tilde{f}_y^{-1}(y)) + \left( \overline{\mathbf{\Gamma}}^s \overline{\mathbf{\Gamma}}^{s,\top} \right)^{-1} \tilde{b}_s. \tag{40}$$

Denote $M_z := \left(\overline{\boldsymbol{\Gamma}}^z \overline{\boldsymbol{\Gamma}}^{z,\top}\right)^{-1} \overline{\tilde{\boldsymbol{\Gamma}}}^{z,\top}$, $M_s := \left(\overline{\boldsymbol{\Gamma}}^s \overline{\boldsymbol{\Gamma}}^{s,\top}\right)^{-1} \overline{\tilde{\boldsymbol{\Gamma}}}^{s,\top}$ and $a_s = \left(\overline{\boldsymbol{\Gamma}}^s \overline{\boldsymbol{\Gamma}}^{s,\top}\right)^{-1} \tilde{b}_s$, $a_z = \left(\overline{\boldsymbol{\Gamma}}^z \overline{\boldsymbol{\Gamma}}^{z,\top}\right)^{-1} \tilde{b}_z$. The left is to prove that $M_z$ and $M_s$ are invertible matrices. Denote $\bar{x} = f^{-1}(x)$.

Applying the [35, Lemma 3] we have that there exists $k_s$ points $\bar{x}^1, ..., \bar{x}^{k_s}, \tilde{\bar{x}}^1, ..., \tilde{\bar{x}}^{k_z}$ such that $\left((\mathbf{T}^s)'_i(\bar{x}^1), ..., (\mathbf{T}^s)'_i(\bar{x}^{k_s})\right)$ for each $i \in [q_s]$ and $\left((\mathbf{T}^z)'_i(\tilde{\bar{x}}^1), ..., (\mathbf{T}^z)'_i(\tilde{\bar{x}}^{k_z})\right)$ for each $i \in [q_t]$ are linearly independent. By differentiating Eq. (38) and Eq. (39) for each $\bar{x}^i$ with $i \in [q_s]$ and $\tilde{\bar{x}}^i$ with $i \in [q_z]$ respectively, we have that

$$\left(J_{\mathbf{T}^s}(\bar{x}^1), ..., J_{\mathbf{T}^s}(\bar{x}^{k_s})\right) = M_s \left(J_{\mathbf{T}^s \circ \tilde{f}_x^{-1} \circ f_x}(\bar{x}^1), ..., J_{\mathbf{T}^s \circ \tilde{f}_x^{-1} \circ f}(\bar{x}^{k_s})\right) \tag{41}$$

$$\left(J_{\mathbf{T}^z}(\tilde{\bar{x}}^1), ..., J_{\mathbf{T}^z}(\tilde{\bar{x}}^{k_z})\right) = M_z \left(J_{\mathbf{T}^z \circ \tilde{f}_x^{-1} \circ f_x}(\tilde{\bar{x}}^1), ..., J_{\mathbf{T}^z \circ \tilde{f}_x^{-1} \circ f_x}(\tilde{\bar{x}}^{k_z})\right). \tag{42}$$

The linearly independence of $\left((\mathbf{T}^s)'_i(\bar{x}^1), ..., (\mathbf{T}^s)'_i(\bar{x}^{k_s})\right)$, $\left((\mathbf{T}^z)'_i(\tilde{\bar{x}}^1), ..., (\mathbf{T}^z)'_i(\tilde{\bar{x}}^{k_z})\right)$ imply that the $\left(J_{\mathbf{T}^s}(\bar{x}^1), ..., J_{\mathbf{T}^s}(\bar{x}^{k_s})\right)$ and $\left(J_{\mathbf{T}^z}(\tilde{\bar{x}}^1), ..., J_{\mathbf{T}^z}(\tilde{\bar{x}}^{k_z})\right)$ are invertible, which implies the invertibility of matrix $M_s$ and $M_z$. The rest is to prove $p_{f_y}(y|[f_x]_{\mathcal{S}}^{-1}(x)) = p_{\tilde{f}_y}(y|[\tilde{f}_x]_{\mathcal{S}}^{-1}(x))$. This can be shown by applying Eq. (24) again. Specifically, according to Eq. (24), we have that

$$\int_{\mathcal{X}} p_{\varepsilon_x}(x - \bar{x}) p(y|[f_x]_{\mathcal{S}}^{-1}(\bar{x})) p_{\mathbf{T}^s, \boldsymbol{\Gamma}^s, \mathbf{T}^z, \boldsymbol{\Gamma}^z}(f^{-1}(\bar{x})|d^e) \mathrm{vol} J_{f^{-1}}(\bar{x}) d\bar{x}$$

$$= \int_{\mathcal{X}} p_{\varepsilon_x}(x - \bar{x}) p(y|[\tilde{f}_x]_{\mathcal{S}}^{-1}(\bar{x})) p_{\mathbf{T}^s, \boldsymbol{\Gamma}^s, \mathbf{T}^z, \boldsymbol{\Gamma}^z}(\tilde{f}^{-1}(\bar{x})|d^e) \mathrm{vol} J_{\tilde{f}^{-1}}(\bar{x}) d\bar{x}. \tag{43}$$

Denote $l_{\mathbf{T}^s, \boldsymbol{\Gamma}^s, \mathbf{T}^z, \boldsymbol{\Gamma}^z, f_y, f_x, y}(x) := p_{f_y}(y|[f_x]_{\mathcal{S}}^{-1}(\bar{x})) p_{\mathbf{T}^s, \boldsymbol{\Gamma}^s, \mathbf{T}^z, \boldsymbol{\Gamma}^z}(f^{-1}(\bar{x})|d^e) \mathrm{vol} J_{f_x^{-1}}(\bar{x})$, we have

$$\int_{\mathcal{X}} p_{\varepsilon_x}(x - \bar{x}) l_{\mathbf{T}^s, \boldsymbol{\Gamma}^s, \mathbf{T}^z, \boldsymbol{\Gamma}^z, f_y, f_x, y}(\bar{x}) d\bar{x} = \int_{\mathcal{X}} p_{\varepsilon_x}(x - \bar{x}) l_{\tilde{\mathbf{T}}^s, \tilde{\boldsymbol{\Gamma}}^s, \tilde{\mathbf{T}}^z, \tilde{\boldsymbol{\Gamma}}^z, \tilde{f}_y, \tilde{f}_x, y}(\bar{x}) d\bar{x} \tag{44}$$

$$\Longrightarrow (l_{\mathbf{T}^s, \boldsymbol{\Gamma}^s, \mathbf{T}^z, \boldsymbol{\Gamma}^z, f_y, f_x, y} * p_{\varepsilon_x})(x|d^e) = (l_{\tilde{\mathbf{T}}^s, \tilde{\boldsymbol{\Gamma}}^s, \tilde{\mathbf{T}}^z, \tilde{\boldsymbol{\Gamma}}^z, \tilde{f}_y, \tilde{f}_x, y} * p_{\varepsilon_x})(x|d^e) \tag{45}$$

$$\Longrightarrow F[l_{\tilde{\mathbf{T}}^s, \tilde{\boldsymbol{\Gamma}}^s, \tilde{\mathbf{T}}^z, \tilde{\boldsymbol{\Gamma}}^z, \tilde{f}_y, \tilde{f}_x, y}](\omega) \varphi_{\varepsilon_x}(\omega) = F[l_{\mathbf{T}^s, \boldsymbol{\Gamma}^s, \mathbf{T}^z, \boldsymbol{\Gamma}^z, f_y, f_x, y}](\omega) \varphi_{\varepsilon_x}(\omega) \tag{46}$$

$$\Longrightarrow F[l_{\mathbf{T}^s, \boldsymbol{\Gamma}^s, \mathbf{T}^z, \boldsymbol{\Gamma}^z, f_y, f_x, y}](\omega) = F[l_{\tilde{\mathbf{T}}^s, \tilde{\boldsymbol{\Gamma}}^s, \tilde{\mathbf{T}}^z, \tilde{\boldsymbol{\Gamma}}^z, \tilde{f}_y, \tilde{f}_x, y}](\omega) \tag{47}$$

$$\Longrightarrow l_{\mathbf{T}^s, \boldsymbol{\Gamma}^s, \mathbf{T}^z, \boldsymbol{\Gamma}^z, f_y, f_x, y}(x) = l_{\tilde{\mathbf{T}}^s, \tilde{\boldsymbol{\Gamma}}^s, \tilde{\mathbf{T}}^z, \tilde{\boldsymbol{\Gamma}}^z, \tilde{f}_y, \tilde{f}_x, y}(x) \tag{48}$$

$$\Longrightarrow p_{f_y}(y|[f_x]_{\mathcal{S}}^{-1}(x)) p_{\mathbf{T}^s, \boldsymbol{\Gamma}^s, \mathbf{T}^z, \boldsymbol{\Gamma}^z}(f^{-1}(x)|d^e) \mathrm{vol} J_{f_x^{-1}}(x)$$

$$= p_{\tilde{f}_y}(y|[\tilde{f}_x]_{\mathcal{S}}^{-1}(x)) p_{\tilde{\mathbf{T}}^s, \tilde{\boldsymbol{\Gamma}}^s, \tilde{\mathbf{T}}^z, \tilde{\boldsymbol{\Gamma}}^z}(\tilde{f}^{-1}(x)|d^e) \mathrm{vol} J_{\tilde{f}_x^{-1}}(x). \tag{49}$$

Taking the $\log$ transformation on both sides of Eq. (49), we have that

$$\log p_{f_y}(y|[f_x]_{\mathcal{S}}^{-1}(x)) + \log p_{\mathbf{T}^s, \boldsymbol{\Gamma}^s, \mathbf{T}^z, \boldsymbol{\Gamma}^z}(f^{-1}(x)|d^e) + \log \mathrm{vol} J_{f_x^{-1}}(x)$$

$$= \log p_{\tilde{f}_y}(y|[\tilde{f}_x]_{\mathcal{S}}^{-1}(x)) + \log p_{\tilde{\mathbf{T}}^s, \tilde{\boldsymbol{\Gamma}}^s, \tilde{\mathbf{T}}^z, \tilde{\boldsymbol{\Gamma}}^z}(\tilde{f}^{-1}(x)|d^e) + \log \mathrm{vol} J_{\tilde{f}_x^{-1}}(x). \tag{50}$$

Subtracting Eq. (50) with $y_2$ from Eq. (50) with $y_1$, we have

$$\frac{p_{f_y}(y_2|[f_x]_{\mathcal{S}}^{-1}(x))}{p_{f_y}(y_1|[f_x]_{\mathcal{S}}^{-1}(x))} = \frac{p_{\tilde{f}_y}(y_2|[\tilde{f}_x]_{\mathcal{S}}^{-1}(x))}{p_{\tilde{f}_y}(y_1|[\tilde{f}_x]_{\mathcal{S}}^{-1}(x))} \tag{51}$$

$$\Longrightarrow \int_{\mathcal{Y}} \frac{p_{f_y}(y_2|[f_x]_{\mathcal{S}}^{-1}(x))}{p_{f_y}(y_1|[f_x]_{\mathcal{S}}^{-1}(x))} dy_2 = \int_{\mathcal{Y}} \frac{p_{\tilde{f}_y}(y_2|[\tilde{f}_x]_{\mathcal{S}}^{-1}(x))}{p_{\tilde{f}_y}(y_1|[\tilde{f}_x]_{\mathcal{S}}^{-1}(x))} dy_2 \tag{52}$$

$$\Longrightarrow p_{f_y}(y_1|[f_x]_{\mathcal{S}}^{-1}(x)) = p_{\tilde{f}_y}(y_1|[\tilde{f}_x]_{\mathcal{S}}^{-1}(x)), \tag{53}$$

for any $y_1 \in \mathcal{Y}$. This completes the proof. $\qquad\square$

The following theorem describes the identifiability result for non-independence case for $S, Z$.

**Theorem B.6** ($\sim_{\exp}$-identifiability). *For $\theta$ in the LaCIM $p_\theta^e(x, y) \in \mathcal{P}_{\exp}$ for any $e \in \mathcal{E}_{\mathrm{train}}$ with the general exponential form of $p_{\mathbf{T}^t, \boldsymbol{\Gamma}^t_{c, d^e}}(t)$, i.e., $p_{\mathbf{T}^t, \boldsymbol{\Gamma}^t_{c, d^e}}(t) = \exp(\langle \mathbf{T}^t(t), \boldsymbol{\Gamma}^t_{c, d^e} \rangle + B(t) - A^t_{c, d^e})$ $(t = s, z)$, under the same assumptions with those in Thm. 4.4, we have that the parameters $\theta := \{f_x, f_y, \mathbf{T}^s, \mathbf{T}^z\}$ are $\sim_{\exp}$ identifiable.*

*Proof.* The proof is the same to above with general form of $\mathbf{T}^t(t)$, $\boldsymbol{\Gamma}^t_{c, d^e}$, $B(t)$ and $A^t_{c, d^e}$. $\qquad\square$

**Understanding the assumption (4) in Theorem B.5 and B.6.** Recall that we assume the $D$ in LaCIM is the source variable for generating data in corresponding domain. Here we also use the $\mathcal{D}$ to denote the space of $D$, then we have the following theoretical conclusion that the as long as the image set of $\mathcal{D}$ is not included in any sets with Lebesgue measure 0, the assumption (4) holds. This conclusion means that the assumption **(4)** holds generically. For more general conclusion of assumption (4) with $\left[[\mathbf{\Gamma}^t_{d^{e_2}} - \mathbf{\Gamma}^t_{d^{e_1}}]^\top, ..., [\mathbf{\Gamma}^t_{d^{e_m}} - \mathbf{\Gamma}^t_{d^{e_1}}]^\top\right]^\top$ replaced by $\left[[\mathbf{\Gamma}^t_{c_2,d^{e_1}} - \mathbf{\Gamma}^t_{c_1,d^{e_1}}]^\top, ..., [\mathbf{\Gamma}^t_{c_R,d^{e_m}} - \mathbf{\Gamma}^t_{c_1,d^{e_1}}]^\top\right]^\top$, we have the similar conclusion with the set $\mathcal{D}$ replaced by $\mathcal{C} \otimes \mathcal{D}$.

**Theorem B.7.** *Denote* $h^{t=s,z}(d) := \left(\Gamma^t_{1,1}(d) - \Gamma^t_{1,1}(d^{e_1}), ..., \Gamma^t_{q_t,k_t}(d) - \Gamma^t_{1,1}(d^{e_1})\right)^\top$, $h(\mathcal{D}) := h^s(\mathcal{S}) \oplus h^z(\mathcal{Z}) \subset \mathbb{R}^{q_z * k_z} \oplus \mathbb{R}^{q_s * k_s}$, *then assumption (4) holds if* $h(\mathcal{D})$ *is not included in any zero-measure set of* $\mathbb{R}^{q_z * k_z} \oplus \mathbb{R}^{q_s * k_s}$. *Denote* $r_s := q_s * k_s$ *and* $r_z := q_z * k_z$.

*Proof.* With loss of generality, we assume that $r_s \leq r_z$. Denote $Q$ as the set of integers $q$ such that there exists $d^{e_2}, ..., d^{q+1}$ that the $\text{rank}([h^z(d^{e_2}), ..., h^z(d^{e_{q+1}})]) = \min(q, r_z)$ and $\text{rank}([h^s(d^{e_2}), ..., h^s(d^{e_{q+1}})]) = \min(q, r_s)$. Denote $u := \max(Q)$. We discuss two possible cases for $u$, respectively:

- Case 1. $u < r_s \leq r_z$. Then there exists $d^{e_2}, ..., d^{e_{u+1}}$ s.t. $h^z(d^{e_2}), ..., h^z(d^{e_{u+1}})$ and $h^s(d^{e_2}), ..., h^s(d^{e_{u+1}})$ are linearly independent. Then $\forall c$, we have $h^z(d) \in L(h^z(d^{e_2}), ..., h^z(d^{e_{u+1}}))$ or $h^s(d) \in L(h^s(d^{e_2}), ..., h^s(d^{e_{u+1}}))$. Therefore, so we have $h^z(d) \oplus h^s(d) \in [L(h^z(d^{e_2}), ..., h^z(d^{e_{u+1}})) \oplus \mathbb{R}^{r_s}] \cup [\mathbb{R}^{r_z} \oplus L(h^s(d^{e_2}), ..., h^s(d^{e_{u+1}}))]$, which has measure 0 in $\mathbb{R}^{r_z} \oplus \mathbb{R}^{r_s}$.

- Case 2. $r_s \leq u < r_z$. Then there exists $d^{e_2}, ..., d^{e_{u+1}}$ s.t. $h^z(d^{e_2}), ..., h^z(d^{e_{u+1}})$ are linearly independent and $rank([h^s(d^{e_1}), ..., h^s(d^{e_u})]) = r_s$. Then $\forall c$, we have $h^z(d) \in L(h^z(d^{e_1}), ..., h^z(d^{e_{u+1}}))$, which means that $h^z(d) \oplus h^s(d) \in L(h^z(d^{e_1}), ..., h^z(d^{e_{u+1}})) \oplus \mathbb{R}^{r_s}$, which has measure 0 in $\mathbb{R}^{r_z} \oplus \mathbb{R}^{r_s}$.

The above two cases are contradict to the assumption that $h(\mathcal{D})$ is not included in any zero-measure set of $\mathbb{R}^{r_z} \oplus \mathbb{R}^{r_s}$. $\square$

**Lemma B.8.** *Consider the cases when* $k_s \geq 2$. *Then suppose the assumptions in theorem B.5 are satisfied. Further assumed that*

- *The sufficient statistics* $\mathbf{T}^s_{i,j}$ *are twice differentiable for each* $i \in [q_s]$ *and* $j \in [k_s]$.

- $f_y$ *is twice differentiable.*

*Then we have* $M_s$ *in theorem B.5 is block permutation matrix.*

*Proof.* Directly applying [35, Theorem 2] with $f_x, A, b, \mathbf{T}, x$ replaced by $f_y, M_s, a_s, \mathbf{T}^s, y$. $\square$

**Lemma B.9.** *Consider the cases when* $k_s = 1$. *Then suppose the assumptions in theorem B.5 are satisfied. Further assumed that*

- *The sufficient statistics* $\mathbf{T}^s_i$ *are not monotonic for* $i \in [q_s]$.

- $g$ *is smooth.*

*Then we have* $M_s$ *in theorem B.5 is block permutation matrix.*

*Proof.* Directly applying [35, Theorem 3] with $f_x, A, b, \mathbf{T}, x$ replaced by $f_y, M_s, a_s, \mathbf{T}^s, y$. $\square$

*Proof of Theorem B.6.* According to theorem B.5, there exist invertible matrices $M_s$ and $M_z$ such that

$$\mathbf{T}(f_x^{-1}(x)) = A\tilde{\mathbf{T}}(\tilde{f}_x^{-1}(x)) + b$$
$$\mathbf{T}^s([f_x^{-1}]_{\mathcal{S}}(x)) = M_s\tilde{\mathbf{T}}^s([\tilde{f}_x^{-1}]_{\mathcal{S}}(x)) + a_s.$$
$$\mathbf{T}^s(f_y^{-1}(y)) = M_s\tilde{\mathbf{T}}^s(\tilde{f}_y^{-1}(y)) + a_s,$$

where $\mathbf{T} = [\mathbf{T}^{s,\top}, \mathbf{T}^{z,\top}]^\top$, and

$$A = \begin{pmatrix} M_s & 0 \\ 0 & M_z \end{pmatrix}. \tag{54}$$

By further assuming that the sufficient statistics $\mathbf{T}^s_{i,j}$ are twice differentiable for each $i \in [q_s]$ and $j \in [k_s]$ for $k_s \geq 2$ and not monotonic for $k_s = 1$. Then we have that $M_s$ is block permutation matrix. By further assuming that $\mathbf{T}^z_{i,j}$ are twice differentiable for each $i \in [n_z]$ and $j \in [k_z]$ for $k_z \geq 2$ and not monotonic for $k_z = 1$ and applying the lemma B.8 and B.9 respectively, we have that $A$ is block permutation matrix. Therefore, $M_z$ is also a block permutation matrix. $\qquad\square$

*Proof of Theorem 4.4.* We consider the general case when $\mathcal{C} := \cup_{r=1}^R \{c_r\}_{r=[R]}$. We have that

$$\sum_{r=1}^R p_\theta(x, y|c_r) \, \mathrm{P}(C = c_r|d^e) = \sum_{r=1}^R p_{\tilde\theta}(x, y|c_r) \, \mathrm{P}(C = c_R|d^e). \tag{55}$$

The Eq. (B.2) for each $e$ here can be replaced by

$$\sum_{r=1}^R p(c_r|d^e) \tilde p_{\mathbf{T}^s, \mathbf{\Gamma}^s, \mathbf{T}^z, \mathbf{\Gamma}^z, f_x}(x|c_r) = \sum_{r=1}^R \tilde p(c_r|d^e) \tilde p_{\tilde{\mathbf{T}}^s, \tilde{\mathbf{\Gamma}}^s, \tilde{\mathbf{T}}^z, \tilde{\mathbf{\Gamma}}^z, \tilde f_x}(x|d^e). \tag{56}$$

According to [2, Corollary 3], if we additionally assume that

$$\left\{ \left( \mathbf{T}^s([f^{-1}]_{\mathcal{S}}(x)), \mathbf{T}^z([f^{-1}]_{\mathcal{Z}}(x)) \right); \mathcal{B}(x) > 0 \right\} \text{ contains a non-empty set,}$$

then we have that the $p(c_r|d^e) = \tilde p(c_r|d^e)$ for each $r \in [R], e$. In other words, the $L := [P(C|d^{e_1})^\top, ..., P(C|d^{e_m})^\top]$ can be identified. Let $\Delta = [\tilde p_{\mathbf{T}^s, \mathbf{\Gamma}^s, \mathbf{T}^z, \mathbf{\Gamma}^z, f_x}(x|c_1) - \tilde p_{\tilde{\mathbf{T}}^s, \tilde{\mathbf{\Gamma}}^s, \tilde{\mathbf{T}}^z, \tilde{\mathbf{\Gamma}}^z, \tilde f_x}(x|c_1), \cdots, \tilde p_{\mathbf{T}^s, \mathbf{\Gamma}^s, \mathbf{T}^z, \mathbf{\Gamma}^z, f_x}(x|c_m) - \tilde p_{\tilde{\mathbf{T}}^s, \tilde{\mathbf{\Gamma}}^s, \tilde{\mathbf{T}}^z, \tilde{\mathbf{\Gamma}}^z, \tilde f_x}(x|c_m)]^\top$, then the concantenation of Eq. (56) in a matrix form can be written as $L\Delta = 0$. Since we have assumed in assumption (5) theorem 4.4 that the $L$ has full column rank, therefore we have that $\Delta = 0$, *i.e.* $\tilde p_{\mathbf{T}^s, \mathbf{\Gamma}^s, \mathbf{T}^z, \mathbf{\Gamma}^z, f_x}(x|c_r) = \tilde p_{\tilde{\mathbf{T}}^s, \tilde{\mathbf{\Gamma}}^s, \tilde{\mathbf{T}}^z, \tilde{\mathbf{\Gamma}}^z, \tilde f_x}(x|c_r)$ for each $r \in [R]$. The left proof is the same with the one in theorem B.6. $\qquad\square$

### B.3 Proof of Theorem 4.5

*Proof of Theorem 4.5.* Due to Eq. (55), it is suffices to prove the conclusion for every $c_r \in \{c_r\}_{r\in[R]}$. Motivated by [3, Theorem 2] that the distribution $p^e(s, z)$ defined on bounded set can be approximated by a sequence of exponential family with sufficient statistics denoted as polynomial terms, therefore the $\mathbf{T}^{t=s,z}$ are twice differentiable hence satisfies the assumption (2) in theorem 4.4 and assumption (1) in lemma B.8. Besides, the lemma 4 in [3] informs us that the KL divergence between $p_{\theta_0}(s, z|c_r)$ $(\theta_0 := (f_x, f_y, \mathbf{T}^z, \mathbf{T}^s, \mathbf{\Gamma}_0^z, \mathbf{\Gamma}_0^s)$ and $p_{\theta_1}(s, z|c_r)$ $(\theta_1 := (f_x, f_y, \mathbf{T}^z, \mathbf{T}^s, \mathbf{\Gamma}_1^z, \mathbf{\Gamma}_1^s)$ (the $p_{\theta_0}(s, z|c_r), p_{\theta_1}(s, z|c_r)$ belong to exponential family with polynomial sufficient statistics terms) can be bounded by the $\ell_2$ norm of $[(\mathbf{\Gamma}^s(c_r) - \mathbf{\Gamma}_1^s(c_r))^\top, (\mathbf{\Gamma}_0^z(c_r) - \mathbf{\Gamma}_1^z(c_r))^\top]^\top$. Therefore, $\forall \epsilon > 0$, there exists a open set of $\Gamma(c_r)$ such that the $D_{\mathrm{KL}}(p(s, z|c_r), p_\theta(s, z|c_r)) < \epsilon$. Such an open set is with non-zero Lebesgue measurement therefore can satisfy the assumption (4) in theorem 4.4, according to result in theorem B.7. The left is to prove that for any $p$ defined by a LaCIM following Def. 4.1, there is a sequence of $\{p_m\}_n \in \mathcal{P}_{\exp}$ such that the $d_{\mathrm{Pok}}(p, p_n) \to 0$ that is equivalent to $p_n \xrightarrow{d} p$. For any $A, B$, we consider to prove that

$$I_n \overset{\Delta}{=} \left| p(x \in A, y \in B|c_r) - p_n(x \in A, y_n \in B|c_r) \right| \to 0, \tag{57}$$

where $p_n(x \in A, y_n \in B|c_r) = \int_{\mathcal{S}} \int_{\mathcal{Z}} p(x \in A|s, z) p(y_n \in B|s) p_n(s, z|c_r) ds dz$ with

$$y_n(i) = \frac{\exp((f_{y,i}(\boldsymbol{s}) + \varepsilon_{y,i})/T_n)}{\sum_i \exp((f_{y,i}(\boldsymbol{s}) + \varepsilon_{y,i})/T_n)}, \quad i = 1, ..., k, \tag{58}$$

for $y \in \mathbb{R}^k$ denoting the $k$-dimensional one-hot vector for categorical variable and $\varepsilon_{y,1,...,k}$ are Gumbel i.i.d. According to [44, Proposition 1] that the $y_n(i) \xrightarrow{d} y(i)$ with

$$p(y(i) = 1) = \frac{\exp(f_{y,i}(\boldsymbol{s}))}{\sum_i \exp((f_{y,i}(\boldsymbol{s}))}, \quad as \ T_n \to 0. \tag{59}$$

As long as $f_y$ is smooth, we have that the $p(y_n|s)$ is continuous. We have that

$$
\begin{aligned}
I_n &= \left| p(x \in A, y \in B|c_r) - \int_{\mathcal{S} \times \mathcal{Z}} p(x \in A|s, z) p(y_n \in B|s) p_n(s, z|c_r) ds dz \right| \\
&\leq \left| p(x \in A, y \in B|c_r) - p(x \in A, y_n \in B|c_r) \right| \\
&\quad + \left| p(x \in A, y_n \in B|c_r) - \int_{\mathcal{S} \times \mathcal{Z}} p(x \in A|s, z) p(y_n \in B|s) p_n(s, z|c_r) ds dz \right| \\
&= \left| \int_{\mathcal{S} \times \mathcal{Z}} p(x \in A|s, z) \left( p(y \in B|s) - p(y_n \in B|s) \right) p(s, z|c_r) ds dz \right| \\
&\quad + \left| \int_{\mathcal{S} \times \mathcal{Z}} p(x \in A|s, z) p(y_n \in B|s) \left( p(s, z|c_r) - p_n(s, z|c_r) \right) \right| \\
&\leq \underbrace{\left| \int_{M_s \times M_z} p(x \in A|s, z) \left( p(y \in B|s) - p(y_n \in B|s) \right) p(s, z|c_r) ds dz \right|}_{I_{n,1}} \\
&\quad + \underbrace{\left| \int_{(M_s \times M_z)^{c_r}} p(x \in A|s, z) \left( p(y \in B|s) - p(y_n \in B|s) \right) p(s, z|c_r) ds dz \right|}_{I_{n,2}} \\
&\quad + \underbrace{\left| \int_{M_s \times M_z} p(x \in A|s, z) p(y_n \in B|s) \left( p(s, z|c_r) - p_n(s, z|c_r) \right) \right|}_{I_{n,3}} \\
&\quad + \underbrace{\left| \int_{(M_s \times M_z)^{c_r}} p(x \in A|s, z) p(y_n \in B|s) \left( p(s, z|c_r) - p_n(s, z|c_r) \right) \right|}_{I_{n,4}}.
\end{aligned}
\tag{60}
$$

For $I_{n,1}$, if $y$ is itself additive model with $y = f_y(\boldsymbol{s}) + \varepsilon_y$, then we just set $y_n \overset{d}{=} y$, then we have that $I_{n,1} = 0$. Therefore, we only consider the case when $y$ denotes the categorical variable with softmax distribution, *i.e.*, Eq. (59). $\forall c_r \in \mathcal{C} := \{c_1, ..., c_R\}$ and $\forall \epsilon > 0$, there exists $M_s^{c_r}$ and $M_z^{c_r}$ such that $p(s, z \in M_s^{c_r} \times M_z^{c_r}|c_r) \leq \epsilon$; Denote $M_s \overset{\Delta}{=} \cup_{k=1}^m M_s^{c_r}$ and $M_z \overset{\Delta}{=} \cup_{k=1}^m M_z^{c_r}$, we have that $p(s, z \in M_s \times M_z|c) \leq 2\epsilon$ for all $c_r \in \mathcal{C}$. Since $\forall s_1 \in M_s$, $\exists N_{s_1}$ such that $\forall n \geq N_{s_1}$, we have that $\left| p(y \in B|s_1) - p(y \in B|s_1) \right| \leq \epsilon$ from that $y_n \overset{d}{\to} y$. Besides, there exists open set $\mathcal{O}_{s_1}$ such that $\forall s \in \mathcal{O}_{s_1}$ and

$$
\left| p(y \in B|s_1) - p(y \in B|s_1) \right| \leq \epsilon, \ \left| p(y_n \in B|s_1) - p(y_n \in B|s_1) \right| \leq \epsilon.
$$

Again, according to Heine–Borel theorem, there exists finite $s$, namely $s_1, ..., s_l$ such that $M_s \subset \cup_{i=1}^l \mathcal{O}(s_i)$. Then there exists $N \overset{\Delta}{=} \max\{N_{s_1}, ..., N_{s_l}\}$ such that $\forall n \geq N$, we have that

$$
\left| p(y \in B|s) - p(y_n \in B|s) \right| \leq 3\epsilon, \ \forall s \in M_s.
\tag{61}
$$

Therefore, $I_{n,1} \leq \int_{M_s \times M_z} 3\epsilon p(x \in A|s, z) p(s, z|c) ds dz \leq 3\epsilon$. Hence, $I_{n,1} \to 0$ as $n \to \infty$. Besides, we have that $I_{n,2} \leq \int_{M_s \times M_z} 2\epsilon p(s, z|c_r) ds dz \leq 2\epsilon$. Therefore, we have that $\left| \int_{\mathcal{S} \times \mathcal{Z}} p(x \in \right.$

$A|s,z)\left(p(y \in B|s) - p(y_n \in B|s)\right)p(s,z|c_r)dsdz\big| \to 0$ as $n \to \infty$. For $I_{n,3}$, we have that

$$I_{n,3} = \left| \int_{M_s \times M_z} p(x \in A|s,z)p(y_n \in B|s)\mathbb{1}(s,z \in M_s \times M_z)\left(p(s,z|c_r) - p_n(s,z|c_r)\right)dsdz \right|$$

$$\leq \underbrace{\left| \int_{M_s \times M_z} p(x \in A|s,z)p(y_n \in B|s)p(s,z|c_r)\left(\frac{1}{p(s,z \in M_s \times M_z|c_r)} - 1\right)dsdz \right|}_{I_{n,3,1}}$$

$$+ \underbrace{\left| \int_{M_s \times M_z} p(x \in A|s,z)p(y_n \in B|s)p(s,z|c_r)\left(\frac{1}{p(s,z \in M_s \times M_z|c_r)} - 1\right)dsdz \right|}_{I_{n,3,2}}.$$

$$(62)$$

The $I_{n,3,1} \leq \frac{\epsilon}{1-\epsilon}$. Denote $\tilde{p}(s,z|c_r) := \frac{p(s,z|c_r)\mathbb{1}(s,z \in M_s \times M_z)}{p(s,z \in M_s \times M_z|c_r)}$, according to [3, Theorem 2], there exists a sequence of $p_n(s,z|c)$ defined on a compact support $M_s \times M_z$ such that $\forall c_r \in \mathcal{C}$, we have that

$$p_n(s,z|c_r) \xrightarrow{d} p(s,z|c_r).$$

Applying again the Heine–Borel theorem, we have that $\forall \epsilon, \exists N$ such that $\forall n \geq N$, we have

$$\left| \tilde{p}(s,z|c_r) - p_n(s,z|c_r) \right| \leq \epsilon, \tag{63}$$

which implies that $I_{n,3,2} \to 0$ as $n \to \infty$ combining with the fact that $p(x,y|s,z)$ is continuous with respect to $s,z$. For $I_{n,4}$, we have that

$$I_{n,4} = \left| \int_{M_s \times M_z} p(x \in A|s,z)p(y_n \in B|s)p(s,z|c_r) \right| \leq \left| \int_{M_s \times M_z} p(s,z|c_r) \right| \leq \epsilon, \tag{64}$$

where the first equality is from that the $p_n(s,z|c_r)$ is defined on $M_s \times M_z$. Then we have that

$$\left| \int_{\mathcal{S} \times \mathcal{Z}} p(x \in A|s,z)p(y_n \in B|s)\left(p(s,z|c_r) - p_n(s,z|c_r)\right) \right| \to 0, \ as \ n \to \infty. \tag{65}$$

The proof is completed. $\square$

## C   Reparameterization for LaCIM

We provide an alternative training method to avoid parameterization of prior $p(s,z|\tilde{d}^e)$ to increase the diversity of generative models in different environments. Specifically, motivated by [26] that any distribution can be transformed to isotropic Gaussian with the density denoted by $p_{\mathrm{Gau}}$, we have that for any $e \in \mathcal{E}_{\mathrm{train}}$, we have

$$p^e(x,y) = \int_{\mathcal{S} \times \mathcal{Z}} p_{f_x}(x|s,z)p_{f_y}(y|s)p(s,z|\tilde{d}^e)dsdz$$

$$= \int_{\mathcal{S} \times \mathcal{Z}} p(x|(\varphi_s^e)^{-1}(s'), (\varphi_z^e)^{-1}(z'))p(y|\varphi_s(s'))p(s',z')ds'dz',$$

with $s', z' := \varphi_s^e(s), \varphi_z^e(z) \sim \mathcal{N}(0,I)$. We can then rewrite ELBO for LaCIM for environment $e$ as:

$$\mathcal{L}_{\theta,\psi,\varphi^e}^e = \mathbb{E}_{p^e(x,y)}\left[-\log q_\psi^e(y|x)\right]$$

$$+ \mathbb{E}_{p^e(x,y)}\left[-\mathbb{E}_{q_\psi^e(s,z|x)}\frac{q_\psi(y|(\varphi_s^e)^{-1}(s))}{q_\psi^e(y|x)}\log \frac{p_\theta((\varphi_s^e)^{-1}(s), (\varphi_z^e)^{-1}(z))p(s,z)}{q_\psi^e(s,z|x)}\right], \tag{66}$$

where $p(s,z)$ denotes the density function of isotropic gaussian.

# D More Related Works

## D.1 Identifiability

Earlier works that identify the latent confounders rely on strong assumptions regarding the causal structure, such as the linear model from latent to observed variable or ICA in which the latent component are independent [66], or noise-free model [64, 12]. The [22, 31] extend to the Additive Noise Model (ANM) and other causal discovery assumptions. Although the [41] relaxed the constraints put on the causal structure, it required the latent noise is with small strength, which does not match with many realistic scenarios, such as the structural MRI of Alzheimer's Disease considered in our experiment. The works which also based on the independent component analysis (ICA), *i.e.*, the latent variables are (conditionally) independent, include [12, 14]; recently, a series of works extend the above results to deep nonlinear ICA [25, 27, 35, 36, 75]. However, these works require that the value of confounder of these latent variables is fixed, which cannot explain the spurious correlation in a single dataset. In contrast, our result incorporates these scenarios by assuming that each sample has a specific value of the confounder.

## D.2 Comparisons with data augmentation & architecture design

The goal of data augmentation [65] is increase the variety of the data distribution, such as geometrical transformation [34, 73], flipping, style transfer [16], adversarial robustness [45]. On the other way round, an alternative kind of approaches is to integrate into the model corresponding modules that improve the robustness to some types of variations, such as [77, 47].

However, these techniques can only make effect because they are included in the training data for neural network to memorize [79]; besides, the improvement is only limited to some specific types of variation considered. As analyzed in [78, 39], the data augmentation trained with empirical risk minimization or robust optimization [6] such as adversarial training [45, 58] can only achieve robustness on interpolation (convex hull) rather than extrapolation of training environments.

## D.3 Comparisons with existing works in domain adaptation

Apparently, the main difference lies in the problem setting that (i) the domain adaptation (DA) can access the input data of the target domain while ours cannot; and (ii) our methods need multiple training data while the DA only needs one source domain. For methodology, our LaCIM shares insights but different from DA. Specifically, both methods assume some types of invariance that relate the training domains to the target domain. For DA, one stream is to assume the same conditional distribution shared between the source and the target domain, such as covariate shifts [23, 5, 33, 70] in which $P(Y|X)$ are assumed to be the same across domains, concept shifts [81] in which the $P(X|Y)$ is assumed to be invariant. Such an invariance is related to representation, such as $\Phi(X)$ in [82] and $P(Y|\Phi(X))$ in [50, 15, 46].

However, these assumptions are only distribution-level rather than the underlying causation which takes the data-generating process into account. Taking the image classification again as an example, our method first proposes a causal graph in which the latent factors are introduced as the explanatory/causal factor of the observed variables. These are supported by the framework of generative model [35, 36, 37, 71] which has a natural connection with the causal graph [59] that the edge in the causal graph reflects both the causal effect and also the generating process. Until now, perhaps the most similar work to us is [56] and [75] which also need multiple training domains and get access to a few samples in the target domain. Both works assume a similar causal graph with us but unlike our LaCIM, they do not separate the latent factors which can not explain the spurious correlation learned by supervised learning [28]. Besides, the multiple training datasets in [56] refer to intervened data which may hard to obtain in some applications. We have verified in our experiments that explicitly disentangle the latent variables into two parts can result in better generalization to new distribution than mixing them together.

## D.4 Comparisons with domain generalization

For domain generalization (DG), similar to the invariance assumption in DA, a series of works proposed to align the representation $\Phi(X)$ that is assumed to be invariant across domains [42, 43, 49].

As discussed above, these methods lack the deep delving of the underlying causal structure and preclude the variations of unseen domains.

Recently, a series of works leverage causal invariance to enable OOD generalization on unseen domains, such as [29] which learns the representation that is domain-invariant. Notably, the Invariant Causal Prediction [52] formulates the assumption in the definition of Structural Causal Model and assumes that $Y = X_{\mathcal{S}} \beta^{\star}_{\mathcal{S}} + \varepsilon_Y$ where $\varepsilon_Y$ satisfies Gaussian distribution and $\mathcal{S}$ denotes the subset of covariates of $X$. The [55, 10] relaxes such an assumption by assuming the invariance of $f_y$ and noise distribution $\varepsilon_y$ in $Y \leftarrow f_y(X_{\mathcal{S}}, \varepsilon_y)$ which induces $P(Y|X_{\mathcal{S}})$. A similar assumption is also adopted in [40]. However, these works causally related the output to the observed input, which may not hold in many real applications in which the observed data is sensory-level, such as audio waves and pixels. It has been discussed in [8, 7] that the causal factor should be high-level abstractions/concepts. The [21] considers the style transfer setting in which each image is a linear combination of shape-related variable and contextual-related variable, which respectively correspond to $S$ and $Z$ in our LaCIM in which the nonlinear mechanism (rather than linear combination in [21]) is allowed. Besides, during testing, our method can generalize to the OOD sample with interventions such as adversarial noise and contextual intervention.

Recently, the most notable work is Invariant Risk Minimization [1], which will be discussed in detail in the subsequent section.

### D.5 Comparisons with Invariant Risk Minimization [1] and references therein

The Invariant Risk Minimization (IRM) [1] assumes the existence of invariant representation $\Phi(X)$ that induces the optimal classifier for all domains, *i.e.*, the $\mathbb{E}[Y|Pa(Y)]$ is domain-independent in the formulation of SCM. Similar to our LaCIM, the $Pa(Y)$ can refer to latent variables. Besides, to identify the invariance and the optimal classifier, the training environments also need to be diverse enough. As aforementioned, this assumption is almost necessary to differentiate the invariance mechanism from the variant ones.

The difference of our LaCIM with IRM lies in two aspects: the definition of $Y$ and the methodology. For the label, the IRM defines it as the one obtained after the image (*e.g.*, one label the "dog" based on the image he/she observes); while the label $Y$ is generated concurrently with $X$, that is, the $Y$ is dependent on the semantic features he/she observed. Consider the following scenario as an illustration: the photographer takes an image $X$ and records the label $Y$ at the same time. Besides, in terms of methodology, the theoretical claim of IRM only holds in linear case; in contrast, the $f_x, f_y$ are allowed to be nonlinear.

Some other works share a similar spirit with or based on IRM. The Risk-Extrapolation (REx) [39] proposed to enforce the similar behavior of $m$ classifiers with a variance of which proposed as the regularization function. The work in [78] proposed a Quasi-distribution framework that can incorporate empirical risk minimization, robust optimization, and REx. It can be concluded that the robust optimization only generalizes the convex hull of training environments (defined as interpolation) and the REx can generalize extrapolated combinations of training environments. This work lacks a model of underlying causal structure, although it performs similarly to IRM experimentally. Besides, the [74] proposed to unpool the training data into several domains with different environments and leverages [1] to learn invariant information for the classifier. Recently, the [4] also assumes the invariance to be generating mechanisms and can generalize the capability of IRM when unobserved confounders exist. However, this work also lacks the analysis of identifiability results.

## E Implementation Details and More Experimental Results

All experiments are performed on a workstation with 8 RTX 2080 Ti GPUs. The workstation has Intel(R) Xeon(R) E5-2699A v4 CPU and 256 GB RAM.

The code and data please refer to [https://anonymous.4open.science/r/dfe66206-90f9-4d3f-91b4-2dd8146a2aa2/](https://anonymous.4open.science/r/dfe66206-90f9-4d3f-91b4-2dd8146a2aa2/). The datasets we used are all public and widely applied for different tasks. Therefore, The data do not contain personally identifiable information nor offensive content.

## E.1 Simulation

**Data Generation** We set $m = 5$. We set $q_d = q_s = q_z = q_y = 2$ and $q_x = 4$. For each environment $e \in [m]$ with $m = 5$, we generate 1000 samples $\mathcal{D}^e = \{x_i, y_i\} \overset{i.i.d}{\sim} \int p_{f_x}(x|s,z)p_{f_y}(y|s)p(s,z|c)p(c|d^e)dsdzdc$. The $d^e = (\mathcal{N}(0, I_{q_d \times q_d}) + 5 * e) * 2$; the $c|d^e \sim \mathcal{N}(d^e, I)$; the $s, z|c \sim \mathcal{N}\left(\mu_{\theta^\star_{s,z}}(s,z|c), \sigma^2_{\theta^\star_{s,z}}(s,z|c)\right)$ with $\mu_{\theta^\star_{s,z}} = A^\mu_{s,z} * c$ and $\log \sigma_{\theta^\star_{s,z}} = A^\sigma_{s,z} * c$ ($A^\mu_{s,z}$, $A^\sigma_{s,z}$ are random matrices); the $x|s,z \sim \mathcal{N}\left(\mu_{\theta^\star_x}(x|s,z), \sigma^2_{\theta^\star_x}(x|s,z)\right)$ with $\mu_{\theta^\star_{s,z}} = h(A^{\mu,3}_x * h(A^{\mu,2}_x * h(A^{\mu,2}_x * [s^\top, z^\top]^\top])))$ and $\log \sigma_{\theta^\star_{s,z}} = h(A^{\sigma,3}_x * h(A^{\sigma,2}_x * h(A^{\sigma,2}_x * [s^\top, z^\top]^\top])))$ ($h$ is LeakyReLU activation function with slope $= 0.5$ and $A^{\mu,i=1,2,3}_x, A^{\sigma,i=1,2,3}_x$ are random matrices); the $y|s$ is similarly to $x|s,z$ with $A^{\mu,i=1,2,3}_x, A^{\sigma,i=1,2,3}_x$ respectively replaced by $A^{\mu,i=1,2,3}_y, A^{\sigma,i=1,2,3}_y$.

**Implementation Details** We parameterize $p_\theta(s,z|\tilde{d}^e)$, $q_\psi(s,z|x,y,\tilde{d}^e)$, $p_\theta(x|s,z)$ and $p_\theta(y|s)$ as 3-layer MLP with the LeakyReLU activation function. The Adam with learning rate $5 \times 10^{-4}$ is implemented for optimization. We set the batch size as 512 and run for 2,000 iterations in each trial.

**Visualization.** As shown from the visualization of $S$ is shown in Fig. 4, our LaCIM can identify the causal factor $S$.

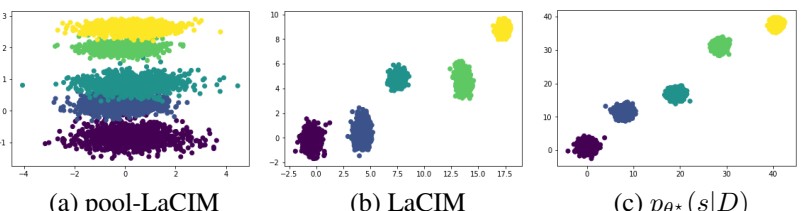

(a) pool-LaCIM         (b) LaCIM         (c) $p_{\theta^\star}(s|D)$

Figure 4: Estimated posterior by (a) pool-LaCIM; (b) LaCIM and (c) the ground-truth. As shown, the LaCIM can identify the $S$ (up to permutation and point-wise transformation), which validates the Eq. (2) in theorem 4.4.

## E.2 Implementation Details for Optimization over $S, Z$

Recall that we first optimize $s^*, z^*$ according to

$$s^*, z^* = \arg \max_{s,z} \log p_\theta(x|s,z).$$

We first sample some initial points from each posterior distribution $q^e_\psi(s|x)$ and then optimize for 50 iterations. We using Adam as optimizer, with learning rate as 0.002 and weight decay 0.0002. The Fig. 5 shows the optimization effect of one run in CMNIST. As shown, the test accuracy keeps growing as iterates. For time saving, we chose to optimize for 50 iterations.

## E.3 Implementations For Baseline

The networks of ERM contain two parts: (i) feature extractor, followed by (ii) classifier. The network structure of the feature extractor and classifier for ERM is the same as that of our encoder and our $p_\theta(y|s)$. We adopt the same structure for IRM as ERM. DANN adopts the same structure of ERM and an additional domain classifier which is the same as that of $p_\theta(y|s)$. sVAE adopt the same structure as LaCIM-$d$ with the exception that the $p_\theta(y|s)$ is replaced by $p_\theta(y|z,s)$. MMD-AAE adopts the same structure of encoder, decoder, and classifier as LaCIM, and an additional 2-layer MLP with channel 256-256-$dim_z$ is used to extract latent $z$. The detailed number of parameters and channel size on each dataset for each method are summarized in Tab. 7, 8.

## E.4 Colored MNIST

**Implementation details** The network structure for inference model is composed of two parts, with the first part shared among all environments and multiple branches corresponding to each environment for the second part. The network structure of the first-part encoder is composed of four blocks, each

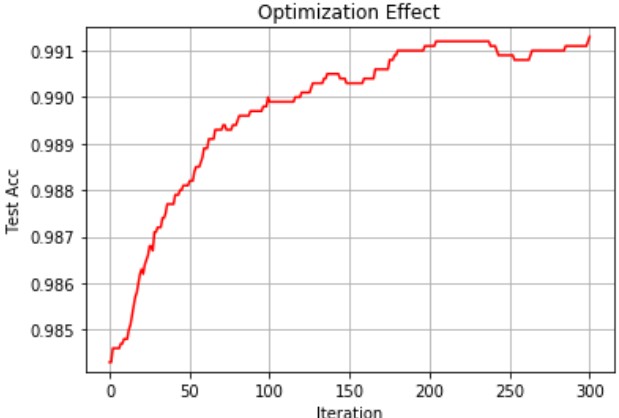

Figure 5: The optimization effect in CMNIST, starting from the point with initial sampling from inference model $q$ of each branch. As shown, the test accuracy increases as iterates.

block is the sequential of Convolutional Layer (Conv), Batch Normalization (BN), ReLU and max-pooling with stride 2. The output number of feature map is accordingly 32, 64, 128, 256. The second part network structure that output the mean and log-variance of $S, Z$ is Conv-bn-ReLU(256) $\rightarrow$ Adaptive (1) $\rightarrow$ FC(256, 256) $\rightarrow$ ReLU $\rightarrow$ FC(256, $q_{t=s,z}$) with FC stands for fully-connected layer. The structure of $\varphi_{t=s,z}$ in Eq. (66) is FC($q_t$, 256) $\rightarrow$ ReLU $\rightarrow$ FC(256, $q_t$). The network structure for generative model $p_\theta(x|s, z)$ is the sequential of three modules: (i) Upsampling with stride 2; (ii) four blocks of Transpose-Convolution (TConv), BN and ReLU with respective output dimension being 128, 64, 32, 16; (iii) Conv-BN-ReLU-Sigmoid with number of channels in the output as 3, followed by cropping step in order to make the image with the same size as input dimension, *i.e.*, $3 \times 28 \times 28$. The network structure for generative model $p_\theta(y|s)$ is commposed of FC (512) $\rightarrow$ BN $\rightarrow$ ReLU $\rightarrow$ FC (256) $\rightarrow$ BN $\rightarrow$ ReLU $\rightarrow$ FC ($|\mathcal{Y}|$). The $q_{t=s,z}$ is set to 32. We implement SGD as optimizer with learning rate 0.5, weight decay $1e-5$ and we set batch size as 256. The total training epoch is 80.

We first explain why we do not flip $y$ with 25% in the manuscript, and then provide further exploration of our method for the setting with flipping $y$.

**Invariant *Causation* v.s. Invariant *Correlation* by Flipping $y$ in [1]** The $y$ is further flipped with 25% to obtain the final label in IRM setting and this step is omitted in ours. The difference lies in the definition for the label $Y$ and the invariance. Our LaCIM defines invariance as the causal relation between $S$ and the label $Y$, while the one in IRM can refer to correlation since randomly flipping $Y$ can break the relations between $S$ and $Y$. As illustrated in Handwriting Sample Form in Fig. 6 in [18], the generating direction should be $Y \rightarrow X$. If we denote $\tilde{Y}$ as the flipped $Y$ (*a.k.a*, the final label in IRM), then there $X \leftarrow Y^\star \rightarrow \tilde{Y}$. In this case, the $\tilde{Y}$ is correlated rather than causally related to the digit $X$. For our LaCIM, the $Y$ is generated by the causal semantic factor of $X$, hence will capture the information from the digit.

**Experiment with IRM setting** We further conduct the experiment on IRM setting, with the final label $y$ defined by flipping the original label with 25%, and further color $p^e$ proportions of digits with corresponding color-label mapping. If we assume the original ground-truth label to be the effect of the digit number of $S$, then the anti-causal relation with $Z$ and $Y$ can make the identifiability of $S$ difficult in this flipping scenario. Note that the causal effect between $S$ and $Y$ is invariant across domains, therefore we adopt to regularize the branch of inferring $S$ to be shared among inference models for multiple environments. Besides, we regularize the causal effect between $S$ and $Z$ to be shared among different environments via pairwise regularization. The combined loss is formulated as:

$$\tilde{\mathcal{L}}_{\psi,\theta} = \mathcal{L}_{\psi,\theta} + \frac{\gamma}{2m^2} \sum_{i=1}^{m} \sum_{j=1}^{m} \|\mathbb{E}_{(x,y)\sim p^{e_i}(x,y)}[y|x] - \mathbb{E}_{(x,y)\sim p^{e_j}(x,y)}[y|x]\|_2^2,$$

where $\gamma > 0$ denotes the regularization hyperparameter. The $q_\psi^e(s, z|x)$ in Eq. (66) factorized as $q_{\psi_z^e}(z)q_{\psi_s}(s)$ and $\varphi_s$ shared among $m$ environments. The appended loss is coincide with recent

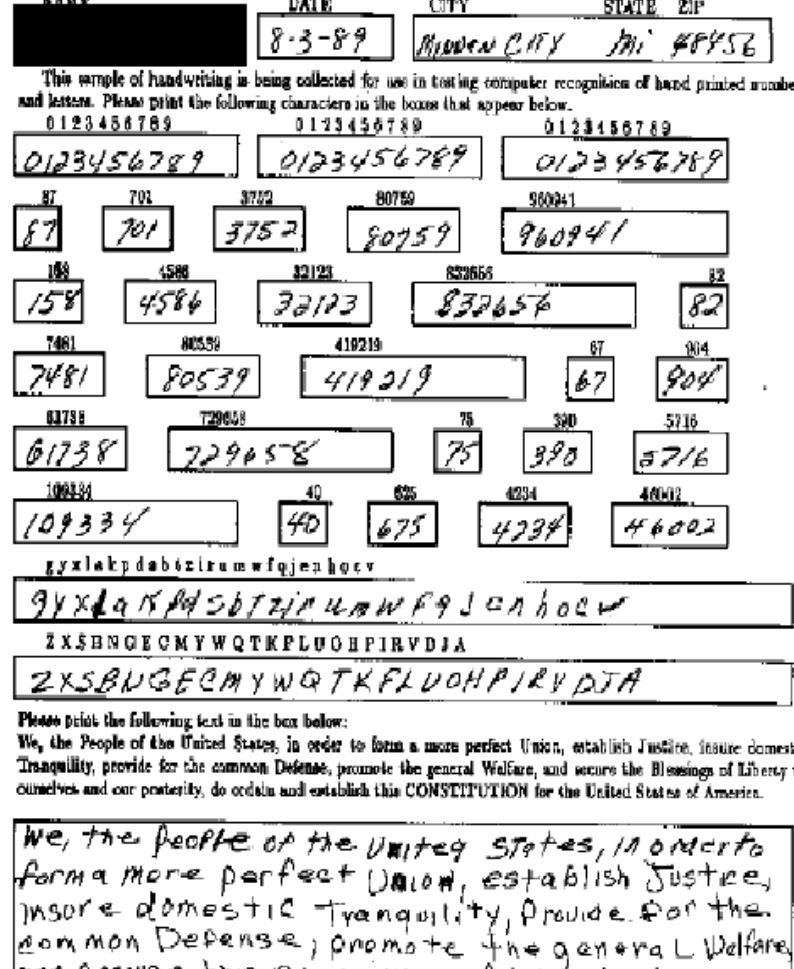

Figure 6: Hand-writing Sample Form. The writer print the digit/character (*i.e.*, $X$) with the label (*i.e.*, $Y$) provided first.

study Risk-Extropolation (REx) in [39], with the difference of separating causal factor $S$ from others. We name such a training method as LaCIM-REx. For implementation details, in addition to shared encoder regarding $S$, we set learning rate as 0.1, weight decay as 0.0002, batch size as 256. we have that $p(y|x) = \int_S q_{\psi_s}(s|x)p_\theta(y|\varphi_s(s))$ for any $x$. We consider two settings: setting#1 with $m2$ and $p^{e_1} = 0.9, p^{e_2} = 0.8$; and setting#2 with $m = 4$ with $p^{e_1} = 0.9, p^{e_2} = 0.8, p^{e_3} = 0.7, p^{e_4} = 0.6$. We only report the number of IRM since the cross entropy performs poorly in both settings. As shown in Tab. 2, our model performs comparably than IRM [1] due to separation of $S$ znd $Z$.

### E.5  NICO

**Implementation Details** Due to the size difference among images, we resize each image into 256×256. The network structure of $p_\theta(z, s|\tilde{d}^e), q_\psi(z, s|x, \tilde{d}^e), p_\theta(x|z, s), p_\theta(y|s)$ for cat/dog clas-

Table 2: Accuracy (%) of Colored MNIST on IRM setting in [1]. Average over three runs.

|  | IRM | LaCIM-REx (**Ours**) |
|---|---|---|
| $m = 2$ | $67.15 \pm 3.79$ | $\mathbf{67.57 \pm 1.37}$ |
| $m = 4$ | $69.37 \pm 1.14$ | $\mathbf{69.50 \pm 0.57}$ |

sification is the same with the one implemented in early prediction of Alzheimer's Disease with exception of 3D convolution/Deconvolution replaced by 2D ones. For each model, we train for 200 epochs using sgd, with the learning rate (lr) set to 0.01, and after every 60 epochs, the learning rate is multiplied by lr decay parameter that is set to 0.2. The weight decay coefficients parameter is set to $5 \times 10^{-4}$. The batch size is set to 30. The training environments which are characterized by $c$ can be referenced in Tab. 3. For visualization, we implemented the gradient-based method [67] to visualize the neuron (in a fully connected layer of CE $x \rightarrow y$ and the $s$ layer of LaCIM that is most correlated to label $y$.

**The $D$ for $m$ environments** We summarize the $D$ of $m = 8$ and $m = 14$ environments in Table 3. Since the distribution of $S, Z$ depends on $D$, we simply define $D$ as the parameterization of $S, Z$. In this context, such a parameterization refers to the proportions of (dog in grass, dog in snow; cat in grass, cat in snow); therefore $D \in \mathbb{R}^4$. As shown, the value of $D$ in the test domain is the extrapolation of the training environments, *i.e.*, the $d^{\text{test}}$ is not included in the convex hull of $\{d^{e_i}\}_{i=1}^{14}$.

Table 3: Training and test environments (characterized by $D$)

|  | cat% on grass | dog% on grass | cat% on snow | cat% on snow |
|---|---|---|---|---|
|  | Training Environment | | | |
| Env#1 ($d^{e_1}$) | 0.6 | 0.4 | 0.1 | 0.9 |
| Env#2 ($d^{e_2}$) | 0.8 | 0.2 | 0.1 | 0.9 |
| Env#3 ($d^{e_3}$) | 0.5 | 0.5 | 0.2 | 0.8 |
| Env#4 ($d^{e_4}$) | 0.8 | 0.2 | 0.2 | 0.8 |
| Env#5 ($d^{e_5}$) | 0.7 | 0.3 | 0.2 | 0.8 |
| Env#6 ($d^{e_6}$) | 0.8 | 0.2 | 0.3 | 0.7 |
| Env#7 ($d^{e_7}$) | 0.7 | 0.3 | 0.3 | 0.7 |
| Env#8 ($d^{e_8}$) | 0.9 | 0.1 | 0.3 | 0.7 |
| Env#9 ($d^{e_9}$) | 0.4 | 0.6 | 0.3 | 0.7 |
| Env#10 ($d^{e_{10}}$) | 0.6 | 0.4 | 0.3 | 0.7 |
| Env#11 ($d^{e_{11}}$) | 0.5 | 0.5 | 0.4 | 0.6 |
| Env#12 ($d^{e_{12}}$) | 0.4 | 0.6 | 0.4 | 0.6 |
| Env#13 ($d^{e_{13}}$) | 0.7 | 0.3 | 0.4 | 0.6 |
| Env#14 ($d^{e_{14}}$) | 0.8 | 0.2 | 0.4 | 0.6 |
|  | Testing Environment | | | |
| Env Test $d^{\text{test}}$ | 0.2 | 0.8 | 0.8 | 0.2 |

**More Visualization Results** Fig. 7 shows more visualization results.

**Results on Intervened Data.** We test the robustness of our model on intervened data generated from NICO. Each image is generated from a paired image (image A, image B): combining the scene of image A with the animal from image B. This is equivalent to intervention on the latent space. We generate 120 images. As shown in Tab 4, our LaCIM can outperform others.

Table 4: ACC on intervened dataset from NICO.

| Method | IRM | DANN | NCBB |
|---|---|---|---|
| **ACC** | 50.00 | 49.17 | 49.17 |
| Method | MMD-AAE | DIVA | LaCIM (**Ours**) |
| **ACC** | 49.17 | 50.00 | **55.00** |

**Generation of Intervened Data.** For generating an intervened sample, we replace the scene of an image with the scene from another image, as shown in Fig. 8. This process can be viewed as breaking

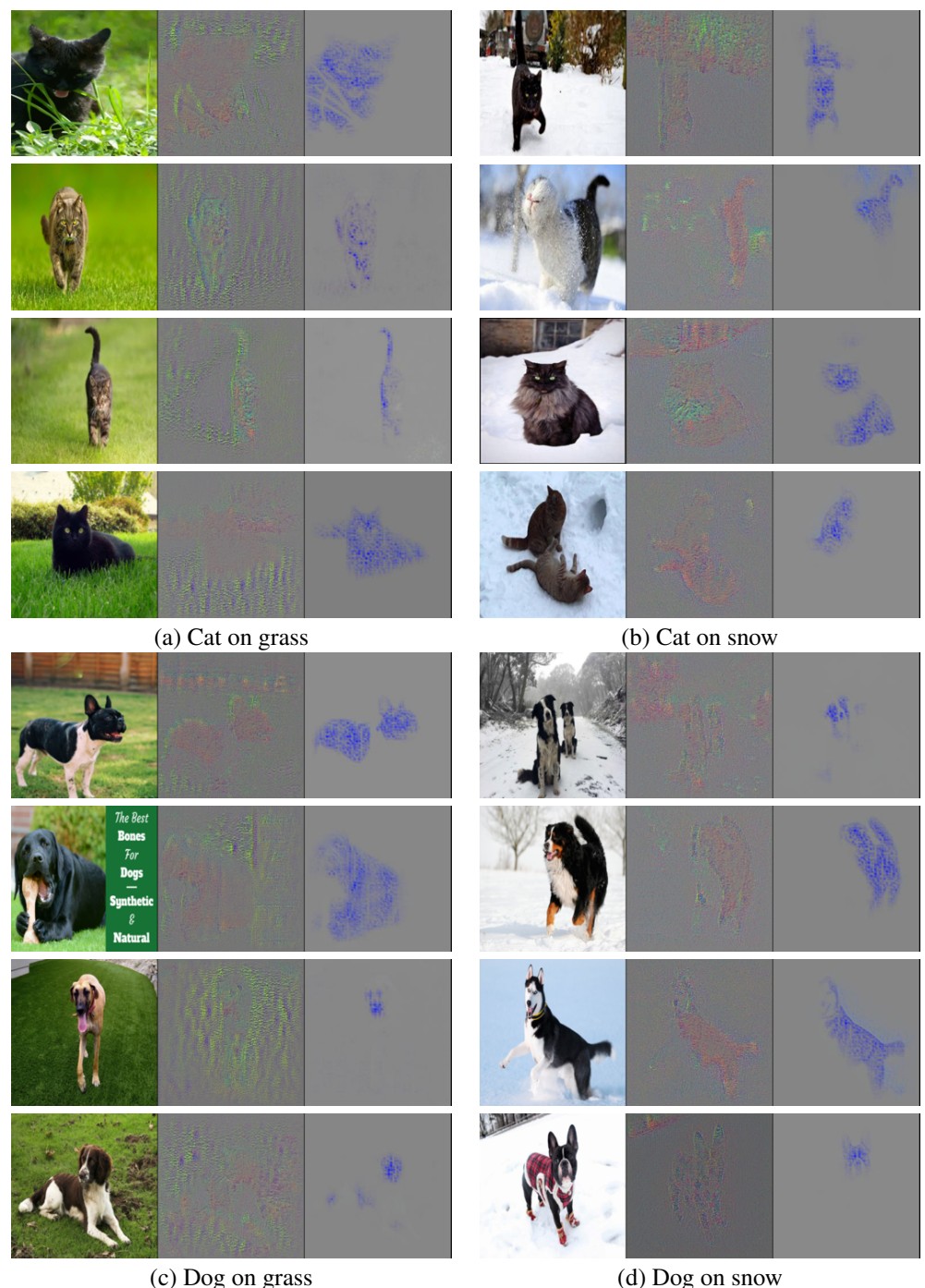

(a) Cat on grass        (b) Cat on snow

(c) Dog on grass        (d) Dog on snow

Figure 7: Visualization on the NICO via gradient-based method [67] for ERM and LaCIM. The selected images are (a) cat on grass, (b) cat on snow, (c) dog on grass and (d) dog on snow.

the dependency between $Z$ and $Z$. We generate 120 images, including 30 images of types: cat on grass, dog on grass, cat on snow, and dog on grass.

### E.6 ADNI

**Dataset Description.** The dataset contains in total 317 samples with 48 AD, 75 NC, and 194 MCI.

| Dog on snow | Cat on snow |
|:---:|:---:|
| 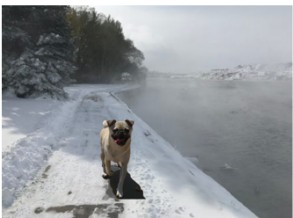 | 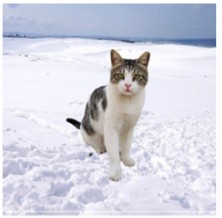 |
| Dog on grass | Cat on grass |
| 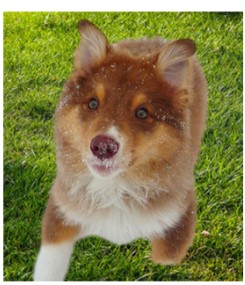 | 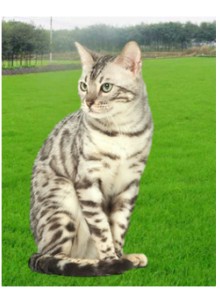 |

Figure 8: The constructed interventional dataset which includes of dog on snow, dog on grass, cat on snow, and dog on grass.

**Denotation of Attributes** $D$. The $D \in \mathbb{R}^9$ includes personal attributes (*e.g.*, age [19], gender [76] and education years [48] that play as potential risks of AD), gene ($\varepsilon_4$ allele), and biomarkers (*e.g.*, changes of CSF, TAU, PTAU, amyloid$_\beta$, cortical amyloid deposition (AV45) [24]).

**Implementation Details** The $S, Z \in \mathbb{R}^{64}$. For the shared part of $q_\psi(s, z|x, \tilde{d}^e)$, we concatenate outputs of feature extractors of $X$ and $\tilde{d}^e$: the feature extractor for $x$ is composed of four Convolution-Batch Normalization-ReLU (CBNR) blocks and four Convolution-Batch Normalization-ReLU-MaxPooling (CBNR-MP) blocks with structure 64 BNR $\to$ 128 CBNR-MP $\to$ 128 CBNR $\to$ 256 CBNR-MP $\to$ 256 CBNR $\to$ 512 CBNR-MP $\to$ 512 CBNR $\to$ 1024 CBNR-MP; the feature extractor of $\tilde{d}^e$ is composed of three Fully Connection-Batch Normalization-ReLU (FC-BNR) blocks with structure 128 $\to$ 256 $\to$ 512. for the part specific to each domain, $\mu_{s,z}(x, d)$ and $\log \sigma_{s,z}(x, d)$ are generated by the sub-network which is composed of 1024 FC-BNR $\to$ 1024 FC-BNR $\to$ $q_{z,s}$ FC-BNR. The $z, s$ can be reparameterized by $\mu_{s,z}(x, d)$ and $\log \sigma_{s,z}(x, d)$ are fed into a sub-network which is composed of $q_{z,s}$ FC-BNR $\to$ 1024 FC-BNR $\to$ $q_{z,s}$ FC-BNR to get rid of the constraint of Gaussian distribution. For the prior model $p_\theta(s, z|\tilde{d}^e)$, it shares the same structure without feature extractor of $x$. For $p_\theta(x|s, z)$, the network is composed of three DeConvolution-Batch Normalization-ReLU (DCBNR) blocks and three Convolution-Batch Normalization-ReLU (CBNR) blocks, followed by a convolutional layer, with structure 256 DCBNR $\to$ 256 CBNR $\to$ 128 DCBNR $\to$ 128 CBNR $\to$ 64 DCBNR $\to$ 64 CBNR $\to$ 48 Conv. For $p_\theta(y|s)$, the network is composed of 256 FC-BNR $\to$ 512 FC-BNR $\to$ 3 FC-BNR. For prior model $p_\theta(s, z|\tilde{d}^e) \mathcal{N}(\mu_{s,z}(\tilde{d}^e), \text{diag}(\sigma_{s,z}^2(\tilde{d}^e)))$ the $\mu_{s,z}(x, \tilde{d}^e)$ and $\log \sigma_{s,z}(x, \tilde{d}^e)$ are parameterized by Multi Perceptron Neural Network (MLP). The decoders $p_\theta(x|s, z)$ are $p_\theta(y|s)$ parameterized by Deconvolutional neural network. For all methods, we train for 200 epochs using SGD with weight decay $2 \times 10^{-4}$ and learning rate 0.01 and is multiplied by 0.2 after every 60 epochs. The batch size is set to 4.

**The $D$ variable in training and test.** The selected attributes include Education Years, Age, Gender (0 denotes male and 1 denotes female), AV45, amyloid$_\beta$, and TAU. We split the data into $m = 2$ training environments and test according to different values of $D$. The Tab. 5 describes the data distribution in terms of the number of samples, the value of $D$ (Age and TAU).

Table 5: Training and test environments (characterized by $c$) in early prediction of AD

|  | Training Env#1 | Training Env#1 | Test |
|---|---|---|---|
|  | Age | | |
| Number of AD | 17 | 17 | 14 |
| Number of MCI | 76 | 83 | 35 |
| Number of NC | 34 | 27 | 14 |
| Average value of $d$ (years): | 68.75 | 72.78 | 81.74 |
|  | TAU | | |
| Number of AD | 11 | 22 | 15 |
| Number of MCI | 75 | 78 | 41 |
| Number of NC | 40 | 27 | 18 |
| Average value of $d$: | 215.34 | 286.69 | 471.72 |

## E.7 Robustness on Security

We consider the DeepFake-related security problem, which targets on detecting small perturbed fake images that can spread fake news. The [57] provides FaceForensics++ dataset from 1000 Youtube videos for training and 1,000 benchmark images from other sources for testing. We split the train data into $m = 2$ environments according to video ID. The considerable result in Tab. 6 verifies potential value on security.

**Implementation Details.** We implement data augmentations, specifically images with 30 angle rotation, with flipping horizontally with $50\%$ probability. We additionally apply random compressing techniques, such as JpegCompression. For inference model, we adopt Efficient-B5 [72], with the detailed network structure as: FC(2048, 2048) $\rightarrow$ BN $\rightarrow$ ReLU $\rightarrow$ FC(2048, 2048) $\rightarrow$ BN $\rightarrow$ ReLU $\rightarrow$ FC(2048, $q_{t=s,z}$). The structure of reparameterization, *i.e.*, $\varphi_{t=s,z}$ is FC($q_{t=s,z}$, 2048) $\rightarrow$ BN $\rightarrow$ ReLU $\rightarrow$ FC(2048, 2048) $\rightarrow$ BN $\rightarrow$ ReLU $\rightarrow$ FC(2048, $q_{t=s,z}$). The network structure for generative model, *i.e.*, $p_\psi(x|s, z)$ is TConv-BN-ReLU($q_{t=s,z}$, 256) $\rightarrow$ TConv-BN-ReLU(256, 128) $\rightarrow$ TConv-BN-ReLU(128, 64)$\rightarrow$ TConv-BN-ReLU(64, 32) $\rightarrow$ TConv-BN-ReLU(32, 32) $\rightarrow$ TConv-BN-ReLU(32, 16) $\rightarrow$ TConv-BN-ReLU(16, 16) $\rightarrow$ Conv-BN-ReLU(16, 3) $\rightarrow$ Sigmoid, followed by cropping the image to the same size $3 \times 224 \times 224$. We set $q_{t=s,z}$ as 1024. We implement SGD as optimizer, with learning rate 0.02, weight decay 0.00005, and run for 9 epochs.

Table 6: Accuracy (%) of robustness on FaceForensics++. Average over three runs.

| ERM | IRM | LaCIM (**Ours**) |
|---|---|---|
| $82.8 \pm 0.99$ | $83.4 \pm 0.59$ | $\mathbf{84.47 \pm 0.90}$ |

## E.8 Network Structure

Table 7: General framework table for our method and baselines on Data $\in$ {CMNIST, NICO, ADNI, DeepFake} Dataset. We denote the dimension of $z$ or $s$ as $\dim_{z,s}$. We list the output dimension (e.g. the channel number) of each module, if it is different from the one in Tab. 8.

| Dataset \ Method | ERM | MMD-AAE | DANN | DIVA | LaCIM |
|---|---|---|---|---|---|
| Data:CMNIST | $\text{Enc}_x^{\text{Data}}$
$\text{FC}(256,\dim_z)$
$\text{Dec}_y^{\text{Data}}$ | $\text{Enc}_x^{\text{Data}}$
$\text{FC-BN-ReLU}(256,256)$
$\text{FC}(256,256)\to z$
$\text{Dec}_y^{\text{Data}};\text{Dec}_x^{\text{Data}}$ | $\text{Enc}_x^{\text{Data}}$
$\text{DANN-CLS}_y^{\text{Data}};\text{DANN-CLS}_y^{\text{Data}}$ | $p_\theta^{\text{Data}}(x|z_d,z_x,z_y)$
$p_{\theta_d}^{\text{Data}}(z_d|d)$
$p_{\theta_y}^{\text{Data}}(z_y|y)$
$q_{\phi_d}^{\text{Data}}(z_d|x)$
$q_{\phi_x}^{\text{Data}}(z_x|x)$
$q_{\phi_y}^{\text{Data}}(z_y|x)$ | $\text{Enc}_x^{\text{Data}}$
$\text{Enc}_z^{\text{Data}}\times m$
$\Phi_{z,s}^{\text{Data}}\times m$
$\text{Dec}_y^{\text{Data}};\text{Dec}_x^{\text{Data}}$ |
| # of Params | 1.12M | 1.23M | 1.1M | 1.69M | 0.92M |
| hyper-Params | lr: 0.1
wd:0.00005 | lr: 0.01
wd: 0.0001 | lr: 0.1
wd: 0.0002 | lr: 0.001
wd: 0.00001 | lr: 0.01
wd: 0.0002 |
| Data : $NICO$ | $\text{Enc}_x^{\text{Data}}$
$\text{FC}(1024,\dim_z)$
$\text{Dec}_y^{\text{Data}}$ | $\text{Enc}_x^{\text{Data}}$
$\text{FC-BN-ReLU}(1024,1024)$
$\text{FC}(1024,1024)\to z$
$\text{Dec}_y^{\text{Data}};\text{Dec}_x^{\text{Data}}$ | $\text{Enc}_x^{\text{Data}}$
$\text{DANN-CLS}_y^{\text{Data}};\text{DANN-CLS}_y^{\text{Data}}$ | $p_\theta^{\text{Data}}(x|z_d,z_x,z_y)$
$p_{\theta_d}^{\text{Data}}(z_d|d)$
$p_{\theta_y}^{\text{Data}}(z_y|y)$
$q_{\phi_d}^{\text{Data}}(z_d|x)$
$q_{\phi_x}^{\text{Data}}(z_x|x)$
$q_{\phi_y}^{\text{Data}}(z_y|x)$ | $\text{Enc}_x^{\text{Data}}$
$\text{Enc}_z^{\text{Data}}\times m$
$\Phi_{z,s}^{\text{Data}}\times m$
$\text{Dec}_y^{\text{Data}};\text{Dec}_x^{\text{Data}}$ |
| # of Params ($m = 8$) | 18.08M | 19.70M | 19.13M | 14.86M | 18.25M |
| # of Params ($m = 14$) | 18.08M | 19.70M | 26.49M | 14.87M | 19.70M |
| hyper-Params | lr: 0.01
wd: 0.0002 | lr: 0.2
wd: 0.0001 | lr: 0.05
wd: 0.0005 | lr: 0.001
wd: 0.0001 | lr: 0.01
wd: 0.0001 |
| Data:ADNI | $\text{Enc}_x^{\text{Data}}$
$\text{FC}(1024,\dim_z)$
$\text{Dec}_y^{\text{Data}}$ | $\text{Enc}_x^{\text{Data}}$
$\text{FC-BN-ReLU}(1024,1024)$
$\text{FC}(1024,1024)\to z$
$\text{Dec}_y^{\text{Data}};\text{Dec}_x^{\text{Data}}$ | $\text{Enc}_x^{\text{Data}}$
$\text{DANN-CLS}_y^{\text{Data}};\text{DANN-CLS}_y^{\text{Data}}$ | $p_\theta^{\text{Data}}(x|z_d,z_x,z_y)$
$p_{\theta_d}^{\text{Data}}(z_d|d)$
$p_{\theta_y}^{\text{Data}}(z_y|y)$
$q_{\phi_d}^{\text{Data}}(z_d|x)$
$q_{\phi_x}^{\text{Data}}(z_x|x)$
$q_{\phi_y}^{\text{Data}}(z_y|x)$ | $\text{Enc}_x^{\text{Data}}$
$\text{Enc}_z^{\text{Data}}\times m$
$\Phi_{z,s}^{\text{Data}}\times m$
$\text{Dec}_y^{\text{Data}};\text{Dec}_x^{\text{Data}}$ |
| # of Params | 28.27M | 36.68M | 30.21M | 33.22M | 37.78M |
| hyper-Params | lr: 0.01
wd: 0.0002 | lr: 0.005
wd: 0.0002 | lr: 0.01
wd: 0.0002 | lr: 0.005
wd: 0.0001 | lr: 0.005
wd: 0.0002 |

Table 8: Network Structure of Modules used in our method and baselines.

| Method | CMNIST | NICO | ADNI |
|---|---|---|---|
| $\mathrm{Enc}_x^{\mathrm{Data}}$ | Conv-BN-ReLU(dim$_{\mathrm{input}}$,64,3,1,1)
MaxPool(2)
Conv-BN-ReLU(64,128,3,1,1)
MaxPool(2)
Conv-BN-ReLU(128,256,3,1,1)
MaxPool(2)
Conv-BN-ReLU(256,256,3,1,1)
AdaptivePool(1)
Flatten() | Conv-BN-ReLU(dim$_{\mathrm{input}}$,128,3,1,1)
Conv-BN-ReLU(128,256,3,2,0)
MaxPool(2)
Conv-BN-ReLU(256,256,3,1,1)
Conv-BN-ReLU(256,512,3,1,1)
MaxPool(2)
Conv-BN-ReLU(512,512,3,1,1)
Conv-BN-ReLU(512,512,3,1,1)
MaxPool(2)
Conv-BN-ReLU(512,512,3,1,1)
Conv-BN-ReLU(512,1024,3,1,1)
AdaptivePool(1)
Flatten() | Conv3d-BN-ReLU(dim$_{\mathrm{input}}$,128,3,1,1)
Conv3d-BN-ReLU(128,256,3,2,0)
MaxPool(2)
Conv3d-BN-ReLU(256,256,3,1,1)
Conv3d-BN-ReLU(256,512,3,1,1)
MaxPool(2)
Conv3d-BN-ReLU(512,512,3,1,1)
Conv3d-BN-ReLU(512,512,3,1,1)
MaxPool(2)
Conv3d-BN-ReLU(512,512,3,1,1)
Conv3d-BN-ReLU(512,1024,3,1,1)
AdaptivePool(1)
Flatten() |
| $\mathrm{Dec}_x^{\mathrm{Data}}$ | UnFlatten()
Upsample(2)
Tconv-BN-ReLU(dim$_{\mathrm{input}}$,128,2,2,0)
Tconv-BN-ReLU(128,64,2,2,0)
Tconv-BN-ReLU(64,32,2,2,0)
Tconv-BN-ReLU(32,16,2,2,0)
Conv(16,3,3,1,1)
Sigmoid()
Cropping(28) | UnFlatten()
Upsample(16)
Tconv-BN-ReLU(dim$_{\mathrm{input}}$,256,2,2,0)
Conv-BN-ReLU(256,256,3,1,1)
Tconv-BN-ReLU(256,128,2,2,0)
Conv-BN-ReLU(128,128,3,1,1)
Tconv-BN-ReLU(128,64,2,2,0)
Conv-BN-ReLU(64,64,3,1,1)
Tconv-BN-ReLU(64,32,2,2,0)
Conv-BN-ReLU(32,32,3,1,1)
Conv(32,3,3,1,1)
Sigmoid() | UnFlatten()
Upsample(6)
Tconv3d-BN-ReLU(dim$_{\mathrm{input}}$,256,2,2,0)
Conv3d-BN-ReLU(256,256,3,1,1)
Tconv3d-BN-ReLU(256,128,2,2,0)
Conv3d-BN-ReLU(128,128,3,1,1)
Tconv3d-BN-ReLU(128,64,2,2,0)
Conv3d-BN-ReLU(64,64,3,1,1)
Tconv3d-BN-ReLU(64,64,2,2,0)
Conv3d-BN-ReLU(64,64,3,1,1)
Conv3d(64,1,3,1,1)
Sigmoid() |
| $\mathrm{Enc}_d^{\mathrm{Data}}$ | FC-BN-ReLU($d$, 128)
FC-BN-ReLU(128, 256) | FC-BN-ReLU($d$, 256)
FC-BN-ReLU(256, 512)
FC-BN-ReLU(512, 512) | FC-BN-ReLU($d$, 256)
FC-BN-ReLU(256, 512)
FC-BN-ReLU(512, 512) |
| $\mathrm{Dec}_y^{\mathrm{Data}}$ | FC-BN-ReLU(dim$_{z,s}$, 512)
FC-BN-ReLU(512, 256)
FC(256,2) | FC-BN-ReLU(dim$_{z,s}$, 512)
FC-BN-ReLU(512, 256)
FC(256,2) | FC-BN-ReLU(dim$_{z,s}$, 512)
FC-BN-ReLU(512, 256)
FC(256,2) |
| $\mathrm{Dec\text{-}CE}_y^{\mathrm{Data}}$ | FC-BN-ReLU(dim$_{z,s}$, 512)
FC-BN-ReLU(512, 256)
FC(256,2) | FC-BN-ReLU(dim$_{z,s}$, 1024)
FC-BN-ReLU(1024, 2048)
FC(2048,2) | FC-BN-ReLU(dim$_{z,s}$, 512)
FC-BN-ReLU(512, 256)
FC(256,2) |
| $\mathrm{DANN\text{-}CLS}_y^{\mathrm{Data}}$ | FC-BN-ReLU(256, 32)
FC-BN-ReLU(32, 2) | FC-BN-ReLU(1024, 2048)
FC-BN-ReLU(2048, 2) | FC-BN-ReLU(1024, 1024)
FC-BN-ReLU(1024, 2) |
| $\Phi_{z,s}^{\mathrm{Data}}$ | FC-ReLU(dim$_{z,s}$, 256)
FC-ReLU(256, dim$_{z,s}$) | FC-ReLU(dim$_{z,s}$, 1024)
FC-ReLU(1024, dim$_{z,s}$) | FC-ReLU(dim$_{z,s}$, 1024)
FC-ReLU(1024, dim$_{z,s}$) |
| $\mathrm{Enc}_{z,s}^{\mathrm{Data}}$ | FC-ReLU(256, 256)
FC-ReLU(256, dim$_{z,s}$) | FC-ReLU(1024, 1024)
FC-ReLU(1024, dim$_{z,s}$) | FC-ReLU(1024, 1024)
FC-ReLU(1024, dim$_{z,s}$) |
| $p_\theta^{\mathrm{Data}}(x|z_d, z_x, z_y)$ | FC-BN-ReLU(1024)
UnFlatten()
Upsample(8)
TConv-BN-ReLU(64,128,5,1,0)
Upsample(24)
TConv-BN-ReLU(128,256,5,1,0)
Conv(256, 256*3,1,1,0) | FC-BN-ReLU(1024)
UnFlatten()
Upsample(16)
TConv-BN-ReLU(64,128,5,1,0)
Upsample(64)
TConv-BN-ReLU(128,256,5,1,0)
Upsample(256)
Conv(256, 3,1,1,0) | FC-BN-ReLU(1024)
UnFlatten()
Upsample(8)
TConv3d-BN-ReLU(16,64,5,1,0)
Conv3d-BN-ReLU(64,128,3,1,1)
Upsample(24)
TConv3d-BN-ReLU(128,128,5,1,0)
Conv3d-BN-ReLU(128,128,3,1,1)
Upsample(48)
Conv3d-BN-ReLU(128,32,3,1,1)
Conv3d(32, 1,1,1,0) |
| $p_{\theta_d}^{\mathrm{Data}}(z_d|d)$
$p_{\theta_y}^{\mathrm{Data}}(z_y|y)$ | FC-BN-ReLU(dim$_{d,y}$, 64)
FC(64,64); FC(64,64) | FC-BN-ReLU(dim$_{d,y}$, 64)
FC(64,64); FC(64,64) | FC-BN-ReLU(dim$_{d,y}$, 64)
FC(64,64); FC(64,64) |
| $q_{\phi_d}^{\mathrm{Data}}(z_d|x)$
$q_{\phi_x}^{\mathrm{Data}}(z_x|x)$
$q_{\phi_y}^{\mathrm{Data}}(z_y|x)$ | Conv-BN-ReLU(3,32,5,1,0)
MaxPool(2)
Conv-BN-ReLU(32,64,5,1,0)
MaxPool(2)
Flatten()
FC(1024, 64); FC(1024, 64) Data | Conv-BN-ReLU(3,32,3,2,1)
MaxPool(2)
Conv-BN-ReLU(32,64,3,2,1)
MaxPool(2)
Conv-BN-ReLU(64,64,3,2,1)
MaxPool(2)
Flatten()
FC(1024, 64); FC(1024, 64) Data | Conv3d-BN-ReLU(1,64,3,2,1)
Conv3d-BN-ReLU(64,128,3,1,1)
MaxPool(3)
Conv3d-BN-ReLU(128,256,3,1,1)
Conv3d-BN-ReLU(256,256,3,1,1)
MaxPool(2)
Conv3d-BN-ReLU(256,256,3,1,1)
Conv3d-BN-ReLU(256,128,3,1,1)
MaxPool(2)
Flatten()
FC(1024, 64); FC(1024, 64) Data |