# OpenReview forum: "Recovering Latent Causal Factor for Generalization to Distributional Shifts"
_NeurIPS.cc/2021/Conference — NeurIPS 2021 Poster_

### Official Review · Reviewer_wuYX · 2021-07-16

**Rating:** 6
**Confidence:** 2

**Summary:**

In this paper, the authors propose a method to estimate the invariant causal mechanism. By modeling the process as the causal graph in Fig.1 (c), the authors first provide the condition when disentangling causal factors is feasible. Based on the theoretical result, they present the method to infer causal factor and non-causal factor with X based on VAE.

**Limitations And Societal Impact:**

Yes

**Main Review:**

The authors address an important problem in this paper. The method is with solid theoretical guarantee. The experiments validate the effectiveness of the proposed method. I like this paper. Hence I give a positive score. I possibly change my score according to the comments of other reviewers.

I have two suggestions:

1. The writing could be improved. The theoretical part is quite hard to follow. There are many notations without clear illustration, e.g. k_t,q_t, \theta. Does k_t imply the sample number and q_t imply the domain number?

2. I suggest one additional citation, where the authors propose a method that also takes latent causal factors into account when there are distributional shifts.

IJCAI 2017 Causal discovery from nonstationary/heterogeneous data: Skeleton estimation and orientation determination.

**Time Spent Reviewing:**

4

---

> ### Author Response · Authors · 2021-08-09
> **Response to Reviewer wuYX**
>
> Thanks for your acknowledgment and efforts in reviewing our manuscript.
>
> **About Notations.** The $q_t$ denotes the dimension of $t=s,z$ while $k_t$ denotes the number of natural parameters for each dimension of $t=s,z$. We will make the corresponding part in section 4.2 clearer and easier to follow in the updated version.
>
> **About reference.** Thanks for your suggestion! We will cite this paper in the updated version.

---

> > ### Comment · Reviewer_wuYX · 2021-09-03
> > **Thank you**
> >
> > Dear authors:
> >
> >   Thank you for your response. I am sorry that I am not quite familiar with the related topic. Hence it is hard for me to give some constructive suggestions. As far as I can see, the proposed method is solid. I thus keep my positive score.
> >
> > Best Regards

---

### Official Review · Reviewer_ss3N · 2021-07-16

**Rating:** 7
**Confidence:** 4

**Summary:**

This work proposes an identifiable causal model with latent variables to model spurious correlations between labels and irrelevant context (such as background or light setting) in order to construct a classifier which is robust to certain distribution shifts. The authors show under which conditions the irrelevant context and important features can be disentangled from only input observations and labels. They propose a VAE-like approach to learn the model which they use to come up with an invariant classifier. The identifiability of the model is assessed on synthetic data and the robustness to dataset shift is evaluated on multiple real world data sets, drawing a favorable picture of their contribution.

**Limitations And Societal Impact:**

Yes.

**Main Review:**

Originality:

I have been following the literature on nonlinear ICA (from which the identifiability result is inspired) and it is the first time I see these ideas directly applied to out-of-distribution generalization. As far as I know, all important related works have been cited.

Quality:

I believe this paper presents interesting novel ideas with convincing experiments and I enjoyed reading it. However, some algorithmic decisions are not justified and the paper lacks clarity. The focus on identifiability and why it matters for dataset shift generalization is not well explained. I make some of these concerns more specific in what follows, with no specific order:

Some assumptions appear to be implicit. For instance, the fact that q_s + q_z <= q_x. Otherwise, f_x cannot be bijective. Moreover, saying that f_x and f_y are bijective without specifying the domain and codomain of the functions is imprecise. For instance, if q_s + q_z < dim(x) and the codomain is R^dim(x), then f_x cannot be bijective. So what is the codomain?

L172: The two points are said to “...ensure the robustness of such a two-step invariant prediction, ...”. However, the aforementioned two-step procedure is only very vaguely defined up to this point in the paper, thus it was hard for me to accept that we actually want these desiderata. Maybe presenting the procedure before would help?

That being said, even after understanding the procedure, I felt like I had to think a lot to understand why these two points (at line 172) are important to have (and I am still not sure). Could you provide a clear explanation of what would be the consequence of any of these desiderata not holding?

Definition 4.3:
Desiderata 1 (line 172) does not require M_s and M_z to be permutation-scaling matrices, right? In principle, only having “block” identifiability would be enough, no? Do you believe one could come up with weaker assumptions? If not, please mention that having permutation matrices is “extra” and not really required.

Is it possible that $[f\_x^{-1}]\_\mathcal{S}$ maps some points of $\mathcal{X}$ outside $\mathcal{S}$? Similarly for $\mathcal{Z}$? Since the definition supposes only $\mathcal{X} \supset f\_x(\mathcal{S}\times\mathcal{Z})$, it is possible.

This definition assumes implicitly that $f\_x$ is invertible but this is not mentioned until the following theorem. This assumption should be mentionned earlier.

Theorem 4.5 introduces a notion of “asymptotic identifiability”, however, I am not sure I understand its interpretation. It would be helpful to give some intuition for its meaning. That would help me understand its value. Also, I suspect the dimensionality of the sufficient statistic is allowed to grow, however, the definition of $\mathcal{P}\_\text{exp}$ does not allow for this. Moreover, in order for the assumption of Theorem 4.4 to hold, the number of environments $m$ needs to grow as well, right? This should be clearer. I believe an interesting consequence of this Theorem is that even if the assumptions of Theorem 4.4 are not satisfied for some $p \in \mathcal{P}_\text{exp}$, it can still be approximated by an identifiable model. Is that correct (assuming the exponential model is in the appropriate Sobolev space)?

Section 4.3.2 presents the invariant prediction procedure which, as far as I know, is not standard (which is not necessarily a bad thing). I had a hard time interpreting the meaning of it and understanding why it makes sense. I believe the author should add a few lines motivating this approach. One possible interpretation for the max in (6) is that it corresponds to a maximum a posteriori (MAP) estimation of z and s in a counterfactual model in which p(s,z|c) is replaced by p(s)p(z) (thus without correlation). But even with this interpretation, I am not completely satisfied. Of course the procedure is invariant to the environment by design, but is it optimal in some sense?

Section 5: I found the experiments sufficient and convincing. Identifiability is confirmed on synthetic experiments and out-of-distribution generalization is measured against multiple relevant baselines on three datasets. Since I do not follow closely the literature on OOD generalization, I cannot confirm that no important baseline is missing.

I would like to see more than one latent interpolation for the dataset CMNIST. Do they all look as good as the one presented here?

Clarity:

L96: The definition of [f]_A is imprecise, usually the word “restricted” refers to the domain being restricted. At that point I did not understand it and even after reading Definition 4.3, I had to infer it from context. Please clarify.

L139: This sentence is contradictory: “In our scenario, we do not require D to be observed; rather, we only need the domain index d_e...”. This means D is observed, no?

Equation (1): I believe some notation is unnecessarily hard to follow. For instance, the $|c$ symbol should be removed from the two left factors in the first line of (1). Also, having a superscript t on T and \Gamma was a bit confusing. It led me to think that these things could depend on the actual *value* of z or s. Now I understand that this is just to allow different exponential families for Z and S. A solution would be to remove this superscript and replace it with a footnote mentioning that everything works even if we have different sufficient statistics for Z and S.

Some notation is implicitly defined. For instance q_s and q_z being the dimensionality of S and Z, respectively. It would help the reader to mention it explicitly in the text. Similarly for k_s and k_q, which are the dimensionality of the sufficient statistic of each s_i and z_i, respectively. What is the dimensionality of x? Again, implicitly defined on L228.

Theorem 4.4: The fifth assumption was hard to parse, even if I have some experience  with these kinds of assumptions.

L237: “... can hold unless the space of \Gamma belong to a zero-(Lebesgue) measure set.” This statement is not clear. It seems \Gamma can always take only finitely many values and thus is contained in a measure zero set, even if the assumptions are satisfied. Am I misunderstanding something?

The writing of Section 4.3.1 should be completely revised. I had a very hard time actually understanding many steps of the derivations and even just what the actual inference model (q) versus the generative model (p) are. Some conditional density functions are introduced without definitions which might explain why I could not follow. Also, is there a superscript e missing on q_\psi(y | s) at line 271.

In Section 5: Please mention explicitly which representations are used to compute the MCC. I suppose they are the ground-truth (z,s) and the one obtained by solving (6)?

Significance:

I believe using ideas from nonlinear ICA to come up with identifiable latent models for out-of-distribution generalization is an interesting direction and could inspire more work.







**Time Spent Reviewing:**

6.5

---

> ### Author Response · Authors · 2021-08-09
> **Response to Reviewer ss3N**
>
> Thanks for your acknowledgment and valuable suggestions regarding our work. The following list our addresses to your concerns.
>
> **About the bijectivity assumption.** We assumed the surjectivity since it has been empirically verified that the (variational) auto-encoder can extract representation that recover $x$ without loss of information. Besides, we can relax this to only injectivity assumption, by restricting on $\{(x,y): x = f_x(s,z),y=f_y(s) \ \text{for some $s \in \mathcal{S}$ and $z \in \mathcal{Z}$}\}$ during the proof of our main theorem (*i.e.*, changing the conclusion in line 205 to "for any $(x,y) \in f_x(\mathcal{S}\times \mathcal{Z}) \times f_y(\mathcal{S})$").
>
>
> **About two desiderata in lines 172-173.** If the first desideratum does not hold, then the learned $s$ can mix the information of contextual information $z$. This mixture is due to a spurious correlation between $s$ and $z$. This learned spurious correlation can result in a performance drop on unseen domains when this correlation does not hold. If the second desiderata not hold, then the ground-truth predicting mechanism (*i.e.*, $p(y|s^\star)$ for $x \gets f_x(s^\star,z^\star)$) cannot be recovered. Since this ground-truth predicting mechanism can apply to all domains in $\mathcal{E}$, it is sufficient to ensure robustness across domains. Therefore, if this desideratum is not satisfied, then we may lose robustness. We will put the inference procedure before to make them clearer and better motivated in the updated version.
>
> **About permutation matrices $M_s$ and $M_z$.** First, we need $M_s$ and $M_z$ to be invertible to ensure the disentanglement of $s$ and $z$, as long as the extreme case (*i.e.*, the $s$ is represented by $z$) does not hold. Specifically, since we have $<\mathbf{\tilde{T}}^s([f_x^{-1}(x)]_{\mathcal{S}}) = M_s\mathbf{T}^s([f_x^{-1}(x)]_{\mathcal{S}}) + b_s>$, then $<\mathbf{\tilde{T}}^s([f_x^{-1}(x)]_{\mathcal{S}})$ would not depend on $z$, *i.e.*, $\mathbf{\tilde{T}}^s([f_x^{-1}(x)]_{\mathcal{S}}) = k([f_x^{-1}(x)]_{\mathcal{S}}, [f_x^{-1}]_{\mathcal{Z}}(x))>$. Otherwise,  it would be $<\mathbf{T}^s([f_x^{-1}(x)]_{\mathcal{S}}) = M_s^{-1} \left( k([f_x^{-1}(x)]_{\mathcal{S}}, [f_x^{-1}(x)]_{\mathcal{Z}}) - b_s \right)>$, which contradicts to the non-extreme case that makes the isolation of $s$ impossible. We make further assumptions to achieve permutation to ensure that each component of causal factors can be recovered, in order for better explainability.
>
> **About $[f_x^{-1}(x)]_{\mathcal{S}}$.** We will put the invertibility/bijectivity condition of $f_x$ before the Def. 4.3. Generally,
> $[f^{-1}(x)]_{\mathcal{A}}:= \[ \[f^{-1}(x)\](i_1), \[f^{-1}(x)\](i_1),...,\[f^{-1}(x)\](i_k) \]$ where $i_1,...,i_k$ are indexes such that $\mathcal{A} \subset \mathcal{B}(i_1) \times \mathcal{B}(i_2) \times ... \times  \mathcal{B}(i_k)$ where $\mathcal{B}:=\mathbb{R}^p$ with $p$ denoting the dimension of $[f^{-1}(x)]$. We will modify it in the updated version.
>
> **About asymptotic identifiability.** The asymptotic identifiability generalizes the identifiability result in $<\mathcal{P}_{\exp}>$
> to a more general family (*i.e.*, Sobolev space), due to the fact that $<\mathcal{P}_{\exp}>$ is dense in Sobolev space. That is, for $<p \notin \mathcal{P}_{\exp}>$, it can be approximated by an identifiable sequence in $<\mathcal{P}_{\exp}>$. The $k_t,q_t,m$ are allowed to increase. We will modify it in the updated version.
>
> **About the invariant prediction procedure.** This procedure is motivated by not only the invariance property, but more importantly, but also by our identifiability guarantees that imposed as desiderata in lines 172-173. Specifically, we have proved in theorem 4.4 and 4.5 that the estimated $s$ from $f_x^{-1}(x)$ will not mix $z$ (Eq. (2)), this motivated us to maximize $p_{f_x}(x|s,z)$ to estimate $s$. Further, we have proved in Eq. (3) that the $p_{f_y}(y|s^\star)$ can be identified for $x\gets f_x(s^\star,z^\star)$ (Specifically, note that if $<p_{\tilde{f}_y}(y|\tilde{f}_x^{-1})>$ denotes our estimated predicting mechanism, then it equals to $<p_{f_y}(y|s^\star)>$.). This motivated us to implement $<p_{f_y}(y|s)>$ for prediction. Simply speaking, the ground-truth predicting mechanism $<p_{f_y}(y|[f_x^{-1}(x)]_{\mathcal{S}})>$ is composed of two steps: (i) $<[f_x^{-1}(x)]_{\mathcal{S}}>$ to estimate $s$, which does not mix $z$ guaranteed by Eq. (2) and hence motivated to maximize $<p_{f_x}(x|s,z)>$ to estimate $s$; $<p_{f_y}(y|s)>$ for prediction, which motivated our second step.
>
> **More visualization results.** Yes. We have visualized other dimensions of $S$ and $Z$ and will add these visualizations in the updated version. We observe that the different dimensions of $s$ can learn different differentiating semantic information. For example, the first dimension of $S$ can learn to remove the left part of "$0$" to "$7$" as interpolated; while the second dimension can learn to add the dash in the hand-writing "$7$". For the dimension of $Z$, it learned other non-differentiating factors such as width, color.
>
> **About the contradictory sentence of domain variable $D$.** The $D$ denotes the hidden attributes that characterize each domain. But in the unpooled dataset setting, it is natural to know which domain each sample belongs to, *i.e.* the domain index. In example 4.1 (line 154-159), the $D$ can denote the attributes of each sampler (include but not limited to age, weight, working status), which is unobserved. However, we can access the sampler ID as the domain index.
>
>
> **About subscript of the sufficient statistics.** Thank you for your suggestion! We will adopt it to make it simpler and easier to read.
>
> **About the parsing of assumption (5) in theorem 4.4.** It should be rewritten as: "The $L:=[P^{e_1}(C)^\mathsf{T},...,P^{e_m}(C)^\mathsf{T}]^\mathsf{T} \in \mathbb{R}^{m \times R}$ and $<\big[[\mathbf{\Gamma}^{t=s,z}_{c_2,d^{e_1}} - \mathbf{\Gamma}^{t=s,z}_{c_1,d^{e_1}}]^\mathsf{T},...,[\mathbf{\Gamma}^{t=s,z}_{c_{R},d^{e_m}} - \mathbf{\Gamma}^{t=s,z}_{c_1,d^{e_1}}]^\mathsf{T}\big]^\mathsf{T} \in \mathbb{R}^{(R\times m-1) \times (q_t \times k_t)}>$. Both matrices have full column rank." We will modify it in the updated version.
>
> **About the space of $\Gamma$.** The space of $\Gamma$ denotes the $\Gamma(\mathcal{C} \times \mathcal{D})$ where the cardinality of $\mathcal{D}$ can be infinite and $<\{d^e\}_{e\in \mathcal{E}}>$ are only some finite points of $\mathcal{D}$. This means, for any $m$ different values of $D$, it generically holds that the corresponding matrices of $\mathbf{\Gamma}$ have full column rank.
>
> **About section 4.3.1.** This part is about the generative model, which is based on variational auto-encoder (VAE). We have made a brief introduction to make it self-contained, due to the space limit. Other details of VAE can be referred to the paper of VAE [1] In the following, we will introduce more details (and append it in appendix) for open discussion:
>
> The inference model $q_\psi$ is introduced variational distribution that approximates the generative model $p_\theta$ in posterior. Traditionally, one chooses to optimizes $\theta$ via MLE: $\max_\theta E_p[\log{p_\theta(x,y)}]$. Since the dimension of $x,y$ can be high-dimensional, such a optimization may be intractable. To resolve this problem, the VAE introduced a variational distribution $q_\psi$ (in line 267) that mimics the posterior behavior of $p_\theta$; and formulated the evidence lower bound (ELBO) for optimization. As both $q_\psi$ and $p_\theta$ are parameterized probability function that are defined over the same variable set of the ground-truth model $p$ (*i.e.*, the node set in LaCIM in Def. 4.1: $\{C,S,Z,X,Y\}$), any conditional density functions of $q_\psi$ and $p_\theta$ are well-defined. The goal of $q_\psi$ and $p_\theta$ are learning the behavior of $p$. Therefore, $q_\psi(y|s)$ and $p_\theta(y|s)$ are invariant and hence do not depend on $e$ since $p(y|s)$ does not depend on $e$.
>
> **About input of MCC.** Yes. The input of MCC is the ground-truth $Z$ (and $S$) and the one obtained in the inference stage (*i.e.*, the solution of Eq. (6)).
>
> ## **References.**
>
> [1] Kingma, Diederik P., and Max Welling. "Auto-Encoding Variational Bayes." stat 1050 (2014): 1.

---

> > ### Comment · Reviewer_ss3N · 2021-08-31
> > **Response**
> >
> > I thank the authors for their response.
> >
> > I think I understood what is the inference model used. If I understood correctly, there is no "network" representing $q^e(s,z|x,y)$. The only pieces of the inference model are $q^e(s,z | x)$ and $q^e(y|x)$, right? And $p^e(s,z|x,y)$ is approximated by $\frac{q^e(s,z|x)p(y|s)}{q^e(y|x)}$, correct? I did not understood that at the time I wrote my review. It might be easier to understand if the expression $q^e(s,z|x,y)$ is never introduced. The authors could simply start by introducing the conditionals of $q$ which are actually modeled and explain how they are used to approximate $p^e(s,z|x,y)$ (i.e. via $\frac{q^e(s,z|x)p(y|s)}{q^e(y|x)}$).
> >
> > Assuming the authors improve the clarity of their paper based on the suggestions the other reviewers and I have brought up, I will upgrade my score from 6 to 7. I think this is a good contribution.

---

> > > ### Author Response · Authors · 2021-09-01
> > > **Response**
> > >
> > > Thank you for your acknowledgment and for upgrading the score! We will correct the grammar mistakes and explain in a more straightforward way to make the paper easier to read.
> > >
> > > Yes, we only explicitly parameterize $q^e(s,z|x)$ and $q^e(y|x)$ as inference models. The $p^e(s,z|x,y)$ (and $q^e(s,z|x,y)$) are approximated by $\frac{q^e(s,z|x)p(y|s)}{q^e(y|x)}$, with $-\log{q^e(y|x)}$ modeled as cross-entropy loss.

---

### Official Review · Reviewer_1QgY · 2021-07-17

**Rating:** 6
**Confidence:** 4

**Summary:**

This paper introduces the "Latent Causal Invariance Model" (LaCIM), a structural variational autoencoder-like approach for learning to isolate causal factors from "spurious" ones in prediction to improve generalization to new environments. The authors assume a latent generative structure for the data, with separate latent constructs for causal and contextual factors. Under strong parametric assumptions and requirements on the number of source environments the authors prove identifiability of the causal factors. Experiments on imaging datasets (2 benchmarks and a small medical dataset) show LaCIM outperforms some existing domain adaptation baselines as well as Invariant Risk Minimization (IRM). Further, qualitative assessment of images generated by ablating the causal and contextual factors show disentanglement of these factors.

**Limitations And Societal Impact:**

I think the authors could put some more thought into possible impacts of applying LaCIM to more realistic imaging tasks. For example, in facial recognition tasks related to, e.g., law enforcement the distinction between "causal" vs "contextual" factors could have ethical implications or could imply limitations on how LaCIM models are interpreted.

**Main Review:**

To the best of my knowledge, the proposed LaCIM approach is a novel and interesting approach for avoiding "spurious correlations" to improve model generalizability to new contexts. The theoretical results are quite daunting to parse, in part because the methodological ideas build off of both literature on additive noise models (ANM) in causal discovery and independent component analysis (ICA). The authors define a two step process for learning: first learning the generative model (effectively a VAE), then computing the latent representation that maximizes the likelihood for each example and fitting a classifier on top of this. The approach is reasonable, and the experiments show its potential.

I did have a number of minor concerns. The assumed causal structure makes some questionable decisions, there are untestable assumptions regarding the quality of the input data, and the experiments are designed to satisfy the assumptions of LaCIM (meaning comparisons to other methods are likely over optimistic).

Strengths:
- The approach is novel, synthesizing ideas from ANM and ICA to new effect
- The authors prove identifiability of their model. These results are hidden behind rather dense theory, though, so I was unable to check these results.
- Promising experimental evaluation: both quantitative comparisons to other approaches as well as qualitative analysis of disentanglement of factors.

Weaknesses:
- Under the causal model assumed by LaCIM, X and Y are generated concurrently from the contextual factors Z and causal factors S. The authors note this as a strength or unexplored direction in Remark 1 (Ln 148), but for many examples it doesn't seem to make as much sense as assuming X generates or is generated by Y. For example, in Remark 1 the authors talk about disease status (Y) and ultrasound tests (X), but surely whether or not a patient has the disease will effect the results observed in the ultrasound! In most classification tasks, even in imaging, the label is the concept of interest while the observed data (i.e., the image) is a realization of that label. In the qualitative analysis (Line 349), fixing S kept the class label fixed (which the authors state as evidence that LaCIM has learned the "causal" factors), but is it not inefficient to have this redundancy in the role of Y and S? Further, what value is there in having D and C be separate? Isn't it sufficient to just have D in Fig 1c? No additional conditional independence value is added.
- The identifiablity results in Section 4.2 are very interesting, but are at times quite hard to follow. This is in part because of a general lack of clarity in this section (but in general the paper would benefit from one pass to improve writing). The authors justify many of the assumptions in Section 4.2 by pointing to other works that make these assumptions, but readers not familiar with these other works will not understand the significance or consequences of these assumptions. I think the authors assume readers will be familiar with, e.g., assumptions and techniques from ICA.
- The authors discuss the diversity of environments. This is very important and I'm glad to see this! Condition i) (on Line 234) is untestable, right? It depends on R, the cardinality of the latent confounding construct... Does this mean, in general, a user will have no idea if they have enough data or high enough quality data to ensure the correctness of LaCIM?
- In the experiments, the authors explicitly set up different environments to try to follow the generative structure of LaCIM. But this structure does not match up with, e.g., the way the original colored MNIST dataset was generated and thus favors LaCIM. Could LaCIM be applied to the original MNIST dataset? Does IRM (which was the paper which developed the original colored MNIST datasets) outperform LaCIM then? This would speak to sensitivity of results to the causal structural assumptions which would be important to know.
- In the simulated experiment, I think it would be more relevant to examine how the number of domains and the diversity of the data affect the identifiability. Currently, the simulated experiment looks at 3 vs 5 environments, but according to diversity condition i), isn't the relevant factor how the number of environments relates to the cardinality of the latent factor C?
- Minor question: Regarding the learning procedure, won't there be a posterior over S and Z? Couldn't predictions be made by marginalizing over this posterior, rather than optimizing to compute an estimate of s*,z* and then predict $P(Y | s^*)$?



### UPDATES:
Thanks to the authors for their response. I think the greatest improvements to the paper can be made by increasing the clarity in Section 4. I think elements from the discussion of the identifiability and diversity conditions in the authors' response to my comments should be included in the main paper. I believe this will really help readers better understand the method.

I would like to make one comment regrading the experimental methodology: In their response, the authors state that "our method can outperform all compared methods on other benchmarks that satisfy our causal structure." It should be noted that this is expected in order to demonstrate the empirical soundness and validity of the method. For a comprehensive evaluation of the utility of the method, including its possible failure cases, it should also be applied in realistic settings which *may not satisfy the untestable assumptions of the method*. The authors assume a *latent* causal structure, and, in all experiments in the main paper, modify the datasets to satisfy this assumption. But in a real world practical application, a user will not know the latent causal structure. The authors pointed to experiments on the original colored MNIST dataset in the appendix. However, the authors modified the proposed LaCIM method (i.e., used LaCIM-REx) to fit the generative structure of the original colored MNIST dataset. In my view, this defeats the purpose of this experiment, because what we want to see is how the unmodified LaCIM method behaves on the original colored MNIST dataset. For example, from a user perspective, there is no way to differentiate the data generating process (DGP) of the original colored MNIST dataset from the DGP of the modified colored MNIST dataset used by the authors. Thus, in a real use case, the user would be unable to modify the LaCIM methodology as the authors did in the supplemental experiment. Put another way, users should be able to apply the method to problems as they appear (consider, e.g., the [WILDS](https://wilds.stanford.edu/) distribution shift benchmark tasks), or be able to determine that the method is not applicable to their particular problem/application.

Thus, while I remain positive about the work, I think the authors should more clearly investigate and discuss possible limitations of the method for this paper to be a strong and comprehensive . For this reason, I am maintaining my score.

**Time Spent Reviewing:**

2.5

---

> ### Author Response · Authors · 2021-08-09
> **Response to Reviewer 1QgY**
>
> Thanks for the acknowledgment and valuable comments regarding our work. We will address your concerns in the following.
>
> **About concurrent generation of $X$ and $Y$.** We did *not* claim that this concurrent assumption is better than $X \to Y$ or $Y \to X$. We only claimed that this concurrent generating assumption has not been explored in the previous literature; while more importantly, this assumption can be traced back to causal inference and it can hold in many machine learning tasks.
>
> First, this concurrent generation is due to the confounder between $X$ and $Y$ and has been studied a lot in causal inference [2,4]. Conceptually, this assumption holds when the human label $Y$ is *not* based on the input $X$, but on the factors (denoted as $V=(S, Z)$) that also generate the $X$. These factors $V$ can refer to latent topics in topic models [1] where $X, Y$ respectively denote the document and the sentiment/evaluation; medical attributes in disease diagnosis where $X, Y$ respectively denote the ultrasound images and clinician's/examinator's label; ``intention" of which character to write in optical character recognition where $X, Y$ respectively denote the character and the label [2]; object shape, contour, texture in image classification where $X$ ($Y$) denotes the image produced (label recorded) by the photographer; unobserved factors that cause the attributes $A=\{X, Y\}$ where $Y$ is the attribute of interest.
>
> Due to its importance and lack of study in the literature, we investigated this concurrent scenario in this work. Admittedly, this assumption cannot apply to all scenarios (causal scenario $X \to Y$ and anti-causal scenario $Y\to X$). However, as this concurrent generation is conceptually related to these scenarios ($Y \to X$ [3], $X \to Y$ [5]), it would be of great interest for us to extend our method to these scenarios in the future.
>
>
> **About redundancy of $S$ and $Y$.** The $S$ to $Y$ is not deterministic, since the $Y$ denotes the noisy human label based on observation of ground-truth semantic features $S$. Taking the ultrasound’s example in remark 1, the clinicians may make mistakes in predicting the disease status based on observed lesion-related attributes while implementing the examination.
>
> **About the separated definition of $D$ and $C$.** As claimed in Def. 4.1, the value of $D$ is domain-level, that is to say, all samples in each domain share the same value of $D$; while the value of $C$ is sample-level. Without the sample-level variable $C$, the $S$ and $Z$ would be independent in each domain, while this cannot describe the phenomena that the $S$ and $Z$ are correlated in a single domain. For example, the dog can be more correlated with the grass than the snow in a collected dataset.
>
> **About the identifiability assumptions.** Thank you for your suggestions! We will append a more detailed introduction to the assumptions for general readers in the updated version. We will first give some explanations about these assumptions for open discussions:
>
> The assumptions (i)-(iii) are easy to satisfy. Specifically, for assumption (i), the characteristics functions of $\varepsilon_x, \varepsilon_y$ can be almost everywhere non-zero for most continuous variables, such as Gaussian, exponential, beta, gamma. This assumption can ensure the identifiability of $p(f^{-1}(x))$, as shown in Eq. (22) to Eq. (23) in appendix. For assumption (ii), the bijectivity condition can be satisfied since it has been validated that the (variational) auto-encoder can extract meaningful representations from $x$. For the bijectivity condition of y, we first assumed it in theorem 4.4 and then relaxed into categorical variables in theorem 4.5. The assumption (iii) can be uniformly satisfied for all distributions in the strongly exponential family. This assumption can ensure the invertibility of the matrices $M_s$ and $M_z$. The containment of an open set in assumption 4 can ensure the identifiability of $p^e(x,y|c)$; the diversity condition in assumption 5 can result in the invertibility of the $M_z$ and $M_s$ in line 724. We will make them clearer in the updated version.
>
> **About the *diversity condition*.** Yes. This condition is untestable since we do not know the cardinality of $C$. Practically, a larger number of environments can achieve better identification and OOD performance. Theoretically, as long as $m$ is large enough such that $m \geq R$, the *diversity condition* generically hold (*i.e.*, as long as $\mathcal{D}$ (or $\mathcal{C} \times \mathcal{D}$ does not belong to any sets with Lebesgue measure 0, as shown in theorem B.6.). Empirically, we have shown that the OOD performance can improve as $m$ increases ($m=14$ v.s. $m=8$ on NICO), because better identification of latent variables ($m=5$ v.s $m=3$ v.s pool-LaCIM ($m=1$)).
>
>
> **About the problem-setting of Colored-MNIST.**  We have implemented our method on the original colored MNIST setting and achieved comparable results than IRM, as shown in Tab.2 in the appendix. We did not put this result in the main text since the original colored-MNIST setting ($X \to Y$) does not match the concurrent assumption in our LaCIM (*i.e.*, $X, Y$ are generated concurrently). The promising result on $X \to Y$ may due to the accurate inference of $S$ from $X$ due to the powerful vision system of humans [5]. Hence, it is reasonable to view it as a $S \to Y$ process. Thus, modeling by our LaCIM may still bring some benefits. As a kind reminder, our method can outperform all compared methods (including IRM) on other benchmarks that satisfy our causal structure in the experimental part.
>
>
> **About the simulation experiment.** We have implemented the experiments when $R=1$ (the same with the generating process in line 940-947 in the appendix, except that the $c=d^e$). Since the $k_s=k_z=q_s=q_z=2$, thus $m=5$ is sufficient to satisfy the diversity condition. The $m=5$ v.s $m=3$ v.s pool-LaCIM is 0.74 v.s 0.67 v.s 0.34 on $Z$ and 0.84 v.s 0.83 v.s 0.71 on $S$: the $m=5$ outperforms others. In addition, note that even $m=3$ does not satisfy the diversity condition, it performs better than pool-LaCIM ($m=1$). Such a result implied that even if the diversity condition is not satisfied, more environments can also bring benefits in identifiability.
>
>
> Besides, in our simulation experiment, we observed that even the $C$ is a continuous variable (which means the cardinality is infinity), the experimental results (Tab.1) showed that can also identify the $S$ and $Z$ successfully. We will study the theoretical result in the future.
>
> **About posterior inference in the testing stage.** We did not implement the posterior distribution (*i.e.*, maximizing $q^e(s,z|x)$ over $\mathcal{S} \times \mathcal{Z}$) since it varies across environments ( $q^e(s,z|x)=q^e(s,z)q(x|s,z)/q^e(x)$). This means the learned posterior can inherit the bias from training data, which may not generalize on the test data. In the contrary, we implement $p(x|s,z)$ to estimate $s,z$, since the generative model $p(x|s,z)$ is invariant across domains,  as shown in Def. 4.1 and Prop. 4.2.
>
> ## **References.**
>
> [1] Blei, David M., and Jon D. McAuliffe. "Supervised topic models." Proceedings of the 20th International Conference on Neural Information Processing Systems. 2007.
>
> [2] Peters, Jonas, Dominik Janzing, and Bernhard Schölkopf. Elements of causal inference: foundations and learning algorithms. The MIT Press, 2017.
>
> [3] Schölkopf, Bernhard, et al. "On causal and anticausal learning." Proceedings of the 29th International Coference on International Conference on Machine Learning. 2012.
>
> [4] Janzing, Dominik, et al. "Identifying confounders using additive noise models." Proceedings of the Twenty-Fifth Conference on Uncertainty in Artificial Intelligence. 2009.
>
> [5] Biederman, Irving. "Recognition-by-components: a theory of human image understanding." Psychological review 94.2 (1987): 115.

---

> ### Author Response · Authors · 2021-09-10
> **About Real-world evaluation.**
>
> Thanks for the suggestions and efforts in reviewing our paper. We will accordingly include some discussions in our rebuttal as more explanations for better clarification in section 4.
>
> Please allow me to say something about the latent causal structure in real-world applications. In many cases, the causal structure can be (partially) known as the prior knowledge for the problem and the data. As an example of the medical diagnosis in remark 1, whether $Y \to X$ or $X \to Y$ or the $Y \leftrightarrow X$ (i.e., concurrent generation) is determined by the definition of $Y$. If labels are recorded by the clinicians that implement the ultrasound test, then it should be that $X \leftrightarrow Y$. It is also noted that previous works like IRM [1] and [2] all pre-assume the causal structure, whether it can be applied is determined by the problem scenario.
>
> ## **References.**
>
> [1] Arjovsky, Martin, et al. "Invariant risk minimization." arXiv preprint arXiv:1907.02893 (2019).
>
> [2] Ilse, Maximilian, Jakub M. Tomczak, and Patrick Forré. "Designing data augmentation for simulating interventions." arXiv e-prints (2020): arXiv-2005.

---

### Official Review · Reviewer_nVGX · 2021-07-20

**Rating:** 6
**Confidence:** 4

**Summary:**

To deal with the issue of distributional shifts between training and testing domains, the authors propose Latent Causal Invariance Models (LaCIM) consisting of both causal latent factors and non-causal latent factors and the extent of the correlations between them is governed by a domain variable. They theoretically show the identifiability of the causal latent factors and the ground-truth predicting mechanism in the proposed LaCIM. Based on the identifiability, they learn the model by reformulating VAE and then verify it on various real-world data.

**Limitations And Societal Impact:**

The authors did not include any societal impact in the paper.

**Main Review:**

Overall I like the idea and technically the paper makes sense. Here are some of my concerns.

My main concern is that how practical the proposed LaCIM is. If I understand correctly, from Definition 4.1 we know that for any $e$, we have $S⫫Z | C$, that is, $S⫫Z | C, D$. Also, from the assumption over the prior $p_{\boldsymbol{T}, \boldsymbol{\Gamma}}$, we see that $S_i ⫫ S_j | C, D$ for any $i \neq j$ and $Z_i ⫫ Z_j | C, D$ for any $i \neq j$. These conditional independence (given $C$ and $D$) assumptions play a key role in proving the identifiability. Conversely, if the latent variables $S$ and $Z$ do not satisfy these conditional independence assumptions, the identifiability would not hold true and the proposed method would fail to identify the causal factors. Am I right? Actually, in many real-world scenarios, these assumptions do not hold. For example, when some part of Z is affected by $Y$ [1-5], or $Z$ is directly affected by $S$ [6], etc. In fact, [3,7] provided some more general approaches to covering the dependent cases for identifiability, which might be helpful to relax the assumptions in this paper.

Another concern is that it seems in the paper that the authors did not explicitly provide the out-of-distribution generalization guarantees, i.e., some theoretical guarantees on that the learned invariant predictor can be generalized from the training environments in $\mathcal{E}_{train}$ to all the environments in $\mathcal{E}$. Proposition 4.2 is more of a definition than a proof. It might be better if the authors could provide more details about it.

The notations in Section 4.2 seem a bit messy. For example, what is $q_t$? what is $k_t$? etc. All these appear for the first time and should be explained better. In the last sentence of the caption of Figure 1, is it right that "the read and blue respectively means the invariant and varied values/distributions"? or the opposite?


**References** \
[1] Invariant Risk Minimization. Arjovsky et al. 2019 \
[2] Invariant Risk Minimization Games. Ahuja et al. 2020 \
[3] Nonlinear Invariant Risk Minimization: A Causal Approach. Lu et al. 2021 \
[4] Elements of Causal Inference: Foundations and Learning Algorithms. Peters et al., 2017. \
[5] Domain Adaptation under Target and Conditional Shift. Zhang et al., 2013. \
[6] Self-Supervised Learning with Data Augmentations Provably Isolates Content from Style. Kügelgen et al., 2021 \
[7] ICE-BeeM: Identifiable Conditional Energy-Based Deep Models Based on Nonlinear ICA. Khemakhem et al., 2020

**Time Spent Reviewing:**

two hours

---

> ### Author Response · Authors · 2021-08-09
> **Response to Reviewer nVGX**
>
> Thanks for your valuable comments and efforts in reviewing our manuscript. The addresses to your concerns are listed below.
>
> **About the conditional independence in proving the identifiability.** Thank you for your question! We will discuss this issue in three aspects: i) our theoretical extension to the case when component-wise conditional independence (*i.e.*, $S_i \not\perp S_j|C, D$) do not hold; (ii) the role of conditional independence between $S$ and $Z$ in identifiability; and (iii) the rationality of conditional independence assumption in the causal learning scenario.
>
> First, our theoretical results indeed can be easily extended to the case when each component of $S$ (or $Z$) is not necessarily conditionally independent. As long as $S \perp Z|C,D$, we have $< p^e(s,z|c) = \frac{B(s)}{A^s_{c,d^e}}\exp(\langle \mathbf{T}^s(s), \mathbf{\Gamma}_{c,d^e}(s)  \rangle)*\frac{B(z)}{A^z_{c,d^e}} \exp(\langle \mathbf{T}^z(z),\mathbf{\Gamma}_{c,d^e}(z) \rangle). >$
>
> With assumption 5 similarly made on $<\mathbf{\Gamma}_{c,d^e}(s), \mathbf{\Gamma}_{c,d^e}(z)>$, we can achieve the disentanglement of $S$ and $Z$ (*i.e.*, $< \tilde{\mathcal{T}}^s([\tilde{f}^{-1}_x(x)]_{\mathcal{S}}) = M_s\mathcal{T}^s([f^{-1}_x(x)]_{\mathcal{S}})>$ and $<\tilde{\mathcal{T}}^z([\tilde{f}^{-1}_x(x)]_{\mathcal{Z}}) = M_z\mathcal{T}^z([f^{-1}_x(x)]_{\mathcal{Z}}) >$ ), by following from similar derivations in theorem B.5 in appendix. We will append this result in the updated version.
>
> Next, as far as our concerned, it might be necessary to assume conditional independence (between $S$ and $Z$) for our identifiability result to hold. Different from [6,7] ([3,7] in your references), the main goal of our identifiability is disentangling $S$ from $Z$ (*i.e.*, the function of $S$ does not depend on $Z$, in Eq.(2)), which was lacked in the current literature of identifiability. If the $S$ are causally related to $Z$ ($S \to Z$ or $Z \to S$), then $Z$ (or $S$) is a function of $S$ (or $Z$), which contradicts the goal of disentanglement.
>
>
> Finally, please allow me to say something about conditional independence, from its motivation and its relation to the learning method.
> * First, this assumption is motivated by the observations that causal factors (*i.e.*, $S$) are often spuriously correlated with contextual features (*i.e.*, $Z$) (as described in lines 41-42, lines 126-133, and remark 1). Such contextual features, as refer to the background/texture features in image classification [1,2]; hypothesis in QA task [3,4], were often mixed with causal factors during learning.
> Therefore, although this scenario cannot cover all types of distributional shifts for certain (and we did \emph{not} claim that we can), it exists widely in many tasks and is never explored in the literature of causal learning (as claimed in remark 1). In the above tasks, it is unreasonable to causally related the $Z$ with $Y$ or $S$, as the $Z$ are expected not to be learned for prediction. Instead, the reason lies in the statistical correlation of it with $S$, and this correlation can be broken in unseen domains, *i.e.*, the dog is more associated with the grass in dataset A but this association does not hold in dataset B (as claimed in lines 132-133). Statistically, we should relate the contextual features $Z$ with $S$ via the confounder.
> *Next, we discuss its relation to the learning method. Together with other causal assumptions embedded in LaCIM, this conditional independence can result in the identifiability conclusion, *i.e.*, the isolation of causal factors for prediction. Then the proposed VAE is guaranteed to recover the true latent factors as long as it can fit $p^e(x,y)$ well.
> *Besides, the causal graphs that embed the causal assumptions [5], depending on the real problem and the understanding of generating process. If the causal graph changes, the learning method should be correspondingly redesigned. This may be the reason why there is no single learning method in the literature that can unify all types of causal assumptions, to the best of our knowledge. Although other scenarios of distributional shifts are interesting and are often met in many applications, they do not belong to the scope in this paper and we will study them in the future.
>
>
> **About the out-of-distribution generalization.** We have left the OOD generalization analysis in theorem A.1 in appendix. As shown, the distributional shifts mainly come from the inconsistency of posteriors $p^e(s,z|x)$ across environments. To see how our method can amend this problem, note that our proposed generative model can consistently learn $\{p^e(x,y)\}$ and $p_{f_x}(x|s,z)$ and $p_{f_y}(y|s)$. Then we can recover the causal factor (up to permutation) and the ground-truth predictor, as respectively guaranteed by Eq. (2) and Eq. (3) in Def. 4.3 and theorem 4.4. Specifically, as shown in Eq. (3), if we denote the $\tilde{f}_y$ and $\tilde{f}_x$ as the estimated functions, then for a sample $x \gets f_x(s^\star,z^\star)$, we can obtain that $<p_{\tilde{f}_y}(\tilde{s}) = p_{f_y}(s^\star)>$. This motivates us to estimate $s$ via maximizing $<p_{f_x}(x|s,z)>$ and predict with $<p_{f_y}(y|s)>$. Therefore, the OOD generalization naturally holds.
>
> **About Proposition 4.2.** This proposition can be directly obtained from the invariance of $f_x, f_y$ and distributions of $\varepsilon_x, \varepsilon_y$ across environments $\mathcal{E}$ in Def. 4.1. Guided by this proposition, we proposed to estimate the causal factor $s$ via maximizing $p_{f_x}(x|s,z)$ over $\mathcal{S} \times \mathcal{Z}$ and use it for prediction via $p(y|s)$, since they are invariant across domains.
>
> **About Notations.** The $q_t$ denotes the dimension of $t$ ($t$ can denote variable $S$ or variable $Z$), while $k_t$ denotes the number of natural parameters for each dimension of $t$. Yes, the red and blue respectively means the invariant and varied values/distributions. Sorry for the misunderstanding, we will make our notations clearer in the updated version.
>
> ## **References.**
> [1] Jo, Jason, and Yoshua Bengio. "Measuring the tendency of CNNs to Learn Surface Statistical Regularities." arXiv e-prints (2017): arXiv-1711.
>
> [2] Geirhos, Robert, et al. "ImageNet-trained CNNs are biased towards texture; increasing shape bias improves accuracy and robustness." International Conference on Learning Representations. 2018.
>
> [3] Poliak, Adam, et al. "Hypothesis Only Baselines in Natural Language Inference." Proceedings of the Seventh Joint Conference on Lexical and Computational Semantics. 2018.
>
> [4] Gururangan, Suchin, et al. "Annotation Artifacts in Natural Language Inference Data." Proceedings of the 2018 Conference of the North American Chapter of the Association for Computational Linguistics: Human Language Technologies, Volume 2 (Short Papers). 2018.
>
> [5] Pearl, Judea. "Causal inference in statistics: An overview." Statistics surveys 3 (2009): 96-146.
>
> [6] Lu, Chaochao, et al. "Nonlinear Invariant Risk Minimization: A Causal Approach." arXiv preprint arXiv:2102.12353 (2021).
>
> [7] Khemakhem, Ilyes, et al. "ICE-BeeM: Identifiable Conditional Energy-Based Deep Models Based on Nonlinear ICA." Advances in Neural Information Processing Systems 33 (2020).

---

### Decision · Program_Chairs · 2021-09-27

**Decision:**

Accept (Poster)

**Comment:**

This paper investigates the problem that distribution shift may degrade the prediction accuracy of the learned model. It proposes a way to distinguish between the underlying causal factors and other factors, together with a strategy to deal with distribution shift by using only causal factors for prediction. The studied problem is interesting, and reviewers agree that both theoretical studies and empirical results are convincing.